# Complete Genome Sequences and Genome-Wide Characterization of *Trichoderma* Biocontrol Agents Provide New Insights into their Evolution and Variation in Genome Organization, Sexual Development, and Fungal-Plant Interactions

Wan-Chen Li,[a] Ting-Chan Lin,[a,b] Chia-Ling Chen,[a] Hou-Cheng Liu,[a] Hisn-Nan Lin,[a] Ju-Lan Chao,[a] Cheng-Hsilin Hsieh,[a] Hui-Fang Ni,[c] Ruey-Shyang Chen,[b] 🄳 Ting-Fang Wang[a,b]

[a]Institute of Molecular Biology, Academia Sinica, Taipei, Taiwan
[b]Department of Biochemical Science and Technology, National Chiayi University, Chiayi, Taiwan
[c]Department of Plant Protection, Chiayi Agricultural Experiment Station, Council of Agriculture, Chiayi, Taiwan

**ABSTRACT** *Trichoderma* spp. represent one of the most important fungal genera to mankind and in natural environments. The genus harbors prolific producers of wood-decaying enzymes, biocontrol agents against plant pathogens, plant-growth-promoting biofertilizers, as well as model organisms for studying fungal-plant-plant pathogen interactions. Pursuing highly accurate, contiguous, and chromosome-level reference genomes has become a primary goal of fungal research communities. Here, we report the chromosome-level genomic sequences and whole-genome annotation data sets of four strains used as biocontrol agents or biofertilizers (*Trichoderma virens* Gv29-8, *Trichoderma virens* FT-333, *Trichoderma asperellum* FT-101, and *Trichoderma atroviride* P1). Our results provide comprehensive categorization, correct positioning, and evolutionary detail of both nuclear and mitochondrial genomes, including telomeres, AT-rich blocks, centromeres, transposons, mating-type loci, nuclear-encoded mitochondrial sequences, as well as many new secondary metabolic and carbohydrate-active enzyme gene clusters. We have also identified evolutionarily conserved core genes contributing to plant-fungal interactions, as well as variations potentially linked to key behavioral traits such as sex, genome defense, secondary metabolism, and mycoparasitism. The genomic resources we provide herein significantly extend our knowledge not only of this economically important fungal genus, but also fungal evolution and basic biology in general.

**IMPORTANCE** Telomere-to-telomere and gapless reference genome assemblies are necessary to ensure that all genomic variants are studied and discovered, including centromeres, telomeres, AT-rich blocks, mating type loci, biosynthetic, and metabolic gene clusters. Here, we applied long-range sequencing technologies to determine the near-completed genome sequences of four widely used biocontrol agents or biofertilizers: *Trichoderma virens* Gv29-8 and FT-333, *Trichoderma asperellum* FT-101, and *Trichoderma atroviride* P1. Like those of three *Trichoderma reesei* wild isolates [QM6a, CBS999.97(*MAT1-1*) and CBS999.97(*MAT1-2*)] we reported previously, these four biocontrol agent genomes each contain seven nuclear chromosomes and a circular mitochondrial genome. Substantial intraspecies and intragenus diversities are also discovered, including single nucleotide polymorphisms, chromosome shuffling, as well as genomic relics derived from historical transposition events and repeat-induced point (RIP) mutations.

**KEYWORDS** biocontrol, CAZyme, comparative genomics, fungal-plant interaction, gene cluster, genome annotation, genome sequences, secondary metabolism biosynthesis, sexual development, *Trichoderma*

Address correspondence to Ruey-Shyang Chen, rschen@mail.ncyu.edu.tw, or Ting-Fang Wang, tfwang@gate.sinica.edu.tw.

*T*richoderma (Hypocreales, Ascomycota) species are highly successful colonizers of the rhizosphere (as mycoparasites or versatile symbionts) or wherever decaying plant material is available. They are easy to culture, grow rapidly, and often outgrow or even attach to other microbes encountered in their natural habitats. Although some *Trichoderma* species are of clinical significance, many more are beneficial to humans and in natural processes. For example, the *Trichoderma reesei* QM6a wild isolate and its cellulase-producing mutants are widely used for commercial production of enzymes that degrade plant cell walls or to generate therapeutic proteins (1–3). Several *Trichoderma* spp. are also used in agriculture as biocontrol agents (or biopesticides) against plant pathogens or as plant-growth-promoting biofertilizers, including *T. asperellum*, *T. atroviride*, *T. koningi*, *T. harzianum*, *T. rossicum*, and *T. virens*. *Trichoderma* spp. also have long been model organisms for studying molecular mechanisms underlying industrial enzyme production, secondary metabolite (SM) biosynthesis, effector-like proteins, mycoparasitism, fungal-plant pathogen interactions and asexual sporulation (conidiation) in filamentous fungi (1–16). Recently, *T. reesei* has become an important emerging model filamentous fungus for studying ascomycete mating, meiosis, post-meiotic mitosis and repeat-induced point (RIP) mutation (17–27). RIP is a fungus-specific genome-level defense mechanism against mobile elements or transposons. RIP occurs premeiotically and targets duplicated sequences and causes frequent conversion of C:G base pairs to T:A within such duplicated sequences (28–30).

About 3 decades ago, the *Trichoderma* genus was divided into five taxonomic sections: *Hypocreanum*, *Longibrachiatum*, *Pachybasium*, *Saturnisporum*, and *Trichoderma* (31). Thanks to the rapid development of next-generation sequencing (NGS) and genome-wide annotation technology, researchers have gained more insights into the genomes of several *Trichoderma* species, including plant-cell-wall degradation enzymes, effector-like proteins and genes putatively involved in the biosynthesis of SMs (2, 3, 10, 32–37). Because the lengths of NGS reads are too short to exclude unidentified nucleotides and assembly errors, further validation is required before these NGS-based genome sequences and resulting annotation data sets can be used systematically. In a worst-case scenario, false-positive assemblies can result in incorrect assignments of gene gain or loss. NGS-based genome sequences also cannot reveal accurate information about genome synteny, diversity, or evolution (21, 38).

To obtain highly accurate and chromosome-level reference genomes to explore important biological questions, we previously applied third-generation sequencing (TGS) technology and the "Funannotate" gene-prediction tool to determine and annotate the nearly complete genome sequences of three *T. reesei* wild isolates, i.e., QM6a, CBS999.97(*MAT1-1*) and CBS999.97(*MAT1-2*) (21, 26, 27). Here, we further determined and annotated the nearly complete genome sequences of four *Trichoderma* biocontrol agents, *Trichoderma virens* Gv29-8 and FT-333, *Trichoderma asperellum* FT-101 and *Trichoderma atroviride* P1 (https://github.com/tfwangasimb/Trichoderma-biocontrol/releases; supplemental information [SI], Table S1). Hereafter, CBS999.97(*MAT1-1*) and CBS999.97(*MAT1-2*) and the four biocontrol agents are referred to as CBS1-1, CBS1-2, Gv29-8, FT-333, FT-101, and P1, respectively. We demonstrate in this study that these highest-quality genome resources represent powerful tools for comparative multiomics analyses of these economically important *Trichoderma* spp.

## RESULTS

**General properties of the nearly complete genome sequences of seven *Trichoderma* strains.** High-quality and chromosome-level genome sequences of Gv29-8, FT-333, FT-101, and P1 were determined by TGS technology, before performing genome-wide annotations (Table 1, Tables S2–S5 and supplemental data sets [SD], DS1–DS8). Like those of *T. reesei* wild isolates (21, 26, 38, 39), the nearly complete genomes of Gv29-8, FT-333, FT-101, and P1, each harbors seven telomere-to-telomere nuclear chromosomes (Fig. 1).

**Telomeres, subtelomeres, AT-rich blocks, retrotransposons, and centromeres.** Using our assembled telomere-to-telomere genomes, we could assess and compare their structural components at fine-scale resolution. Telomeric repeats are evolutionarily conserved

**TABLE 1** Summary of the annotated genes in the seven nearly complete *Trichoderma* genome sequences

| Species | *T. reesei* CBS999.97 | *T. reesei* CBS999.97 | *T. reesei* | *T. virens* | *T. virens* | *T. asperellum* | *T. atroviride* |
|---|---|---|---|---|---|---|---|
| Strain | (*MAT1-1*) | (*MAT1-2*) | QM6a | Gv29-8 | FT-333 | FT-101 | P1 |
| Sequencing technology | PacBio | PacBio | PacBio | Oxford Nanopore | Oxford Nanopore | PacBio | Oxford Nanopore |
| Locus_tag | TRC1 | TRC2 | TrQ | TrV | TrVFT-333 | TrA | TrAt |
| Genome size (base pairs; bps) | 34,319,199 | 34,324,311 | 34,922,528 | 40,979,523 | 41,418,917 | 37,545,380 | 37,300,646 |
| No. chromosomes | 7 | 7 | 7 | 7 | 7 | 7 | 7 |
| N50 (bps) | 5,258,134 | 5,262,578 | 5,311,445 | 6,490,838 | 6,644,895 | 5,512,738 | 5,658,044 |
| GC (%) | 51.63 | 51.63 | 51.08 | 47.35 | 47.07 | 47.06 | 48.72 |
| No. of AT-blocks (≥500 bps)[a] | 2259 | 2250 | 2249 | 3577 | 3367 | 4570 | 5510 |
| Length of AT-rich blocks (kbs) | 2,530 | 2,510 | 3,125 | 4,641 | 4,856 | 5,167 | 4,813 |
| Percentage of AT-rich blocks (%) | 7.37 | 7.77 | 8.95 | 11.33 | 11.72 | 13.76 | 12.90 |
| Mitogenome size (bps) | 38,995 | 39,005 | 42,130 | 27,943 | 31,081 | 30,285 | 29,981 |
| BUSCO genome metrics (%)[b] | 99.3 | 99.3 | 99.3 | 97.8 | 96.6 | 98.5 | 97.1 |
| Single complete (S) % | S:99.3 | S:99.3 | S:99.3 | S: 97.4 | S:95.9 | S:98.5 | S:96.8 |
| Duplicated complete (D) % | D:0.0 | D:0.0 | D:0.0 | D:0.4 | D:0.7 | D:0.0 | D:0.3 |
| Fragment (F) % | F:0.0 | F:0.0 | F:0.0 | F:0.3 | F:1.2 | F:0.1 | F:0.8 |
| Missing (M) % | M:0.7 | M:0.7 | M:0.7 | M:1.9 | M:2.2 | M:1.4 | M:2.1 |
| BUSCO protein metrics (%)[b] | 99.2 | 98.9 | 98.4 | 95.5 | 94.2 | 98.6 | 90.8 |
| Single complete (S) % | S:93.5 | S:92.2 | S:91.8 | S: 95.1 | S:93.7 | S:96.2 | S: 88.4 |
| Duplicated complete (D) % | D:5.7 | D:6.7 | D:6.6 | D:0.4 | D:0.5 | D:2.4 | D:2.4 |
| Fragment (F) % | F:0.3 | F:0.3 | F:0.3 | F:1.6 | F:2.5 | F:0.8 | F:5.9 |
| Missing (M) % | M:0.5 | M:0.8 | M:1.3 | M:2.9 | M:3.3 | M:0.6 | M:3.3 |
| Predicted protein-coding genes | 10,292 | 10,225 | 10,238 | 12,263 | 11,895 | 12,041 | 13,327 |
| Predicted proteins | 11,090 | 11,087 | 11,038 | 12,064 | 11,698 | 12,454 | 13,583 |
| tRNA genes | 150 | 144 | 159 | 202 | 200 | 184 | 185 |
| Predicted gene clusters | 32 | 32 | 31 | 57 | 57 | 54 | 45 |
| Transcription Factors (TF) | 691 | 710 | 680 | 739 | 641 | 882 | 843 |
| HET domains (PF06985) | 41 | 42 | 44 | 68 | 60 | 55 | 75 |
| Ankyrins (PF00023) | 79 | 81 | 75 | 117 | 137 | 105 | 114 |
| CAZymes[c] | | | | | | | |
| Auxiliary activity (AA) | 48 | 51 | 49 | 66 | 58 | 54 | 58 |
| Carbohydrate-binding modules (CBM) | 11 | 11 | 12 | 16 | 14 | 13 | 13 |
| Carbohydrate esterases (CE) | 33 | 32 | 33 | 47 | 42 | 43 | 39 |
| Glycoside hydrolases (GH) | 199 | 198 | 199 | 230 | 222 | 240 | 229 |
| Glycosyl transferases (GT) | 89 | 91 | 88 | 71 | 72 | 77 | 77 |
| Polysaccharide lyases (PL) | 7 | 7 | 7 | 7 | 7 | 8 | 9 |
| CAZ-GCs | 31 | 31 | 29 | 35 | 26 | 31 | 33 |
| Predicted proteins with signal peptides (SP) | | | | | | | |
| Total secretory signal peptides[d] | 866 | 874 | 840 | 1,072 | 995 | 1,050 | 1,041 |
| Oxidoreductases (GO:0016491) | 43 | 42 | 45 | 61 | 60 | 50 | 54 |
| Hydrolases (GO:0016787) | 203 | 197 | 202 | 247 | 225 | 253 | 250 |
| Transferases (GO:0016740) | 10 | 11 | 11 | 9 | 8 | 11 | 9 |
| Catalytic activity (GO:0003824) | 284 | 280 | 282 | 357 | 328 | 347 | 350 |
| Lysases (GO:0016829) | 2 | 2 | 2 | 3 | 4 | 3 | 4 |
| Ligases (GO:0016874) | 0 | 1 | 0 | 0 | 0 | 0 | 1 |
| Isomerases (GO:0016853) | 5 | 5 | 5 | 5 | 5 | 5 | 7 |
| Peptidoglycan muralytic activity (GO:0061783) | 2 | 2 | 2 | 2 | 2 | 0 | 1 |
| Secondary metabolite biosynthesis (SMB) | | | | | | | |
| NRPS | 7 | 7 | 11 | 17 | 22 | 15 | 7 |
| PKS/NRPS-like proteins | 17 | 19 | 18 | 32 | 40 | 38 | 30 |
| Hybrid PKS-NRPS | 4 | 4 | 1 | 11 | 12 | 5 | 4 |
| NRPS-like proteins | 8 | 8 | 7 | 16 | 16 | 19 | 22 |
| Type I Iterative PKS | 10 | 10 | 10 | 16 | 15 | 12 | 16 |
| PKS-like proteins | 0 | 0 | 0 | 0 | 1 | 0 | 0 |
| Cytochrome P450 (CYP) PF00067 | 79 | 80 | 71 | 104 | 87 | 72 | 67 |
| SM-BGC[e] | 32 | 32 | 32 | 58 | 57 | 52 | 46 |

[a]As described previously (21), AT-rich blocks with GC contents ≥12% and ≥6% lower than the average GC content of all predicted genes (56.5%) and the entire QM6a genome (51.1%), respectively.

[b]Gene annotation completeness was evaluated in BUSCO (v4.1.4) using the database for fungi_odb10.

[c]The CAZyme genes were determined by using the dbCAN2 meta server (http://bcb.unl.edu/dbCAN2) (96).

[d]The *SignalP* server was used to predict the presence and location of signal peptide cleavage sites in amino acid sequences (56).

[e]The gene clusters were determined by using the antiSMASH software tool.

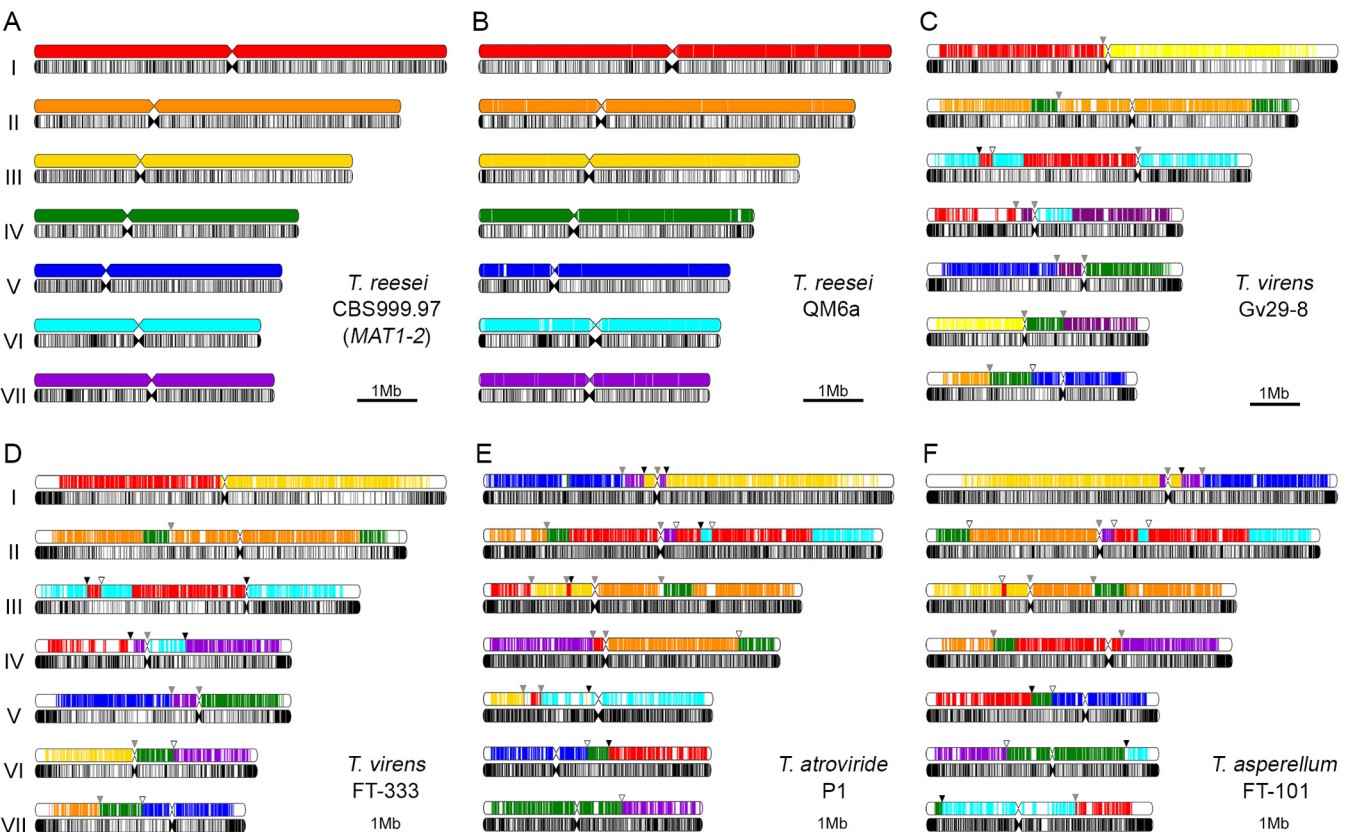

**FIG 1** Diagrammatic representations of the seven chromosomes of CBS1-2 (A), QM6a (B), Gv29-8 (C), FT-333 (D), P1 (E), and FT-101 (F). For comparative genome analyses, we identified orthologous gene pairs using the annotation results generated by Funannotate v1.8 (91) (supplemental data set DS8). The colors of chromosome fragments represent orthologous proteins consistent with their colors in CBS1-2 to clearly show chromosomal rearrangements. Locations of predicted centromeres are shown by restricted width. White fragments in QM6a, Gv29-8, FT-333, P1, and FT-101 represent strain-specific sequences that do not exist in CBS1-2, respectively. Locations of AT-rich blocks are indicated by black bars in the lower chromosomal maps. Black, white and gray arrows indicate disruption of synteny occurring at AT-rich blocks of the subject genomes, the CBS1-2 genome, or both, respectively.

among these seven *Trichoderma* genomes, i.e., TTAGGG at the 3′-termini and the reverse complement CCCTAA at the 5′-termini (21, 32). Compared with those of CBS1-2, nearly all subtelomeres of the four *Trichoderma* biocontrol agent genomes are hypervariable and contain species-specific sequences (Fig. 1). New subtelomeric variants can be created by distinct mechanisms, including alternative lengthening of telomeres (ALT) and break-induced replication (BIR). ALT can lengthen telomeres without utilizing telomerase. It is a homologous recombination (HR)-based process that involves copying telomeric DNA template (40, 41). During BIR, homologous templates from either the same chromosome or even a nonallelic region can be used for template replication and to establish new subtelomeres (42, 43). A typical example of BIR in *T. reesei* is the reciprocal exchange between the right arm terminus of ChII in CBS1-1 or the right arm terminus of ChIV in CBS1-2 and QM6a, respectively (Fig. S1) (20, 26, 27). Another example is the subtelomeric fragment in the ChI left arm in FT-333. This subtelomeric fragment in Gv29-8 has relocated to the interior of ChIV, i.e., about a quarter of the chromosome's length from the terminus of its right arm (Fig. S2B).

Compared with the genomes of the three *T. reesei* wild isolates, those of the four biocontrol agents contain more interspersed AT-rich blocks (≥500 bp). The respective overall numbers and genomic contents are 2,249 and 8.95% (QM6a), 2,259 and 7.37% (CBS1-1), 2,250 and 7.77% (CBS1-2) (26, 38), 3,577 and 11.33% (Gv29-8), 3,367 and 11.72% (FT-333), 4,570 and 13.76% (FT-101), and 5,510 and 12.90% (P1). These results are compatible with the average genomic GC contents of Gv29-8, FT-333, FT-101, and P1, all of which are lower than those of the three *T. reesei* strains (Table 1). The higher numbers of AT-rich blocks in these four *Trichoderma* species also partly account for their larger genome sizes. Because the numbers of AT-rich blocks of different lengths

in *Neurospora crassa* (Sordarales, Ascomycota) and QM6a (21) are similar to those in CBS1-1, CBS1-2, and the four biocontrol agents (Fig. S3), all seven *Trichoderma* genomes have undergone extensive transposon invasions followed by RIP mutations.

The overall numbers of authentic transposons (i.e., which have not been extensively mutated or degenerated by RIP) in these seven genomes are 70 (QM6a), 62 (CBS1-1), 62 (CBS1-2) (27, 38), 93 (Gv29-8), 94 (FT-333), 92 (FT-101), and 78 (P1), respectively (Fig. 1 and Table S2). Because CBS1-1 and CBS1-2 were derived from two ascospores of a heterothallic CBS999.97 fruiting body, their genomes had undergone at least one round of RIP during sexual development (17, 27). In contrast, QM6a, Gv29-8, FT-333, FT-101, and P1 have been propagated asexually since they were isolated. Therefore, the two CBS999.97 strains contain fewer authentic transposable elements.

As in the filamentous fungal model organism *Neurospora crassa* (Sordariales, Ascomycota) (44–46) and QM6a (21), the putative centromeric loci in all seven *Trichoderma* genomes are the longest AT-rich blocks and the longest regions of each chromosome lacking an open-reading frame (ORF) or putative protein-encoding genes (Fig. S4–S8 and Table S3). Using BLASTN with an E value of 1e-8 (identity >80%) (47), all putative centromeric loci contain an array of repeats that are either short repetitive sequences or the relics derived from historical transposition and RIP events. Notably, there are no or very few authentic transposons in all putative centromeres (Fig. S4–S8 and Table S2). This scenario is consistent with our recent finding that all seven putative centromeres of QM6a and CBS1-1 generated no or only a few RIP mutations upon sexual crossing (26).

**Extensive gross chromosome rearrangements between the genomes of different *Trichoderma* species.** Conserved synteny between the nearly complete *Trichoderma* genomes was revealed by ideogrammatic representations (Fig. 1 and Fig. S1) or by CIRCOS plots (48) (Fig. S2). Genomes of the same species (i.e., QM6a, CBS1-2, and CBS1-2 or Gv29-8 and FT-333) or of the same section/clade (i.e., P1 and FT-101) display a higher degree of chromosome synteny than when different *Trichoderma* species or sections are compared, respectively (Fig. S2). Disruption of synteny often (but not always) occurs at long AT-rich blocks or even within centromeres (Fig. 1) (see Discussion).

**Highly divergent genomic contents of four *Trichoderma* species.** It was reported previously that the core genome derived from 14 of the most common *Trichoderma* species contains ~7,000 genes (37). Our annotation results (SD, DS2–DS8) indicate that four representative *Trichoderma* species (CBS1-2, Gv29-8, P1, and FT-101) share 7,202 core protein-encoding genes. The four biocontrol agents (Gv29-8, FT-333, FT-101, and P1) possess an additional ~800 conserved genes. The genomes of CBS1-2, Gv29-8, P1, and FT-101 each encodes 2,152, 2,439, 3,779, or 2,555 species-specific genes, equivalent to one-fifth to one-third of their overall protein-encoding genes (Fig. 2A). Gene ontology (GO) analyses further revealed that only 25% to 33% of the species-specific genes encode functionally annotated proteins (Fig. S9). Although the three *T. reesei* genomes have the smallest genome sizes and encode the lowest overall numbers of protein-encoding genes and tRNA genes, the Benchmarking Universal Single-Copy Ortholog (BUSCO) protein metrics of QM6a, CBS1-1, CBS1-2, and FT-101 are higher than those of Gv29-8, FT-333 and P1 (Table 1) (see Discussion).

**Mating type loci and sexual development genes.** CBS1-1 and CBS1-2 are sexually competent. In contrast, like QM6a, Gv29-8, FT-333, FT-101, and P1 have been propagated asexually since they were isolated. To validate the accuracy of our genome annotation results, we first confirmed that only the *ham5* gene in QM6a encodes a truncated protein (19) (Fig. S10). We then surveyed ~160 gene orthologs in CBS1-2 and/or three other filamentous fungal model organisms [*Neurospora crassa*, *Sordaria macrospora* (Sphaeriales, Ascomycota), *Saccharomyces cerevisiae* (Saccharomycetales, Ascomycota)] (SD, DS9). All these gene orthologs have been implicated as being involved in or even essential to fungal sexual development (see reviews of 2, 3, 49–52), and their annotated functions are described in SD, DS9. The seven *Trichoderma* genomes encode nearly all of the normal protein homologs deemed to play roles in sexual mating signaling systems (e.g., pheromones, light, cell communication, and hyphal fusion), RIP, quelling, meiotic silencing by unpaired DNA (MSUD), meiotic DNA recombination, chromosome individualization or condensation,

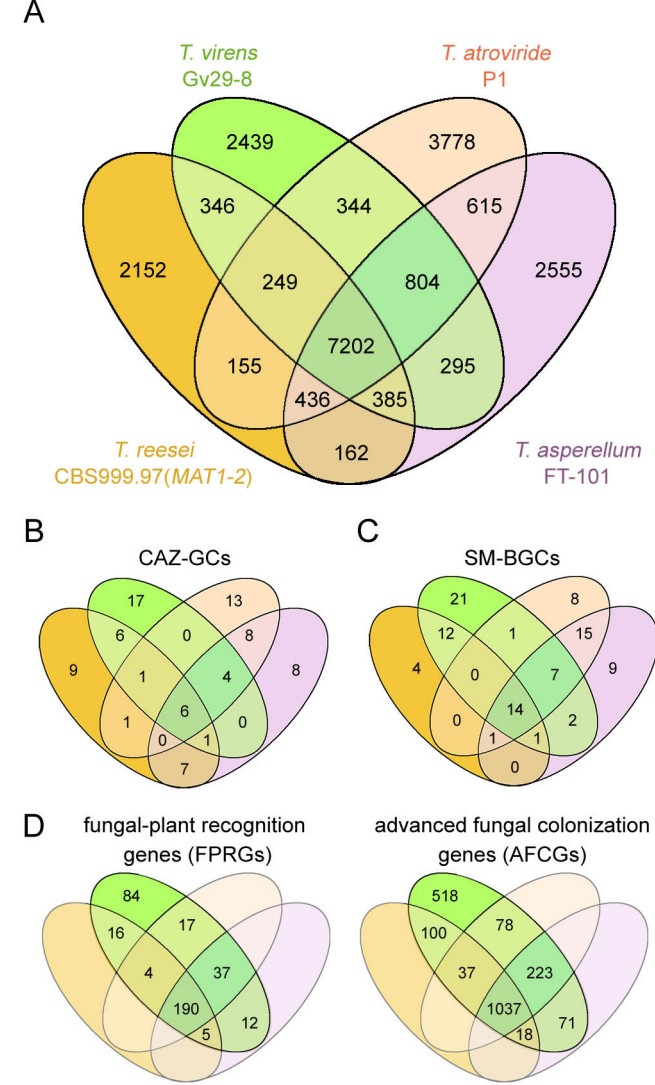

**FIG 2** Comparison of the predicted proteomes encoded by the near complete genome sequences of CBS1-2, Gv29-8, P1, and FT-101. Venn diagram of the overall number of annotated protein-encoding genes (A), the number of CAZ-GCs (B), the number of SM-BGCs (C), and evolutionarily conserved FPRGs and AFCGs (D) (see *SD*, DS12-14).

sister chromatid cohesion, chromosome synapsis, as well as the formation of fruiting bodies (SD, DS9). P1 and Gv29-8, like QM6a, each possesses a *MAT1-2* locus with a normal *mat1-2-1* gene. FT-101, like CBS1-1, has a normal *MAT1-1* locus with three mating type genes: *mat1-1-1*, *mat1-1-2*, and *mat1-1-3*. The *MAT1-1* locus of FT-333 only comprises *mat1-1-1* and *mat1-1-2*, but not *mat1-1-3* (Fig. 3). The genomes of QM6a, CBS1-1, and CBS1-2 all possess an ortholog of *N. crassa male barren-3* (*mb-3*) (53). We were unable to identify or annotate this gene in the genomes of Gv29-8, FT-333, FT-101, or P1. Further investigations are needed to confirm whether Gv29-8, FT-333, FT-101, and P1 exhibit a male barren phenotype or if the *mb-3*-like gene in *T. reesei* is essential for male fertility. Interestingly, Gv29-8, FT-333, and P1, but not FT-101, also lack an ortholog of *sad3*, a gene essential for meiotic silencing by unpaired DNA (MUSD) and normal sexual development in *N. crassa* (54) (SD, DS9). MUSD, an RNA interference (RNAi)-related genome defense mechanism, occurs in prophase I of meiosis when unpaired DNA sequences are present and leads to the silencing of all homologous genes in the diploid ascus cell (54).

**Transcription factors.** In terms of overall numbers of transcription factors (TF) genes, the ranking is FT-101 > P1 > Gv29-8 > CBS1-2 > CBS1-1 > QM6a > FT-333 (Table 1). The

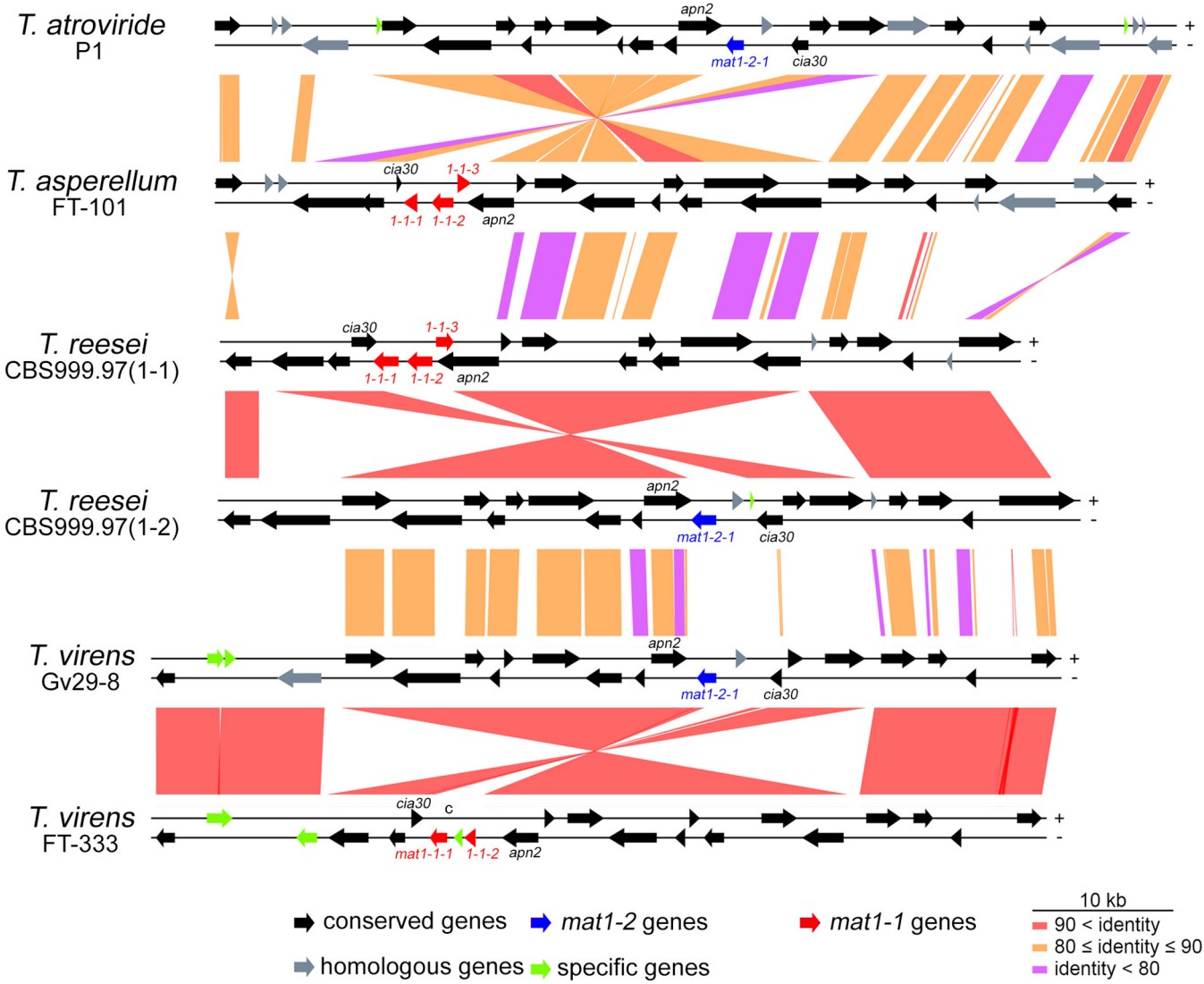

**FIG 3** Comparison of the nucleotide sequences within and around the mating-type loci in P1, FT-101, CBS1-1, CBS1-2, Gv29-8 and FT-333. The tracks between two strains are color-coded to indicate nucleotide sequence identity. The mating-type genes (*mat1–1-1*, *mat1–1-2*, *mat1–1-3* and *mat1–2-1*) are dissimilar in sequence, but they are found at the same loci on the third chromosomes and are all flanked by two evolutionarily conserved genes, the DNA lyase *apn2* and the complex I intermediate-associated protein 30 gene *cia30*.

main differences are due to specific gain or loss of two transcription factor subfamilies, i.e., the fungal Zn(2)-Cys(6) binuclear cluster domain subfamily (InterPro identity: 111138) and the fungal-specific TF domain family (InterPro identity: 007219). Compared with the genomes of Gv29-8, FT-333, FT-101, and P1, the three *T. reesei* genomes possess a species-specific PAS-fold protein, but lack a helix-turn-helix TF (Table S4). The physiological functions of these unique TFs need to be further explored. It is important to note that gene numbers of other TF families are nearly identical among the seven genomes. Thus, these TFs constitute the "core" transcriptional regulatory circuitry of the genus *Trichoderma*.

**Predicted signal peptide proteins.** Signal peptides (SPs) are short amino acid sequences in the amino termini of many newly synthesized proteins that target to membranes or membrane-embedded export machines. The SignalP server has been widely used to predict the presence and location of SP cleavage sites in amino acid sequences of SP proteins (55, 56). The three *T. reesei* genomes each possesses 840 to 870 SP proteins. In contrast, those of Gv29-8, FT-333, FT-101, and P1 encode 1,072, 995, 1,050, and 1,041 SP proteins, respectively. GO indicates that all seven *Trichoderma* genomes are highly enriched with SP proteins having catalytic, oxidoreductive, or

hydrolytic activities (Table 1). SP proteins revealed by the SignalP server are not necessarily cleaved or cleavable *in vivo*. Further experimental investigations are needed to validate if they are indeed secreted proteins.

**Carbohydrate-active enzymes and CAZyme gene clusters.** Compared with the three *T. reesei* genomes, those of Gv29-8, FT-333, FT-101, and P1 encode more auxiliary activity (AA) proteins, carbohydrate esterases (CE) and glycoside hydrolases (GH). In contrast, all three *T. reesei* genomes possess more glycosyltransferase (GT) genes than those of Gv29-8, FT-333, FT-101 and P1 (Table 1). Further investigations will be needed to elucidate this unique property of *T. reesei*. There is considerable evidence showing that carbohydrate-active enzymes (CAZymes) cooperate with other CAZymes and signature proteins (e.g., transporters and transcription factors), and that the respective genes tend to form physically-linked CAZyme gene clusters (CAZ-GCs) in polysaccharide utilization loci (PUL) (57). To identify potential CAZ-GCs in these strains, we have developed a high-stringency predictive software that identified 31, 31, 29, 35, 26, 31, and 33 CAZ-GCs in QM6a, CBS1-1, CBS1-2, Gv29-8, FT-333, FT-101, and P1, respectively (Table 1, Table S5, and SD, DS10). These CAZ-GCs have been named according to their chromosomal location, e.g., CAZ-GC 3.2 indicates the second CAZ-GC from the left arm of ChIII. These results provide a comprehensive basis for further exploring CAZymes and PUL in *Trichoderma* spp. (Fig. 4). We identified 9, 17, 13, and eight species-specific CAZ-GCs in CBS1-2, Gv29-8, P1, and FT-101, respectively. Notably, consistent with their phylogenetic relationships, *T. reesei* and *T. virens* share 14 common CAZ-GCs, whereas *T. atroviride* and *T. asperellum* share 18 common CAZ-GCs (Fig. 2B).

**Secondary metabolite biosynthetic genes and gene clusters.** Relative to the three *T. reesei* nuclear genomes, those of P1, FT-101, Gv29-8, and FT-333 have undergone expansions in almost all secondary metabolite biosynthetic genes and gene clusters (SM-BGC) subfamilies, including polyketide synthases (PKSs), nonribosomal peptide synthases (NRPSs) and cytochromes P450 (CYP450s) (Table 1). Using the Antibiotics and Secondary Metabolite Analysis Shell (antiSMASH) software tool (58), we have annotated a variety of SM-BGCs in the seven genomes: 32 in QM6a, 32 in CBS1-1, 32 in CBS1-2, 59 in Gv29-8, 57 in FT-333, 52 in FT-101, and 46 in P1 (Table 1 and Table S5). These SM-BGCs have also been named according to their chromosomal location; for example, SM-BGC 2.3 indicates the third SM-BGC from the left arm of ChII (Fig. 4). The four different *Trichoderma* species only share 14 common SM-BGCs. We identified four, 21, eight, and nine species-specific SM-BGCs in CBS1-2, Gv29-8, P1, and FT-101, respectively. Notably, consistent with their phylogenetic relationships (37), *T. reesei* and *T. virens* share 27 common SM-BGCs, whereas *T. atroviride* and *T. asperellum* share 37 common SM-BGCs (Fig. 2B).

To address the intriguing question of how SM-BGCs form and are regulated, we compared the biosynthetic genes of the five best-characterized SM-BGCs (see reviews of [2, 10]) to explore their commonalities and differences (Table S6). First, BGCs for siderophore (SID), ferrichrome (FRC), and conidial green pigment (CGP) are found in all four *Trichoderma* species we examined here. SID-BGCs and FRC-BGCs exhibit gene order conservation and contain none or only one AT-rich block (>500 bp) (Fig. S11 and S12). There are only three evolutionarily conserved genes in CGP-BGCs. Their neighboring sequences are quite variable and contain more AT-rich blocks. The neighboring genes of CGP-BGC in P1 are more conserved with those in FT-101, but they are completely different from those in Gv29-8 and CBS1-2 (Fig. S13). Second, the sorbicillinoid (SOR)-BGC is *T. reesei*-specific, as only *T. reesei* can secrete yellow sorbicillinoid compounds (59). The 5′ and 3′ termini of SOR-BGC are physically linked to *usk1* and a well-characterized CAZ-GC harboring three CAZyme genes: *axe1* (acetyl xylan esterase), *cip1* (a CBM-containing auxiliary factor), and *cel61a* (previously named endoglucanase IV) (32) (Fig. S14). The entire chromosomal region (i.e., *usk1*-SOR-BGC-*axe1*-*cip1*-*cel61a*) contains 14 protein-encoding genes, but no AT-rich blocks. Although Gv29-8, P1, and FT-101 each possesses eight to 10 orthologs of those 14 protein-encoding genes, their orthologs are scattered across at least five different chromosomes (Fig. S14). Third, FT-101 possesses a terpene cyclase (TrA_010949) that is highly similar in amino acid

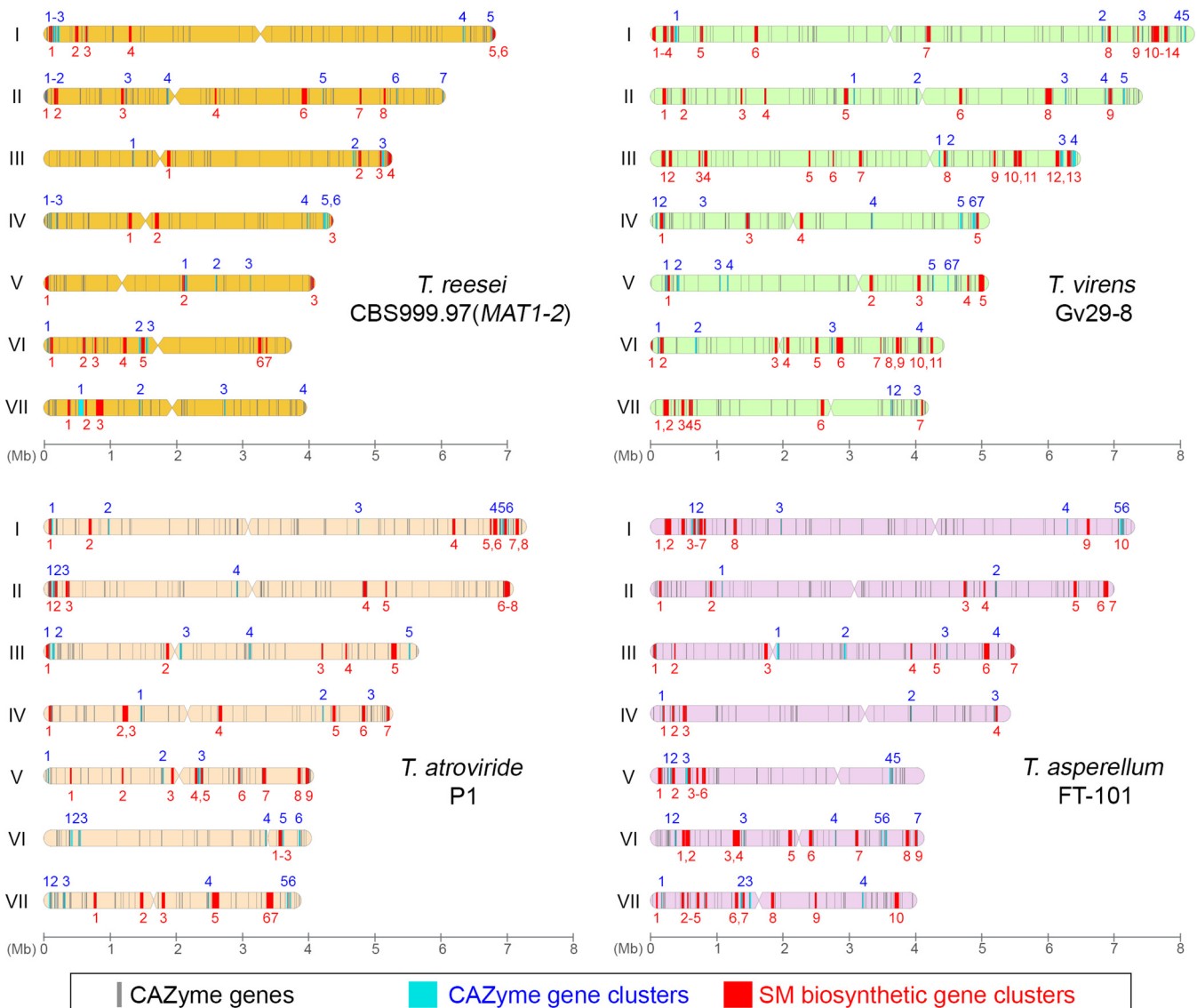

**FIG 4** Locations of CAZyme genes (in gray), CAZ-GCs (in cyan) and SM-BGCs (in red) along chromosomal maps. Chromosome maps of CBS1-2, Gv29-8, P1 and FT-101 are shown. Locations of predicted centromeres are represented by narrowed segments.

sequence to the protein product of the *T. brevicompactum tri5* trichothecene synthase gene (2). This putative *tri5* gene and seven novel genes together constitute a potential trichothecene (TRI) or TRI-like BGC in FT-101 (i.e., SM-BGC-7.3) (Fig. S15). We were unable to identify or annotate this putative *tri5* gene in P1, CBS1-2, or Gv29-8. It is also noteworthy that the protein products of these seven novel genes in FT-101 are different from the biosynthetic proteins encoded by the canonical TRI-BGCs in *T. arundinaceum* and *T. brevicompactum* (60, 61). Interestingly, six orthologs of these seven novel genes also cluster together in P1, whereas CBS1-2 and Gv29-8 each possesses three and five orthologs of these seven novel genes, respectively (Table S6). Fourth, *T. virens* has two species-specific BGCs: viridin (VIR)-BGC and gliotoxin (GTX)-BGC. These two SM-BGCs produce a group of furano-steroidal antibiotics and GTX or GTX-like compounds to compete with and restrict the growth of plant pathogenic fungi, respectively (36, 62). CBS1-2, P1, and FT-101 possess none or only two orthologs of the VIR biosynthetic genes, respectively (Table S6). Both GTX-BGCs in Gv29-8 and FT-333 contain 13 *gli* genes and are located at the far-right subtelomeres of their first chromosomes. Ten long AT-rich blocks separate these 13 *gli* genes. The order of these long

AT-rich blocks is conserved in FT-333 and Gv29-8, but they exhibit highly variable lengths and sequences (Fig. S16A). Although *T. reesei* does not produce GTX during vegetative growth, QM6a and both CBS999.97 strains each possesses a gene cluster with only six and seven *gli* orthologs, respectively, i.e., SM-BGC-6.4 in CBS1-2 (Table S6 and Fig. S16B). *gliH*, an essential GTX biosynthetic gene, does not exist in any of the three *T. reesei* genomes. QM6a also lacks *gliC*, which encodes a CPY450 oxidoreductase that catalyzes the formation of C-S bonds. These C-S bonds constitute the internal disulfide bridge of GTX and other exopolysaccharide-type fungal toxins (63). The order of the *gli* genes in *T. reesei* is different from that in Gv29-8. There are no AT-rich blocks between the *gli* orthologs in *T. reesei* (Fig. S16B). These results suggest that the ancestral genomes of *T. virens* and *T. reesei* might have undergone multistep processes to acquire and/or reorganize their GTX or GTX-like BGCs. Further investigations will be needed to address the physiological roles of the incomplete GTX-BGCs in *T. reesei*.

**T. reesei Sor7 and T. virens GliT are secreted proteins.** Most of our knowledge about the molecular mechanisms of GTX biosynthesis has come from studies of the GTX-BGC in *Aspergillus fumigatus*, an opportunistic fungal pathogen. *A. fumigatus* GliT is an intracellular FAD-dependent dithiol oxidase that converts the reduced GTX (i.e., dithio-GTX) to GTX, thus conferring host self-resistance toward highly reactive dithio-GTX and tolerance to exogenous GTX (64, 65). We found that the GliT proteins in FT-333 hosts an SP, whereas that of *A. fumigatus* does not (Fig. S17). The GliC proteins in FT-333 and Gv29-8, like that of *A. fumigatus*, are predicted to possess signal peptides. In contrast, that of CBS1-2 does not (Fig. S18). Accordingly, the biosynthetic and regulatory mechanisms of GTX- or GTX-like BGCs in *T. virens* and *T. reesei* likely differ from those of *A. fumigatus*. Our annotation results also reveal that the Sor7 FAD-linked oxireductase of *T. reesei* hosts an SP (Fig. S19). Because the SP-hosting proteins predicted by the SignalP server are not necessarily cleaved or cleavable *in vivo*, we applied a proteomics approach to analyze them in the culture filtrate (CFs) of CBS1-2, FT-333, Gv29-8, FT-101, and P1, respectively, including by means of ammonium sulfate precipitation, in-solution trypsin digestion and LC-MS-MS analysis (SI, Materials and Methods). Although most of the secreted proteins we identified in CFs are functionally unannotated, our results confirm that *T. virens* GliT and *T. reesei* Sor7 are indeed secreted proteins (SD, DS11 and Table S7). Further investigations will be needed to reveal the functional roles of these two FAD-linked oxireductases in regulating biosynthesis and the functions of GTX and sorbicillinoids, respectively.

Our proteomics approach also identified two fungal secreted proteins, i.e., *T. virens* small protein 1 (Sml) and *T. asperellum* KatG2 catalase-peroxidase 2. *T. virens* Sm1 is a secreted elicitor-like protein that induces plant defense responses and systemic resistance by triggering production of reactive oxygen species. Sm1 lacks toxic activity against plants and microbes (66). The functional roles of KatG2 in *T. asperellum* remain unclear. KatG2 was originally identified as an extracellular protein in several phytopathogenic fungi (67). In *Fusarium graminearum* (Hypocreales, Ascomycota), KatG2 is exclusively located on the cell wall of invading hyphal cells and contributes to its pathogenicity by alleviating oxidative stress in the vicinity of invasion hyphae (68).

**Genome-wide transcriptomic analyses reveal evolutionarily conserved and diverse genes contributing to fungal-plant interactions.** *Trichoderma* spp. can colonize plant roots, both externally and internally. Induction of plant defense via fungal-plant interactions is considered one of the most important mechanisms of *Trichoderma*-mediated biological control. Michael Kolomiets and colleagues performed RNA sequencing (RNA-seq) analysis on the roots of maize seedlings grown in hydroponic conditions and treated with *T. virens* Gv29-8 at 6 h and 30 h (69, 70). Notably, the time points represented fungal-plant recognition at 6 h and advanced fungal colonization at 30 h. Gv29-8 undergoes global repression of transcription upon recognition of maize roots and then induces expression of a broad spectrum of genes during root colonization (69). Using the complete genome sequence of Gv29-8 as a reference, we reanalyzed the previously acquired transcriptomic data sets (69, 70) to reveal 365 and 2,082 *T. virens* Gv29-8 genes that are transcriptionally upregulated (fold change [FC] $\geq$ 3 and $P <$ 0.05) at the 6-h and 30-h

time points, respectively (Table S8 and SD, DS12 and DS13). The protein products of 365 putative fungal-plant recognition genes (FPRGs) include 42 predicted SP proteins, 13 proteases, 14 CAZymes, and 75 membrane proteins. Three SM-BGCs and one CAZ-GC (1.4) were specifically upregulated at 6 h. The protein products of 2,082 putative advanced fungal colonization genes (AFCGs) include 349 predicted SP proteins, 100 proteases, 158 CAZymes, 433 membrane proteins, and 433 TFs. Almost 80% of SM-BGCs and CAZ-GCs in *T. virens* have at least one gene that was transcriptionally upregulated at 30 h. Notably, most genes in GTX-BGC-1.1 and CPG-BGC-10.1 are AFCGs, whereas all those in VIR-BGC-5.5 and SID-BGC-6.4 are not (Table S8). Further investigations will be needed to reveal whether and why CPG, GTX, or related SMs have nonantibiotic functions during advanced fungal colonization. Moreover, we found by genome-wide BLASTp searches that 190 (52%) FPRGs and 1,037 (50%) AFCGs are evolutionarily conserved in the genomes of CBS1-2, FT-101, P1, and FT-333, respectively (Fig. 2D, Table S9 and SD, DS12-DS14), prompting us to infer that different *Trichoderma* species might utilize both conserved and diverse SMs, proteins or signaling pathways to mediate fungal-plant interactions.

**Mitogenomes and nuclear-encoded mitochondrial sequences.** Our results also reveal the circular mitogenomes of all seven *Trichoderma* strains (SD, DS9). We reported recently that *T. reesei* mitochondria are inherited maternally (27), so the mitogenomes of CBS1-1 and CBS1-2 should be identical. Indeed, we found that their sequence differences were attributable to sequencing read errors in 18 polyhomonucleotide runs. The lengths of the other six mitogenomes vary significantly, although both number and order of protein-encoding genes, tRNAs and rRNA are all well-conserved. We found that mobile genetic elements play key roles in shaping the mitogenomes of *Trichoderma* (Fig. S20–S22 and SD, DS15).

Although our results of *Trichoderma* mitogenomes are consistent with those of previous studies (2, 21, 71, 72), we have discovered that the mitogenomes of Gv29-8 and FT-333 each harbor three nuclear-encoded mitochondrial sequences (NUMTs). NUMTs, first detected in the mouse genome (73), have now been found in several other eukaryotes (74–76). NUMT integration has been implicated in increasing genetic diversity and facilitating genome evolution (77, 78). NUMTs have yet to be explicitly reported among filamentous fungi. Three lines of evidence suggest that the ancestral genome of *T. virens* underwent NUMT integration. First, FT-333 and Gv29-8 each contains three almost identical NUMTs in the subtelomeric regions of the left arm of their second chromosome (indicated by a black line in Fig. S2B). Their sequence lengths and coordinates in FT-333 are 139 bp (211,858 to 211,996), 166 bp (211,996 to 212,161) and 170 bp (115,955 to 116,124), whereas in Gv29-8 they are 146 bp (15,319 to 115,464), 168 bp (115,464 to 115,631), and 170 bp (115,955 to 116,124) (Fig. S23). The corresponding sequences of these three NUMTs localize within two mitochondrial NADH dehydrogenase subunit genes (*nad5* and *nad6*) and a mitochondrial non-coding sequence, respectively (Fig. S21). Second, these three NUMTs are unlikely to represent mitochondrial DNA contamination arising from sequence assembly errors because the corresponding nucleotide sequences were observed in several long raw reads generated by the nanopore sequencer. Third, all three NUMTs in FT-333 and Gv29-8 are located within an AT-rich block ($\sim$1,500 bp in length) that lacks protein-coding sequences (Fig. S24). It was reported previously that mammalian NUMTs tend to be inserted near retrotransposons and that the insertion sites often represent DNA sequences with high DNA "bendability" and lie immediately adjacent to AT-rich sequences (79). Therefore, filamentous fungi and mammalian cells may display an evolutionarily conserved origin for NUMTs (79).

## DISCUSSION

Although >50 different *Trichoderma* genomes have been determined by NGS technology, almost all of the NGS genomic drafts are far from nearly complete or chromosome-level assemblies due to the shortcomings of short NGS reads and the exceedingly low-complexity nature of *Trichoderma* genomes. In contrast, the seven nearly complete *Trichoderma* genomes we have determined by TGS technology represent the highest quality yet achieved. The greatest benefits of precise genome assemblies are

that they enable accurate determination of structural components in each genome (e.g., telomeres, centromeres, interspersed AT-rich blocks and authentic transposable elements), as well as the chromosome synteny between different *Trichoderma* species. First, the results of our comparative genomic analysis reveal that all seven genomes likely underwent extensive transposon invasions followed by RIP mutations (Fig. S3), explaining why the overall numbers of authentic transposons (i.e., which have not been extensively mutated or degenerated by RIP) in these seven genomes are relatively low (Table S2). Second, we reported previously that the longest AT-rich blocks in all seven QM6a chromosomes are likely their centromeres, as they collectively harbor 24 centromere-specific and conserved satellite repeats (21). Here, we further demonstrate that the putative centromeric loci in all seven *Trichoderma* genomes are not only the longest AT-rich blocks but also the longest regions of each chromosome lacking an open-reading frame (ORF) or putative protein-encoding genes (Fig. S4–S8 and Table S3). Third, the genomes of different *Trichoderma* species exhibit extensive chromosome shuffling. Disruption of chromosome synteny often but not always occurs near or within AT-rich blocks or even the predicted centromeres (Fig. 1). One possibility is that mobilization of ancient retrotransposons might induce chromosome rearrangements. Alternatively, chromosome translocations via recombination of AT-rich blocks or centromere scission (80) may drive chromosome shuffling to reshape the genomes of different species. In this regard, retrotransposition and RIP might act as a critical driver of genome reorganization, eventually leading to genome diversification within a species (microevolution) and between species (macroevolution). Consequently, the newly generated strains harboring chromosome translocations may fail to undergo successful sexual reproduction with the parental wild-type strain because extensive chromosomal heterozygosity results in recombination suppression during meiosis, i.e., via nonhomologous pairing, synaptic adjustments or shifts in recombination (or chiasma) positions (81, 82).

Locally, the order of biosynthetic genes, AT-rich blocks, and retrotransposons in SM-BGCs provides detailed insights into the mechanisms of genome evolution. Our data indicate that evolutionarily conserved SM-BGCs often lack or only contain a few longer AT-rich blocks ($\geq$500 bp). In contrast, the GTX-BGCs in FT-333 and Gv29-8 have 10 such blocks. This unique property of GTX-BGCs had not been reported before because NGS sequence reads are too short to assemble chromosomal regions of low sequence complexity (36). We suggest that GTX-BGCs represent versatile systems for future experimental evolution studies to reveal the functional impacts of AT-rich blocks for the expression and regulation of GTX biosynthesis. Also noteworthy is that orthologs of GTX biosynthetic genes are randomly allocated across the genomes of *T. atroviride* and *T. asperellum*. The gene order of the putative GTX-BGC in *T. reesei* also differs from that of the authentic GTX-BGCs in *T. virens* (Fig. S16). We conclude that diverse and/or multistep mechanisms were involved in the formation of different SM-BGCs during evolution (83), such as lateral gene transfer, unequal crossing over, or transposon-mediated gene transfer. Lateral gene transfer (also called horizontal gene transfer) represents the acquisition of genetic material from another organism (e.g., bacteria), whereas unequal crossing over involves the exchange of sequences between chromosomes. Because the GTX-BGCs of both FT-333 and Gv29-8 are located in the subtelomeres of their first chromosomes, it would be interesting to determine if BIR (42, 43) or other molecular mechanisms might play critical roles in generating or maintaining this *T. virens*-specific gene cluster.

High-quality and near complete genome sequences also ensure accurate and comprehensive genome annotation. Based on NGS-based genome sequences, it was reported previously that *T. reesei* and its ancestor might have undergone rapid gene loss relative to several other of the most common *Trichoderma* species (37). Notably, a smaller gene inventory might result from less intensive expansion of certain gene families rather than profound loss of distinct protein-encoding genes. In contrast, our genome-wide annotation data indicate that QM6a, CBS1-1, CBS1-2, and FT-101 encode more near-universal single-copy orthologs (NUSCOs) than the genomes of P1, Gv29-8, and FT-333, with the BUSCO protein metrics of QM6a, CBS1-1, CBS1-2, and FT-101

being higher respectively than those of P1, Gv29-8, and FT-333 (Table 1). BUSCOs are ideal for quantifying genome and proteome completeness, as the assumptions for these NUSCOs are evolutionarily sound (84). The genome sizes of QM6a, CBS1-1, and CBS1-2 are much smaller than those of FT-101, P1, Gv29-8, and FT-333. Thus, the genomes of *T. reesei* and its ancestor might not have experienced rapid gene loss. Instead, compared with the genomes of QM6a and the two CBS999.97 strains, the genomes of P1, Gv29-8, and FT-333 appear to have experienced both gene loss and extensive gene family expansions during evolution. In contrast, the genome of FT-101 only underwent extensive gene family expansions and not gene loss (Table 1). These findings represent a striking example of why TGS-based genomes are superior to NGS-based genomes for comparative and functional studies. Because the overall numbers of interspersed AT-rich blocks (≥500 bp) in QM6a, CBS1-1, and CBS1-2 are much lower (50% to 70%) than those in FT-101, P1, Gv29-8, and FT-333, we further infer that the larger genome sizes of these latter and their more extensive gene family expansions might have arisen from more profound mobilization and RIP of ancient retrotransposons in their ancestral genomes.

In this study, we have shown by comparative genomic and transcriptomic analyses that different *Trichoderma* species might utilize both conserved and diverse proteins or signaling pathways to mediate fungal-plant interactions (Fig. 2D). Our genome-wide annotation results also have revealed a comprehensive array of SM-BGCs and CAZ-BGCs in each of the four *Trichoderma* species we considered herein (Fig. 4). The majority of SM-BGCs, CAZ-BGCs, and effector-like SP proteins are poorly understood and, consequently, their corresponding products and physiological functions are largely unknown. Because a variety of genome mining methods and tools have been developed to guide characterization and activation of gene clusters (85–87), the nearly complete genome resources provided in this study will help discover and/or functionally characterize new SMs, CAZymes, and effector-like SP proteins, as well as their roles in and underlying evolutionary mechanisms of mycoparasitism, fungal-plant interactions and biocontrol activities.

**Conclusion.** Our study highlights that high-quality and nearly complete genome sequences are necessary tools for accurate comparative and functional genomic analyses. The data we have presented herein provide numerous insights into fundamental biology and industrial biotechnology.

## MATERIALS AND METHODS

**Fungal strains.** All *Trichoderma* wild-isolate strains described in this study are listed in Table S1. FT-101 and FT-333 were isolated by Ruey-Shyang Chen (National Chia-Yi University). CBS1-1, CBS1-2, Gv29-8, and P1 were provided by Monika Schmoll (Austrian Institute of Technology, Austria).

**Whole-genome DNA sequencing, RNA sequencing, and whole-genome gene prediction.** Isolation of genomic DNA and RNA, as well as PacBio single-molecular real-time (SMRT) genome sequencing and assembly, were carried out as described previously (21, 38, 88). Oxford Nanopore direct DNA library preparation and sequencing were performed at the Genomic Core Lab of the Institute of Molecular Biology, Academia Sinica. In brief, small fragments of purified genomic DNA were removed by using the Ampure cleanup kit (Beckman Coulter). High-quality DNA quantification was conducted using a Qubit fluorometer (Thermo Fisher Scientific). The average length of genomic DNA was scaled using Femto Pulse (Agilent). We used 1 $\mu$g of genomic DNA for library construction. The library contained three or four DNA samples barcoded using the EXP-NBD104 Rapid barcoding kit (Oxford Nanopore Technologies), and adaptors were added with the SQK-LSK109 ligation sequencing kit (Oxford Nanopore Technologies). The library was loaded onto a R9.4.1 Flow Cell system (FLO-MIN106) and processed for 48 h on the MinION platform. The sequencing process was controlled using the MinKNOW software (version 3.6.5; Oxford Nanopore Technologies). All experimental procedures were carried out according to the manufacturer's instructions. FAST5 data files were generated upon completion of sequencing. These files were converted into FASTQ files in Guppy (v3.6.0). All reads were split into separate FASTQ files based on their barcode by means of the qcat tool (v1.1.0), and then Canu (v2.1.1) (89) was used to perform whole-genome assembly on the corresponding FASTQ files for each sample. The parameter of corOutCoverage was set to 60 based on sequencing read depth. Finally, we used medaka (v1.2.0) to polish the assemblies with the raw reads. Because Canu generated expected numbers of contigs for each sample, and a comparison of the genome to wild-type QM6a indicated no broken contigs, we did not perform any manual finishing or validation. The completeness of genome assemblies was evaluated in BUSCO (v4.0.6) (90) against the database of fungi_odb10 (OrthoDB; https://www.orthodb.org). All other experimental procedures have been described previously in detail (38), including *PacBio* SMRT genome sequencing, NGS-

based RNA sequencing, as well as genome-wide gene prediction in Funannotate (91). BUSCO was also used to quantitatively measure the completeness of genome assemblies and gene predictions. We selected evolutionarily informed expectations of gene content based on the Ascomycota odb9 database from OrthoDB (https://www.orthodb.org). Genome or protein matrix scores >95% for model organisms are generally deemed complete reference genomes (92).

**Analysis of genome-wide synteny.** For comparative genome analyses and to identify duplicated regions, we identified orthologous gene pairs using the annotation results generated in Funannotate v1.8 (91). Alignments were performed using BLASTP with an expect value (E) $\leq$ 1e-20. CIRCOS (48) was used to display sequence similarity and conservation. The inner ribbon track of CIRCOS outputs is used to show synteny, whereas the exterior tracks quantify the degree of sequence conservation between the *T. reesei* CBS1-2 genome and those of the three other *Trichoderma* species.

**Comparative genomic analysis.** Funannotate (91) not only parses the protein-coding models from the annotation, but also identifies numbers and classes of carbohydrate-active enzymes (CAZymes), proteases, transcriptional factors, secreted proteins, polyketide synthases (PKSs), and nonribosomal peptide synthetases (NRPSs). To predict secreted proteins, we downloaded the SignalP 4.1 (93), TMHMM 2.0 (94), and big-PI Fungal Predictor (95) programs into "Funannotate" and then applied them with default settings. In this study, effector candidate proteins were defined as predicted secreted proteins (with a signal peptide present, but no trans-membrane domains or glycosylphosphatidylinositol anchors) having lengths of <300 amino acids. To assess if effector candidates presented similarity to known proteins, we performed BLASTP analysis (cutoff E-value $10^{-5}$) using the Swiss-Prot database (downloaded October 22, 2016).

CAZymes were reannotated using the dbCAN2 meta server (http://bcb.unl.edu/dbCAN2) (96). We have developed a software tool "IBM-CAZ" to predict potential CAZyme gene clusters (CAZ-GCs) with the following requirements: a potential CAZ-GC must contain $\geq$3 CAZyme genes or $\geq$2 CAZyme genes with $\geq$1 other specific signature genes (namely, transporters or transcription factors), and be present within $\leq$2 intergenic distances. Although the prediction requirements in IMB-CAZ are more stringent than those of dbCAN2 (96) and dbCAN-PUL (97), IBM-CAZ was able to identify all four previously identified CAZ-GCs in *Trichoderma reesei* QM6a, the ancestor of all currently used cellulase-producing mutants (32).

**Proteomics analysis of secreted proteins.** To identify proteins secreted by different *Trichoderma* strains, we germinated 1 mL of conidia ($OD_{600nm}$ = 0.3) and cultured it at 25°C in 250 mL potato dextrose broth (PDB) medium in shake flasks at 120 rpm for 24 h. The vegetative mycelia and conidia were removed from culture medium by filtering through a 0.22-$\mu$m filter (Merck Millipore, Darmstadt, Germany). The proteins in the culture filtrate (CF) were precipitated by adding ammonium sulfate (AS) to 80% saturation at 4°C. After centrifugation at 10,000 $\times$ *g* for 30 min at 4°C, the AS precipitates were recovered and then resuspended in 500 $\mu$L of 50 mM Tris-HCl (pH 7.5). Protein concentration was determined using Bradford reagent (Sigma-Aldrich).

For in-solution trypsin digestion, the target proteins (20 $\mu$g) were reconstituted in 100 mM Tris-HCl (pH 8.0) and 5% acetonitrile, mixed with 1/10 volume of 100 mM 1,4-dithiothreitol in 100 mM Tris-HCl (pH 8.0), and incubated at 37°C for 1 h. Then, 1/10 volume of 550 mM iodoacetamide in 100 mM Tris-HCl (pH 8.0) was added to the mixture, gently vortexed, and incubated at room temperature for 1 h. Promega Sequencing Grade Modified Trypsin was added to give a final substrate:trypsin ratio of 50:1. The digestion reaction was carried out overnight at 37°C and then the reaction was stopped by adding 5% formic acid to adjust the pH of the solution to below pH 6.0. We determined pH by placing 1-$\mu$L aliquots onto pH paper. The peptide digestion products were desalted using $C_{18}$ Zip-Tips (Millipore, ZTC 18M 096). The peptide solutions were dried down in a SpeedVac vacuum concentrator (Thermo Fisher Scientific) and stored at −20°C before undergoing mass spectrometric analysis.

Liquid chromatography-mass spectrometry (LC-MS-MS) was applied to analyze trypsin-digested peptides. LC-MS analysis was performed by Shu-Yu Lin (Institute of Biological Chemistry, Academia Sinica) using an EASY-nLC 1200 system linked to a Thermo Orbitrap Fusion Lumos mass spectrometer equipped with a Nanospray Flex ion source (Thermo Fisher Scientific) located at the Academia Sinica Common Mass Spectrometry Facilities (https://www.ibc.sinica.edu.tw/facilities/mass-spectrometry-facilities/). All related experimental procedures were described previously in detail (98). Proteomic data were searched against our whole-genome annotation data sets (see below) using the Mascot search engine (v.2.6.2; Matrix Science, Boston, MA, USA) in Proteome Discoverer (v 2.2.0.388; Thermo Fisher Scientific, Waltham, MA, USA). We used the search criterion trypsin digestion. The fixed modification was set as carbamidomethyl (C), and variable modifications were set as oxidation (M), acetylation (protein N-terminal) allowing up to two missed cleavages, and mass accuracy of 10 ppm for the parent ion and 0.6 Da for the fragment ions. The false discovery rate (FDR) was calculated with the Proteome Discoverer Percolator function, and identifications with an FDR > 1% were rejected.

**Data availability.** The complete genome sequences, annotation, and raw data sets have been deposited in the National Center for Biotechnology Information (https://www.ncbi.nlm.nih.gov/bioproject/) under accession number PRJNA700774 (Table S1). All study data are included in the article as well as in *SI* and *SD* appendixes. All genomic databases data and 15 supplemental data sets are also available at the following websites: https://github.com/tfwangasimb/Trichoderma-biocontrol/releases.

## SUPPLEMENTAL MATERIAL

Supplemental material is available online only.

**SUPPLEMENTAL FILE 1**, PDF file, 3.5 MB.

## ACKNOWLEDGMENTS

This work was supported by the Institute of Molecular Biology, Academia Sinica, Taiwan, Republic of China. We thank Shu-Yun Tung (IMB Genomics Core) for the NGS sequencing service, Kun-Hai Ye (IMB Bioinformatics Core) for statistical assistance and bioinformatics consultancy, John O'Brien for English editing, and Yu-Tang Huang (IMB Computer Room) for maintaining the computer workstation. This work was supported by Academia Sinica, Taipei, Taiwan, Republic of China.

H.F.N. and R.S.C. provided the two biocontrol strains, *T. asperellum* FT-101 and *T. virens* FT-333. W.C.L. and C.L.C. performed HMW gDNA isolation and Funannotate gene prediction. J.L.C. performed nanopore sequencing. W.C.L. and H.N.L. assembled genome sequences. T.C.L. performed comparative proteomic analyses. C.H.H. performed in-solution trypsin digestion. W.C.L. and T.F.W. performed comparative genomic analyses. T.F.W. conceived and designed the experiments. T.F.W. and W.C.L. wrote the original and revised manuscripts. All authors read and approved the manuscript.

We declare no conflicts of interest.

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
