## [Reviewer comments · Microbiology Spectrum]

Microbiology Spectrum

Complete Genome Sequences and Genome-Wide Characterization of *Trichoderma* Biocontrol Agents Provide New Insights into their Evolution and Variation in Genome Organization, Sexual Development and Fungal-Plant Interactions

Wan-Chen Li, Ting-Chan Lin, Chia-Ling Chen, Hou-Cheng Liu, Ju-Lan Chao, Hsin-Nan Lin, Hui-Fang Ni, Ruey-Shyang Chen, Cheng-Hsilin Hsieh, and Ting-Fang Wang

Corresponding Author(s): Ting-Fang Wang, Academia Sinica

Review Timeline:

Submission Date:	June 22, 2021
Editorial Decision:	August 2, 2021
Revision Received:	October 21, 2021
Editorial Decision:	October 26, 2021
Revision Received:	October 28, 2021
Accepted:	November 9, 2021

Editor: Christina Cuomo

Reviewer(s): The reviewers have opted to remain anonymous.

Transaction Report:

DOI: <https://doi.org/10.1128/Spectrum.00663-21>

August 2, 2021

Dr. Ting-Fang Wang
Academia Sinica
Institute of Molecular Biology
Taipei, Taipei 115
Taiwan

Re: Spectrum00663-21 (Complete Genome Sequences and Genome-Wide Characterization of Trichoderma Biocontrol Agents Provide New Insights into their Evolution and Variation in Fungal-Plant Interactions, Sexual Development and Genome Defense)

Dear Dr. Ting-Fang Wang:

Thank you for submitting your paper to Microbiology Spectrum. Two reviewers have provided detailed feedback that I would like you to address in a revision. Please carefully review and address the comments provided by the reviewers, including the marked up file provided by reviewer 1, in revising your paper. Also ensure that your genomic data (raw sequences, assemblies and annotations) are released at NCBI when you submit a revision; the provided BioProject PRJNA700774 is currently private.

When submitting the revised version of your paper, please provide (1) point-by-point responses to the issues raised by the reviewers as file type "Response to Reviewers," not in your cover letter, and (2) a PDF file that indicates the changes from the original submission (by highlighting or underlining the changes) as file type "Marked Up Manuscript - For Review Only". Please use this link to submit your revised manuscript - we strongly recommend that you submit your paper within the next 60 days or reach out to me. Detailed information on submitting your revised paper are below.

Link Not Available

Sincerely,

Christina Cuomo

Journals Department
Reviewer comments:

Reviewer #1 (Comments for the Author):

In this work, Li et al. continue the previous research line of the Ting-Fang Wang's group on the gapless genomics of Trichoderma. This group has previously reported a PacBio-based genome of Trichoderma reesei. In this work, they take the other three previously sequenced and best-annotated species (*T. virens*, *T. atroviride*, and *T. asperellum*) and present their third-generation genomics. As the authors also deposit the annotations, the work has its value and therefore is valuable for the community. First, however, the authors must correct embarrassing errors in the ms. The plant-pathogenic fungi-like stramenopiles are not fungi, and they must be corrected all over the manuscript. Trichoderma is not an arbuscular mycorrhizal fungus. The number of plant cell wall degrading enzymes in Trichoderma is not vast (but very limited, as we see from the broad fungal genomic survey). And many others. As the ms has no line numbering, I cannot list it here but marked a pdf file. Even though I support this ms, I regretfully inform the editor that most of the analyses performed here are redundant to previously published data, trivial, or incorrect. Furthermore, the authors ignore the currently available knowledge on Trichoderma

and fungal genomics and analyze their results as there are only these four species sequenced (mainly consider their previous work for the results even though they cite many studies in the introduction), however only for *T. atroviride* there are at least six genomes available in the public domain, while the total number of the genomes for this genus is > 50. Of course, third-generation sequencing presents an advantage, but it does not justify the ignorance of previous results.

The work completely lacks any evolutionary context (the tree in Figure 1 is no advantage compared to the first comparative genomic study on *Trichoderma* published ten years ago). *Trichoderma* belongs to the order Hypocreales, where also there are many (maybe > 100) genomes available. However, the comparisons are made to *Neurospora crassa* and eukaryotic fungi. It was a standard a decade ago.

Surprisingly, the work also contains some wet lab results, but they are equally inconclusive. First, the authors test whether their *Trichoderma* strains secrete antimicrobial water-soluble metabolites while cultivated on rich media. Yes, they do, it has been published numerous (very many) times for fungi and water molds (Oomycota), but it does not mean that *Trichoderma* will overgrow these (or other fungi) in direct confrontations. It has also been reported many times. [This section should either be deleted or described correctly, it is not an inhibition by *Trichoderma* but by *Trichoderma*'s WSM]/

The secretome study could be potentially interesting, but again, it was done in catabolite repressing conditions and repeated numerous previous reports with highly overlapping results.

The transcriptional profiling of the fruiting bodies formation is also strange to find in this article as it is only relevant to one species and not to the others. So, yes, this part could be presented as a focused stand-alone publication.

The same applies to the mitochondrial genome: all results are confirmatory to the previous knowledge.

The Discussion section is short and superficial as the analysis presents minor novelty. However, the dataset itself is valuable. Thus, this ms fits the scope of the *Microbiology Spectrum*.

Please note many comments in the results and supplements. Moreover, the terms "pathogen" and "biocontrol" are used incorrectly in this study. The authors should consider that *Trichoderma* is a mycoparasite and the fact that it attacks plant-pathogenic fungi does not allow to call them "pathogens" in the absence of plants. In this study, the term "pathogen" is more suitable for *Trichoderma*. Biocontrol is an agricultural practice. *Trichoderma* is just a fungus that interacts with other organisms what is used for biological control.

Reviewer #2 (Comments for the Author):

In this manuscript, the authors describe the generation of genome assemblies for four *Trichoderma* biocontrol agent strains using long-read sequencing. They perform detailed characterization of these genome assemblies, annotate them and attempt to gain insights into the evolution of biosynthetic gene clusters in this species. The manuscript is well written except in a few places, the methods are well described and the results are clear. There are a few places where the manuscript can be better organized and results need to be discussed/explained more. These results provide a number of interesting insights into these genomes and will make a very significant contribution to the field. I have a few comments/suggestions that might help them to improve the manuscript.

The result section 1 where they describe the activity against plant pathogens does not seem to fit into this manuscript. It does not connect with the rest of the manuscript. While the growth assays described are relevant, the fungal pathogen part is irrelevant and can be removed.

Throughout the manuscript, the authors compare already published 3 wild-type isolate genomes with 4 biocontrol agents published in this study. They always refer to it by the strain names. I would suggest they use "wild-type isolates" versus "biocontrol agents" when doing this comparison. It will be easy to read and grasp than just the names of the strains.

As a follow-up to the previous comment, it would be nice if the authors provide genome comparative maps for all the seven strains they discuss in this paper. Maybe show chromosome maps for all seven strains in one figure - similar to figure 3 but with a genome comparative view like a synteny analysis.

The authors state "Extensive chromosome rearrangements are likely the main determinants responsible for reproductive isolation of different *Trichoderma* species." While it is a possibility, it can very well be the other way around. Chromosome rearrangements could have occurred after the speciation. Unless authors have a compelling argument or analysis to support their conclusion, I would suggest they refrain from making such a conclusion.

In the next sentence, they say that disruption of synteny mainly occurs next to the AT-rich regions. What do they mean by mainly - what fraction of synteny breaks are associated with AT-rich regions and how many are not? Do these AT-rich regions include centromeres? If so, they may want to discuss this in more detail in light of a number of recent papers describing centromere mediated breaks in other fungi.

The authors define centromeres as "most prominent or longest AT-rich blocks". What do they mean by "most prominent"? Also, how long are the other AT-rich blocks in the genome? Are there AT-rich blocks that are longer than current centromeres but were not considered as centromeres? Or are there other prominent regions that could be centromeres but were considered so? If that is the case, I would suggest them to use the term "predicted centromeres" and not call them centromeres at this point. I would also suggest they include a figure showing specifically the organization of these regions, especially showing the low AT-

rich content. Are there any transposons in them?

The authors state "The higher numbers of AT-rich blocks in these four *Trichoderma* species might also account for (at least partly) their larger genome sizes". What is the basis behind this statement? Have they done any specific analysis to suggest that? What if some of the transposons were just RIPPed to make them AT-rich in these four species but not in other species. One would observe a higher number in that case as well.

The authors describe the presence of NUMTs very well but they do not mention anything about their functional part? For example, what part of mitochondrial DNA do these NUMTs belong to? Which ancestral genome are they referring to? Are these NUMT functional or have a gene sequence?

In the second half of the manuscript, where the authors describe sexual development, gene clusters, and BMGs, they talk about very specific genes such as ham5? In some cases, they fail to describe what these genes code for or their functional relevance? Also, why did the authors specifically focus on these specific genes? There is no justification provided. People who are not familiar with the field will find it hard to understand.

The last results section where authors describe transcriptome analysis in parts A and B can be three separate sections. A and B can be two separate sections with the last part where they describe the evolutionarily conserved genes being the third section. Also, this section can be discussed and explain in more depth with emphasis on their biocontrol activity.

The transcriptome analysis in the sexual development could be much better described as a heat map in a figure. It is too much information in the text otherwise and table 4 and the datasets are not very helpful in clearly understanding it. Besides, if the pattern is as clean as the authors describe in the text, it will be a nice figure to have in the manuscript.

In the same section, authors should clearly define what they call as VGGs, SDIGs, ESDGs, MSDGs, LSDGs, and CSDGs. I think the definition should come first and then the classification - not the other way around as they have presented now.

The second paragraph of the discussion needs references.

Figure 1A - With the very good genome assemblies that authors generated in this study, it would be great to have a phylogenetic tree based on the whole genome data. This will provide a more confident phylogeny analysis as well.

Figure 5 (and S9, S10, S11, S13) - Please do not use Watson-Crick nomenclature for the DNA strands. I would suggest using plus-minus strand nomenclature.

Tables 2, 3, and 4 could be moved to the supplementary information.

Staff Comments:

Preparing Revision Guidelines

For complete guidelines on revision requirements, please see the Instructions to Authors at [link to page]. **Submissions of a paper that does not conform to Microbiology Spectrum guidelines will delay acceptance of your manuscript.**

Please return the manuscript within 60 days; if you cannot complete the modification within this time period, please contact me. If you do not wish to modify the manuscript and prefer to submit it to another journal, please notify me of your decision immediately so that the manuscript may be formally withdrawn from consideration by Microbiology Spectrum.

If you would like to submit an image for consideration as the Featured Image for an issue, please contact Spectrum staff.

**Complete Genome Sequences and Genome-Wide Characterization of
Trichoderma Biocontrol Agents Provide New Insights into their Evolution
and Variation in Fungal-Plant Interactions, Sexual Development and
Genome Defense**

Wan-Chen Li¹, Ting-Chan Lin^{1,2}, Chia-Ling Chen¹, Hou-Cheng Liu¹, Hisn-Nan Lin¹, Ju-Lan
Chao¹, Cheng-Hsilin Hsieh¹, Hui-Fang Ni³, Ruey-Shyang Chen^{2,#}, Ting-Fang Wang^{1,2,#}

¹Institute of Molecular Biology, Academia Sinica, Taipei 115, Taiwan

²Department of Biochemical Science and Technology, National Chiayi University, Chiayi 600,
Taiwan

³Department of Plant Protection, Chiayi Agricultural Experiment Station, Council of
Agriculture, Chiayi 600, Taiwan

#Corresponding authors:

RSC, email: rschen@mail.ncyu.edu.tw

TFW, email: tfwang@gate.sinica.edu.tw

Abstract

Trichoderma spp. represent one of the most important fungal genera to mankind and in natural environments. They are efficient arbuscular ectomycorrhizal fungi and mycoparasites, and are prolific producers of plant-cell-wall degradation enzymes, effectors and natural products. Many species are used as industrial enzyme producers, biocontrol agents, biofertilizers, as well as model organisms for studying fungal-plant-pathogen interactions. Pursuing highly accurate, contiguous, and chromosome-level reference genomes has become a primary goal of fungal research communities. We previously applied third-generation sequence technology to determine and annotate the highest-quality genomic sequences yet of three *T. reesei* wild isolates: QM6a, CBS999.97(*MAT1-1*) and CBS999.97(*MAT1-2*). Here, we report the chromosome-level genomic sequences and whole-genome annotation datasets of four biocontrol strains (*T. virens* Gv29-8, *T. virens* FT-333, *T. asperellum* FT-101 and *T. atroviride* P1). Our results provide comprehensive categorization, correct positioning and evolution of both nuclear and mitochondrial genomes, including telomeres, centromeres, mating-type loci, transposons, nuclear-encoded mitochondrial sequences as well as many new secondary metabolism and carbohydrate-active enzyme gene clusters. Differential transposable element activity and the operation of repeated-induced point mutation (RIP) drive changes in genome sizes. We also identify evolutionarily conserved core genes during plant-fungal interactions and sexual development as well as variations potentially linked to key behavioral traits, including sex, genome defense, secondary metabolism and mycoparasitism. The genomic resources we provide herein significantly extend our knowledge not only of this economically important fungal genus, but also fungal evolution and basic biology in general.

Introduction

Trichoderma (*Hypocreales*, Ascomycota) species are highly successful colonizers of the rhizosphere (as mycoparasites or versatile symbionts) or wherever decaying plant material is available. They are easy to culture, grow rapidly, and often outgrow or even attach to other microbes encountered in their natural habitats. Although some *Trichoderma* species are of clinical significance, many more are important to humans and in natural processes. For example, the original isolate, *T. reesei* QM6a, and its cellulase-producing mutants are widely used to commercially produce enzymes that degrade plant cell walls and to generate therapeutic proteins (Peterson and Nevalainen 2012; Druzhinina and Kubicek 2016; Schmoll, et al. 2016). Several *Trichoderma* spp. are also used in agriculture as biocontrol agents (or biopesticides) against plant pathogens or as plant-growth-promoting biofertilizers, including *T. asperellum*, *T. atroviride*, *T. koningi*, *T. harzianum*, *T. rossicum* and *T. virens*. Accordingly, *Trichoderma* spp. have long been model organisms for studying molecular mechanisms underlying industrial enzyme production, secondary metabolite (SM) biosynthesis, effector proteins, mycoparasitism, fungal–plant–plant pathogen interactions and asexual sporulation (conidiation) in filamentous fungi (Montenecourt and Eveleigh 1977; Harman, et al. 2004; Druzhinina and Kubicek 2005; Schuster and Schmoll 2010; Steyaert, et al. 2010; Masunaka, et al. 2011; Mukherjee, et al. 2012; Peterson and Nevalainen 2012; Vinale, et al. 2012; Druzhinina and Kubicek 2016; Schmoll, et al. 2016; Yao, et al. 2016; Guzman-Guzman, et al. 2019; Ramirez-Valdespino, et al. 2019; Contreras-Cornejo, et al. 2020; Khan, et al. 2020). Recently, *T. reesei* has become an important emerging model filamentous fungus for studying ascomycete sexual mating (Seidl, et al. 2009; Chen, et al. 2012; Linke, et al. 2015), meiosis, postmeiotic mitosis and repeat-induced point (RIP) mutation (Chuang, et al. 2015; Li, et al. 2016; Li, et al. 2017; Dattenbock, et al. 2018; Li, et al. 2018; Li, et al. 2019; Li, et al. 2020; Li, et al. 2021). RIP is a fungus-specific genome-level defense mechanism against mobile elements or transposons. RIP

occurs premeiotically and targets duplicated sequences and causes frequent conversion of C:G base pairs to T:A within such duplicated sequences (Selker 1990; Aramayo and Selker 2013; Gladyshev 2017).

Thanks to the rapid development of next-generation sequencing (NGS) and genome-wide annotation technology, researchers have gained useful insights into the genomes of several *Trichoderma* species (Kubicek, et al. 2011; Druzhinina and Kubicek 2016; Schmoll, et al. 2016; Kubicek, et al. 2019), including a vast repertoire of plant-cell-wall degradation enzymes, as well as genes putatively involved in the biosynthesis of effectors and SMS (Martinez, et al. 2008; Mukherjee, et al. 2012; Atanasova, et al. 2013; Derntl, et al. 2017; Bulgari, et al. 2020). Subsequent comparative analyses of the NGS-based genomes of 12 *Trichoderma* species have provided a broad overview of their phylogenetic relationships (Kubicek, et al. 2019). For example, *T. asperellum* and *T. atroviride* are members of the Section (Sect.) *Trichoderma*. *T. reesei* and *T. virens* belong to Sect. *Longibrachiatus* or the *Harzianum/Virens* clade, respectively. Given that species from Sect. *Longibrachiatus* have the smallest genome sizes and gene inventories, it was proposed that their ancestor might have experienced rapid gene loss. In contrast, evolution of Sect. *Trichoderma* and the *Harzianum/Virens* clade was accompanied by significant gene gain. The highest numbers of genes gained encode ankyrins, HET domain proteins and transcription factors (Kubicek, et al. 2019). This hypothetical model needs to be further validated because the lengths of NGS reads are too short to exclude unidentified nucleotides and assembly errors. In a worst-case scenario, false-positive assemblies can result in incorrect assignments of gene gain or loss. NGS-based genome sequences also cannot reveal accurate information about genome synteny, diversity and evolution. Lastly, a smaller gene inventory might result from less intensive expansion of certain gene families rather than profound loss of distinct protein-encoding genes.

To obtain highly accurate and chromosome-level reference genomes to explore important biological questions, we previously applied third-generation sequencing (TGS) technology and the "Funannotate" gene-prediction tool (Palmer 2017) to determine and annotate the near-complete genome sequences of three *T. reesei* wild isolates, i.e., QM6a, CBS999.97(*MAT1-1*) and CBS999.97(*MAT1-2*) (Li, et al. 2017; Li, et al. 2020; Li, et al. 2021). Hereafter, CBS999.97(*MAT1-1*) and CBS999.97(*MAT1-2*) are referred to as CBS1-1 and CBS1-2, respectively. To further explore the biology and evolution of the genus *Trichoderma*, we have determined and annotated the near-complete genome sequences of four *Trichoderma* biocontrol agents. We demonstrate in this study that these highest-quality genome resources represent powerful tools for comparative multi-omics analyses of these economically important *Trichoderma* spp.

Results

Antagonism of *Trichoderma* spp. against different plant fungal pathogens

Four different plant pathogenic fungi were used in this study to determine the antagonistic capabilities of various *Trichoderma* strains (Supplemental Information SI (SI), Table S1). *Phellinus noxius* causes brown root rot disease in a wide range of tropical plants, mostly trees (Corner 1932). *Rhizoctonia solani*, *Sclerotium rolfsii* and *Phytophthora* spp. are plant pathogens that cause rice sheath blight (Sneh 1996), Southern blight (Purdy 1979) and *Phytophthora* diseases (Tyler 2002), respectively. The seven *Trichoderma* strains (Figure 1A) we analyzed exhibit variations in colony color and morphology on PDA plates (Figure 1B), as well as in PDB liquid media (Figure 1C). The colors of the vegetative mycelia in PDB media (Figure 1C) are compatible with their phylogenetic classification (Figure 1A). QM6a, CBS1-1, and CBS1-2 vegetative mycelia are orange-yellow or yellow, which is indicative of sorbicillinoid compounds (Abe, et al. 2001; Meng, et al. 2016). In contrast, vegetative mycelia

of the other species are red-brown (FT-333), dark brown (Gv29-8), or milky white (FT-101 and P1), respectively. All *Trichoderma* strains could inhibit vegetative growth of plant fungal pathogens, though their antagonistic capabilities varied slightly depending on the pathogen (SI, Table S2).

General properties of the near-complete genome sequences of seven *Trichoderma* strains

To address important questions relating to the basic biology of common biocontrol agents and respective biotechnological applications, we needed high-quality genome assemblies and genome-wide annotation datasets. Consequently, we applied TGS technology to establish high-quality and chromosome-level genome sequences of Gv29-8, FT-333, FT-101 and P1 (SI, Table S3), and also performed genome-wide annotation of protein-encoding genes (Table 1, SI, Tables S3-S5 and Supplemental Datasets (SD), DS1-DS8). The near-complete genomes of Gv29-8, FT-333, FT-101, and P1, like those of the three *T. reesei* strains (Li, et al. 2017; Li, et al. 2020; Li and Wang 2021; Liu, et al. 2021), harbor seven telomere-to-telomere nuclear chromosomes (Figure 2 and Figure 3). The three *T. reesei* genomes have the smallest genome sizes and encode the lowest overall numbers of protein-encoding genes and tRNA genes (Table 1 and SI, Table S3).

Gross chromosome rearrangements are the main contributors to speciation and genetic divergence

Conserved synteny (i.e., with ≥ 10 consecutive protein-encoding genes) between the near-complete *Trichoderma* genomes was revealed by CIRCOS plots (Krzywinski, et al. 2009) (Figure 2A). Genomes of the same species (i.e., Gv29-8 and FT-333; Figure 2B) or of the same section/clade (i.e., P1 and FT-101; Figure 2C) display a higher degree of chromosome synteny than when different *Trichoderma* species or sections are compared, respectively. According to

the model of chromosomal speciation (White 1978; Coghlan, et al. 2005), gross chromosome rearrangements frequently lead to reproductive isolation. Extensive chromosome rearrangements are likely the main determinants responsible for reproductive isolation of different *Trichoderma* species. Interestingly, disruption of synteny mainly occurs at long AT-rich blocks.

Telomeres, subtelomeres, centromeres and AT-rich blocks

Using our assembled telomere-to-telomere genomes, we could directly assess and compare telomeres, subtelomeres and AT-rich blocks at fine-scale resolution. Telomeric repeats are evolutionarily conserved among these seven *Trichoderma* genomes, i.e., TTAGGG at the 3'-termini and the reverse complement CCCTAA at the 5'-termini (Martinez, et al. 2008; Li, et al. 2017). **The subtelomeres of all seven *Trichoderma* genomes are hypervariable.** New subtelomeric variants can be created by two distinct mechanisms, i.e., alternative lengthening of telomeres (ALT) and break-induced replication (BIR). ALT can lengthen telomeres without utilizing telomerase. It is a homologous recombination (HR)-based process that involves copying telomeric DNA template (Lundblad and Blackburn 1993; Teng and Zakian 1999). During BIR, homologous templates from either the same chromosome or even a nonallelic region can be used for template replication and to establish new subtelomeres (Malkova, et al. 1996; Lydeard, et al. 2007). A typical example of BIR in *T. reesei* is the reciprocal exchange between the right arm terminus of ChII in CBS1-1 or the right arm terminus of ChIV in QM6a and CBS1-2, respectively (Chuang, et al. 2015; Li, et al. 2020; Li, et al. 2021). Another example is the subtelomeric fragment in the ChI left arm in FT-333 (indicated by a dark green line in **Figure 2B**). This subtelomeric fragment in Gv29-8 has relocated to the interior of ChIV, i.e., about a quarter of the chromosome's length from the terminus of its right arm (**Figure 2B**).

As in the filamentous fungal model organism *Neurospora crassa* (Borkovich, et al. 2004; Smith, et al. 2012; Kim, et al. 2014), the most prominent or longest AT-rich blocks in each *Trichoderma* chromosome are the centromeres, which are composed of degenerate transposons, mostly retrotransposons, and simple sequence repeats (Figure 3 and SI, Table S6). Overall numbers of interspersed AT-rich blocks (≥ 500 bp) are 2249 (QM6a) (Li, et al. 2017), 2259 (CBS1-1), 2250 (CBS1-2) (Li, et al. 2020; Li and Wang 2021), 3577 (Gv29-8), 3367 (FT-333), 4570 (FT-101) and 5510 (P1), respectively. These results are compatible with the average genomic GC contents of Gv29-8, FT-333, FT-101 and P1, all of which are lower than those of the three *T. reesei* strains (SI, Table S3). The higher numbers of AT-rich blocks in these four *Trichoderma* species might also account for (at least partly) their larger genome sizes. Thus, the ancestors of Gv29-8, FT-333, FT-101 and P1 might have undergone more profound transposon invasions and been subjected to greater RIP activity than those of *T. reesei*.

The overall numbers of authentic retrotransposons (i.e., which have not been extensively mutated or degenerated by RIP) in these seven nuclear genomes are 70 (QM6a) (Li, et al. 2017), 62 (CBS1-1), 62 (CBS1-2) (Li, et al. 2021; Li and Wang 2021), 93 (Gv29-8), 94 (FT-333), 92 (FT-101) and 78 (P1), respectively (SI, Table S3). Since CBS1-1 and CBS1-2 were derived from two ascospores of a CBS999.97 fruiting body, their genomes had undergone at least one round of RIP during sexual development (Seidl, et al. 2009). In contrast, QM6a, Gv29-8, FT-333, FT-101 and P1 have been propagated asexually since they were isolated. Therefore, the two CBS999.97 strains contain fewer authentic transposable elements.

Mitochondrial genomes (mitogenomes)

Our results also reveal the circular mitogenomes of all seven *Trichoderma* strains (SD, DS9). We reported recently that *T. reesei* mitochondria are inherited maternally (Li, et al. 2021), so the mitogenomes of CBS1-1 and CBS1-2 should be identical. Indeed, we found that their sequence differences were attributable to read errors in 18 polyhomonucleotide runs. The

lengths of the other six mitogenomes vary significantly, although both number and order of protein-encoding genes, tRNAs and rRNA are all well-conserved. We found that mobile genetic elements play key roles in shaping the mitogenomes of *Trichoderma* (*SD*, *DS11* and *SI*, Figures S1-S3).

Nuclear-encoded mitochondrial sequences (NUMTs)

NUMTs, first discovered in the mouse genome (du Buy and Riley 1967), have now been found in several other eukaryotes (Blanchard and Schmidt 1996; Ricchetti, et al. 1999; Yu and Gabriel 1999). NUMT integration has been implicated in increasing genetic diversity and facilitating genome evolution (Ricchetti, et al. 2004; Zhang, et al. 2020). NUMTs have yet to be explicitly reported among filamentous fungi. Three lines of evidence suggest that the ancestral genome of *T. virens* underwent NUMT integration. First, FT-333 and Gv29-8 each contains two almost identical NUMTs in the subtelomeric regions of the left arm of their second chromosome (indicated by a black line in *Figure 2B*). Their sequence coordinates and lengths in FT-333 are 211,858-212,161 (304 bp) and 212,485-212,654 (170 bp), whereas in Gv29-8 they are 115,319-115,631 (313 bp) and 115,955-116,124 (170 bp) (*SI*, *Figure S4*). Second, these two NUMTs are unlikely to represent mitochondrial DNA contamination arising from sequence assembly errors because the corresponding nucleotide sequences were observed in several long raw reads generated by the nanopore sequencer. Third, both of the NUMTs in FT-333 and Gv29-8 are located within an AT-rich block of length ~1500 bp (*SI*, *Figure S5*). AT-rich blocks often are relics of ancient retrotransposition events in the genomes of *Trichoderma* spp. It was reported previously that mammalian NUMTs tend to be inserted near retrotransposons and that the insertion sites often represent DNA sequences with high DNA “bendability” and lie immediately adjacent to AT-rich sequences (Tsuji, et al. 2012). Therefore,

filamentous fungi and mammalian cells may display an evolutionarily conserved origin for NUMTs (Tsuji, et al. 2012).

Highly divergent genomic contents of four *Trichoderma* species

Our high-quality assemblies and annotations of the near-complete genome sequences (*SD*, *DS2-DS8*) provided us with an unprecedented opportunity for comparative and functional genomic analyses. We selected four representative *Trichoderma* species—CBS1-2, Gv29-8, P1 and FT-101—that only share 7202 core protein-encoding genes (*Figure 4A*). Notably, these four genomes each encodes 2152, 2439, 3779 or 2555 species-specific genes, equivalent to one-fifth to one-third of their overall protein-encoding genes (*SI, Figure S6*). Gene ontology (GO) analyses further revealed that only 25-33% of the species-specific genes encode functionally annotated proteins (*SI, Figure S6*).

Mating type loci and sexual development genes

CBS1-1 and CBS1-2 are sexually competent. In contrast, like QM6a, Gv29-8, FT-333, FT-101 and P1 have been propagated asexually since they were isolated. We first confirmed that only the *ham5* gene in QM6a encodes a truncated protein (Linke, et al. 2015) (*SI, Figure S7*). We then surveyed ~160 gene orthologs in CBS1-2 and three other filamentous fungal model organisms (*Neurospora crassa*, *Sordaria macrospora*, *Saccharomyces cerevisiae*) (*SD, DS10*). All these gene orthologs have been implicated as being involved in or even essential to fungal sexual development (see reviews of (Chang, et al. 2012; Hunter 2015; Wang, et al. 2015; Druzhinina and Kubicek 2016; Schmoll, et al. 2016; Fischer and Glass 2019)). The seven *Trichoderma* genomes encode nearly all of the normal protein homologs deemed to play roles in sexual mating signaling systems (e.g., pheromones, light, cell communication and hyphal fusion), RIP, quelling, meiotic silencing by unpaired DNA (MSUD), meiotic DNA

recombination, chromosome individualization or condensation, sister chromatid cohesion, chromosome synapsis, as well as the formation of fruiting bodies (*SD*, *DS1*). P1 and Gv29-8, like QM6a, each possesses a *MAT1-2* locus with a normal *mat1-2-1* gene. FT-101, like CBS1-1, has a normal *MAT1-1* locus with three mating type genes: *mat1-1-1*, *mat1-1-2* and *mat1-1-3*. The *MAT1-1* locus of FT-333 only comprises *mat1-1-1* and *mat1-1-2*, but not *mat1-1-3* (**Figure 5**). The genomes of QM6a, CBS1-1 and CBS1-2 all possess an ortholog of *N. crassa male barren-3* (*mb-3*) (McCluskey, et al.). We were unable to identify or annotate this gene in the genomes of Gv29-8, FT-333, FT-101 or P1. Further investigations are needed to confirm whether Gv29-8, FT-333, FT-101 and P1 exhibit a male barren phenotype or if the *mb-3*-like gene in *T. reesei* is essential for male fertility. Interestingly, Gv29-8, FT-333 and P1, but not FT-101, also lack an ortholog of *sad3*, a gene essential for meiotic silencing by unpaired DNA (MUSD) and normal sexual development in *N. crassa* (Hammond, et al. 2011) (*SD*, *DS10*). MUSD, an RNA interference (RNAi)-related genome defense mechanism, occurs in prophase I of meiosis when unpaired DNA sequences are present and leads to the silencing of all homologous genes in the diploid ascus cell (Hammond, et al. 2011).

Carbohydrate-active enzyme (CAZyme) and CAZyme gene cluster (CAZ-GC)

Compared to the three *T. reesei* genomes, those of Gv29-8, FT-333, FT-101 and P1 encode more auxiliary activity (AA) proteins, glycoside hydrolases (GH), carbohydrate esterases (CE) and glycoside hydrolases (GH). In contrast, all three *T. reesei* genomes possess more glycosyltransferase (GT) genes than those of Gv29-8, FT-333, FT-101 and P1 (**Table 1**). Further investigations will be needed to elucidate this unique property of *T. reesei*. There is considerable evidence showing that CAZymes cooperate with other CAZymes and signature proteins (e.g., transporters and transcription factors), and the respective genes tend to form physically linked CAZyme gene clusters (CAZ-GCs) in polysaccharide utilization loci (PUL)

(Terrapon, et al. 2015). To identify potential CAZ-GCs in these strains, we have developed a high-stringency predictive software (*SI*, “additional Materials and Methods”) which identified 31, 31, 29, 35, 26, 31 and 33 CAZ-GCs in QM6a, CBS1-1, CBS1-2, Gv29-8, FT-333, FT-101 and P1, respectively (**Figure 3, *SI*, Table S6 and *SD*, DS11**). These CAZ-GCs were also named according to their chromosomal location, e.g., CAZ-GC 3.2 indicates the second CAZ-GC from the left arm of ChIII. These results provide a comprehensive basis for further exploring CAZymes and PUL in *Trichoderma* spp. We identified 9, 17, 13, and 8 species-specific CAZ-GCs in CBS1-2, Gv29-8, P1 and FT-101, respectively. Notably, consistent with their phylogenetic relationships, *T. reesei* and *T. virens* share 14 common CAZ-GCs, whereas *T. atroviride* and *T. asperellum* share 18 common CAZ-GCs (***SI*, Figure S8A**).

Transcription factors (TFs)

In terms of overall numbers of TF genes, the ranking is FT-101 > P1 > Gv29-8 > CBS1-2 > CBS1-1 > QM6a > FT-333 (**Table 1**). The main differences are due to specific gain or loss of two transcription factor subfamilies, i.e., the fungal Zn(2)-Cys(6) binuclear cluster domain subfamily (InterPro identity: 111138) and the fungal-specific TF domain family (InterPro identity: 007219). Compared to the genomes of Gv29-8, FT-333, FT-101 and P1, the three *T. reesei* genomes possess a species-specific PAS-fold protein, but lack a helix-turn-helix TF (***SI*, Table S4**). The physiological functions of these unique TFs need to be further explored. It is important to note that gene numbers of other TF families are nearly identical among the seven genomes. Thus, these TFs constitute the "core" transcriptional regulatory circuitry of the genus *Trichoderma*.

Predicted signal peptide (SP) proteins

Many secretory proteins in fungi are known to be involved in fungal interactions with a variety of organisms, including bacteria, animals, plants and other fungi (Kubicek, et al. 2011;

Rovenich, et al. 2014; Vleeshouwers and Oliver 2015). SPs are short amino acid sequences in the amino termini of many newly synthesized proteins that target to membranes or membrane-embedded export machines. The SignalP server has been widely used to predict the presence and location of SP cleavage sites in amino acid sequences of SP proteins (Tamura, et al. 2011; Nielsen 2017). The three *T. reesei* genomes each possess 840-870 SP proteins. In contrast, those of Gv29-8, FT-333, FT-101 and P1 encode 1072, 995, 1050 and 1041 SP proteins, respectively. Gene ontology (GO) indicates that all seven *Trichoderma* genomes are highly enriched with SP proteins having catalytic, oxidoreductive or hydrolytic activities (Table 1). It is important to note that the SP proteins revealed by the SignalP server are not necessarily cleaved or cleavable *in vivo*. Further experimental investigations are needed to validate if they are indeed secreted proteins.

Secondary metabolite biosynthetic genes and gene clusters (SM-BGCs)

Relative to the three *T. reesei* nuclear genomes, those of P1, FT-101, Gv29-8 and FT-333 have undergone expansions in almost all SM-BGC subfamilies, including polyketide synthases (PKSs), non-ribosomal peptide synthases (NRPSs) and cytochromes P450 (CYP450s) (Table 1). Using the antiSMASH (Antibiotics and Secondary Metabolite Analysis Shell) software tool (Medema, et al. 2011), we have annotated a variety of SM-BGCs in the seven genomes: 32 in QM6a, 32 in CBS1-1, 32 in CBS1-2, 59 in Gv29-8, 57 in FT-333, 52 in FT-101 and 46 in P1 (Figure 3 and SI, Table S5). These SM-BGCs were also named according to their chromosomal location, for example, SM-BGC 2.3 indicates the third SM-BGC from the left arm of ChII. The four different *Trichoderma* species only share 14 common SM-BGCs. We identified 4, 21, 8, and 9 species-specific SM-BGCs in CBS1-2, Gv29-8, P1 and FT-101, respectively. Notably, consistent with their phylogenetic relationships, *T. reesei* and *T. virens* share 27 common SM-BGCs, whereas *T. atroviride* and *T. asperellum* share 37 common SM-BGCs (SI, Figure S8B).

To address the intriguing question of how SM-BGCs form and are regulated, we compared the biosynthetic genes of the five best-characterized SM-BGCs (see reviews of (Mukherjee, et al. 2012; Schmoll, et al. 2016)) to explore their commonalities and differences (Table 2). First, BGCs for siderophore (SID), ferrichrome (FRC) and conidial green pigment (CGP) are found in all four *Trichoderma* species we examined here. SID-BGCs and FRC-BGCs exhibit gene order conservation and contain none or only one AT-rich block (>500 bp) (SI, Figure S9 and S10). There are only three evolutionarily conserved genes in CGP-BGCs. Their neighboring sequences are quite variable and contain more AT-rich blocks. The neighboring genes of CGP-BGC in P1 are more conserved with those in FT-101, but they are completely different from those in Gv29-8 and CBS1-2 (SI, Figure S11). Second, the sorbicillinoid (SOR)-BGC is *T. reesei*-specific, as only *T. reesei* can secrete yellow sorbicillinoid compounds (Abe, et al. 2001). The 5' and 3' termini of SOR-BGC are physically linked to *usk1* (Beier, et al. 2020) and a well-characterized CAZ-GC harboring three CAZyme genes: *axe1* (acetyl xylan esterase), *cip1* (a CBM-containing auxiliary factor) and *cel61a* (previously named endoglucanase IV) (Martinez, et al. 2008) (SI, Figure S12). The entire chromosomal region (i.e., *usk1*-SOR-BGC-*axe1*-*cip1*-*cel61a*) contains 14 protein-encoding genes, but no AT-rich blocks. Although Gv29-8, P1 and FT-101 each possesses 8-10 orthologs of those 14 protein-encoding genes, their orthologs are scattered across at least five different chromosomes (Table 2 and SI, Figure S12). Third, FT-101 possesses a terpene cyclase (TrA_010949) that is highly similar in amino acid sequence to the protein product of the *T. brevicompactum* *tri5* trichothecene synthase gene (Schmoll, et al. 2016). This putative *tri5* gene and seven novel genes together constitute a potential trichothecene (TRI) or TRI-like BGC in FT-101 (i.e., SM-BGC-7.3) (SI, Figure S13). We were unable to identify or annotate this putative *tri5* gene in P1, CBS1-2 or Gv29-8. It is also noteworthy that the protein products of these seven novel genes in FT-101 are different from the biosynthetic proteins encoded by

the canonical TRI-BGCs in *T. arundinaceum* and *T. brevicompactum* (Cardoza, et al. 2011; Mousa and Raizada 2015). Interestingly, six orthologs of these seven novel genes also cluster together in P1, whereas CBS1-2 and Gv29-8 each possesses three and five orthologs of these seven novel genes, respectively (Table 2). Fourth, *T. virens* has two species-specific BGCs: viridin (VIR)-BGC and gliotoxin (GTX)-BGC. These two SM-BGCs produce a group of furano-steroidal antibiotics and GTX or GTX-like compounds to compete with and restrict the growth of plant pathogenic fungi, respectively (Bansal, et al. 2018; Bulgari, et al. 2020). CBS1-2, P1 and FT-101 possess none or only two orthologs of the VIR biosynthetic genes, respectively (Table 2). Both GTX-BGCs in Gv29-8 and FT-333 contain 13 *gli* genes and are located at the far-right subtelomeres of their first chromosomes. Ten long AT-rich blocks separate these 13 *gli* genes. The order of these long AT-rich blocks is conserved in FT-333 and Gv29-8, but they exhibit highly variable lengths and sequences (Figure 6A). Although *T. reesei* does not produce GTX during vegetative growth (Bulgari, et al. 2020), QM6a and both CBS999.97 strains each possesses a gene cluster with only six and seven *gli* orthologs, respectively, i.e., SM-BGC-6.4 in CBS1-2 (Figure 6A and Table 2). *gliH*, an essential GTX biosynthetic gene, does not exist in any of the three *T. reesei* genomes. QM6a also lacks *gliC*, which encodes a CPY450 oxidoreductase that catalyzes the formation of C-S bonds. These C-S bonds constitute the internal disulfide bridge of GTX and other exopolysaccharide-type fungal toxins (Scharf, et al. 2012). The order of the *gli* genes in *T. reesei* is different from that in Gv29-8. There are no AT-rich blocks between the *gli* orthologs in *T. reesei* (Figure 6B). These results suggest that the ancestral genomes of *T. virens* and *T. reesei* might have undergone multistep processes to acquire and/or reorganize their GTX or GTX-like BGCs. Further investigations will be needed to address the physiological roles of the incomplete GTX-BGCs in *T. reesei*.

***T. reesei* Sor7 and *T. virens* GliT are secreted proteins**

Most of our knowledge about the molecular mechanisms of GTX biosynthesis has come from studies of the GTX-BGC in *Aspergillus fumigatus*, an opportunistic fungal pathogen. *A. fumigatus* GliT is an intracellular FAD-dependent dithiol oxidase that converts the reduced GTX (i.e., dithio-GTX) to GTX, thus conferring host self-resistance toward highly reactive dithio-GTX and tolerance to exogenous GTX (Scharf, et al. 2010; Schrettl, et al. 2010). We found that the GliT proteins in FT-333 host signal peptide (SP), whereas that of *A. fumigatus* does not (SI, Figure S14). The GliC proteins in FT-333 and Gv29-8, like that of *A. fumigatus*, are predicted to possess signal peptides. In contrast, that of CBS1-2 does not (SI, Figure S15). Accordingly, the biosynthetic and regulatory mechanisms of GTX- or GTX-like BGCs in *T. virens* and *T. reesei* likely differ from those of *A. fumigatus*. Our annotation results also reveal that the Sor7 FAD-linked oxireductase of *T. reesei* hosts an SP (SI, Figure S16). Since the SP-hosting proteins predicted by the SignalP server are not necessarily cleaved or cleavable *in vivo*, we applied a proteomics approach to analyze them in the culture filtrate (CFs) of CBS1-2, FT-333, Gv29-8, FT-101 and P1, respectively, including by means of ammonium sulfate precipitation, in-solution trypsin digestion and LC-MS-MS analysis (SI, Materials and Methods). Although most of the secreted proteins we identified in CFs are functionally unannotated, our results confirm that *T. virens* GliT and *T. reesei* Sor7 are indeed secreted proteins (SD, DS12 and SI, Table S9). Further investigations will be needed to reveal the functional roles of these two FAD-linked oxireductases in regulating biosynthesis and the functions of GTX and sorbicillinoids, respectively.

Our proteomics approach also identified two fungal secreted proteins, i.e., *T. virens* Sm1 (small protein 1) and *T. asperellum* KatG2 catalase-peroxidase 2. *T. virens* Sm1 is a secreted elicitor that induces plant defense responses and systemic resistance by triggering production of reactive oxygen species. Sm1 lacks toxic activity against plants and microbes (Djonovic, et al. 2006). The functional roles of KatG2 in *T. asperellum* remain unclear. KatG2

was originally identified as an extracellular protein in several phytopathogenic fungi (Zamocky, et al. 2012). In *Fusarium graminearum*, KatG2 is exclusively located on the cell wall of invading hyphal cells and contributes to its pathogenicity by alleviating oxidative stress in the vicinity of invasion hyphae (Guo, et al. 2019).

Genome-wide transcriptomic analyses reveal evolutionarily-conserved and diverse genes contributing to sexual development and fungal-plant interactions

(A) Fungal-plant interactions

Trichoderma spp. can colonize plant roots, both externally and internally. Induction of plant defense via fungal-plant interactions is considered one of the most important mechanisms of *Trichoderma*-mediated biological control. Michael Kolomiets and colleagues performed RNA sequencing (RNA-seq) analysis on the roots of maize seedlings grown in hydroponic conditions and treated with *T. virens* Gv29-8 at 6 h and 30 h (Malinich, et al. 2019; Wang, et al. 2020). Notably, the time-points represented fungal-plant recognition at 6 h and advanced fungal colonization at 30 h. Gv29-8 undergoes global repression of transcription upon recognition of maize roots and then induces expression of a broad spectrum of genes during root colonization (Malinich, et al. 2019). Using the complete genome sequence of Gv29-8 as a reference, we reanalyzed the previously acquired transcriptomic datasets (Malinich, et al. 2019; Wang, et al. 2020) to reveal 365 and 2082 *T. virens* Gv29-8 genes that are transcriptionally upregulated (fold-change (FC) ≥ 3 and $P < 0.05$) at the 6-h and 30-h time-points, respectively (Table 3 and SD, DS13 and DS14). The protein products of 365 putative fungal-plant recognition genes (FPRGs) include 42 predicted SP proteins, 13 proteases, 14 CAZymes and 75 membrane proteins. Three SM-BGCs and one CAZ-GC (1.4) were specifically upregulated at 6 h. The protein products of 2082 putative advanced fungal colonization genes (AFCGs) include 349 predicted SP proteins, 100 proteases, 158 CAZymes, 433 membrane proteins and 433 TFs. Almost 80% of SM-BGCs and CAZ-GCs in *T. virens* have at least one gene that was

transcriptionally upregulated at 30-h. Notably, most genes in GTX-BGC-1.1 and CPG-BGC-10.1 are AFCGs, whereas all those in VIR-BGC-5.5 and SID-BGC-6.4 are not (Table 3). Further investigations will be needed to reveal whether and why CPG, GTX or related SMS have non-antibiotic functions during advanced fungal colonization. Moreover, we found by genome-wide BLASTp searches that 190 (52%) FPRGs and 1037 (50%) AFCGs are evolutionarily conserved in the genomes of CBS1-2, FT-101, P1 and FT-333, respectively (Figure 4B, SI, Table S7 and SD, DS15), prompting us to infer that different *Trichoderma* species might utilize both conserved and diverse proteins or signaling pathways to mediate fungal-plant interactions.

(B) Sexual development

To enrich for RNA expressed during vegetative growth and at different developmental stages, we previously developed a protocol for relatively synchronous sexual development under conditions of 25 °C and a 12-h light/dark cycle (Chen, et al. 2012; Li and Wang 2021). In brief, sexual crossing was initiated by mixing the conidia from a QM6a(*MATI-2*) haploid male strain to the mycelium of a CBS1-1 haploid female strain at day 0 (D0) (SI, Figure S17A). We found that this method ensured the entire sexual development process (e.g., mating, stroma, pericethium and ascospores) occurred in a reasonably synchronous manner. The primordial white aggregates emerged at D1, i.e., one day after transferring male conidia onto female mycelia. Protoperithecia-like structures, which were comprised of central ascogenous hyphae and the developing walls, appeared at D2. The wall of fruiting bodies could be differentiated by their tan coloration. The centrum papachyma and young asci differentiated in parallel in perithecia at D4, and the peridium (perithecial wall) also gradually formed. A rosette of undifferentiated asci and an ostiole (an open pore of a perithecium at the surface of stroma) appeared in perithecia at D6. The ostiole then opened up and mature ascospores were ejected from the perithecium at D8 and thereafter (SI, Figure S17B). The experimental samples were

harvested every 24 h for eight days (D1-D8), frozen in liquid nitrogen, and stored at -80 °C. NGS-based RNA sequencing datasets were then used to define five groups of developmentally-regulated genes based on their expression profiles ($FC \geq 3$ and $P < 0.05$) during a developmental time-series, including 1217 vegetative growth genes (VGGs), 767 sexual development (SD) initiation genes (SDIGs), 1219 early SD genes (ESDGs), 403 middle SD genes (MSDGs), 403 late SD genes (LSDGs) and 851 completed SD genes (CSDGs) (Table 4 and SD, DS16-DS21).

VGGs were transcriptionally upregulated only in the vegetative mycelia (D0) of QM6a and CBS1-1, including 63 CAZymes, the cellulase and hemicellulose TF genes *xyr1* and *ace2*, and the cellulose utilization TF gene *vib1*. Notably, most genes (10/16) in SOR-BGC were upregulated only during vegetative growth but not after sexual mating (Table 4). These results are consistent with constitutive hyperproduction of plant-cell-wall degradation enzymes and sorbicillinoid compounds during vegetative growth (Karlsson, et al. 2001; Atanasova, et al. 2013; Meng, et al. 2016; Derntl, et al. 2017). SDIGs were upregulated only from D1 to D3, and their protein products are responsible for DNA replication and repair, cell cycle, colony development, hyphal communication and fusion, the circadian clock and light responses. ESDGs were transcriptionally upregulated from D1 to D8. The protein products of ESDGs are involved in autophagy, hyphal fusion, pheromone-regulated plasma membrane fusion, RNA interference, male fertility, and female sexual development. Interestingly, autophagic proteins are indispensable for the formation of round protoperithecia in the filamentous ascomycete *Sordaria macrospora* (Teichert, et al. 2020). MSDGs were transcriptionally induced after D4, and their protein products are essential components in sexual mating, pheromones, RNAi, MSUD and vacuolar proteolysis (e.g., Pep4). The proteinase Pep4 is a vacuolar enzyme that initiates processing and activation of vacuolar precursors during starvation and meiosis. *S. cerevisiae* diploids lacking Pep4 do not undergo meiosis or sporulation (Zubenko and Jones

1981; Lai, et al. 2011). LSDGs began to be transcriptionally upregulated after D6 and almost all respective protein products have yet to be functionally annotated. Many CSDGs are also VGGs, consistent with our cytological observations that mature ascospores were ejected from the perithecium and then germinated at D8 and thereafter (SI, Figure S17C).

A key finding of our transcriptomic dataset is that VGGs, SDIGs, ESDGs, MSDGs and LSDGs each contain a variety of CAZymes and SM biosynthesis proteins (Table 4). For example, transcription of one CAZ-GC and four SM-BGCs (e.g., the putative GTX-BGC) are specifically upregulated during ESD of *T. reesei*. Next, we isolated crude SMs from vegetative mycelia of QM6a, CBS1-1, CBS1-2, FT-333, Gv29-8, FT-101 and P1, as well as those from the stromata generated by sexual crossing of QM6a and CBS1-1 at different development stages (D2-D6), and then subjected them to thin layer chromatography (TLC) followed by UV visualization. We used GTX (represented as compound **1** on the TLC image in Figure 6C) as a standard for TLC. Only FT-333 and Gv29-8 produced GTX during vegetative growth. The yellow-colored SM (compound **2**) generated by the vegetative mycelia of QM6a and two CBS999.97 stains are sorbicillinoids or their reaction intermediates (Abe, et al. 2001; Meng, et al. 2016; Derntl, et al. 2017). Notably, three new SMs (compound **3-5**) appeared in the D2-D6 stromata. Further investigations will be needed to reveal their chemical structures and biosynthetic mechanisms and physiological functions during sexual development.

Next, we also performed genome-wide BLASTp searches to identify evolutionarily conserved genes in the genomes of Gv29-8, FT-333, FT-101 and P1 (Figure 4C, SI, Table S8 and SD, DS22). These results not only reveal the evolutionarily conserved or core genes involved in SD of different *Trichoderma* species, but also the *T. reesei*-specific SD genes or the SD genes absent from the four biocontrol agents. For example, *sad3* is a MSDG in *T. reesei*. As described above (SD, DS10), the genome of FT-101, but not those of Gv29-8 and P1, has lost this essential sexual development and genome defense gene. Our results also reveal that

an SDIG of CBS1-2 is the ortholog of *mod-A*. The *mod-A* gene of the filamentous fungal model organism *Podospora anserina* encodes modin, an SH3-like protein involved in controlling the development of female organs (Barreau, et al. 1998). An ortholog of *mod-A* exists in the genome of Gv29-8, but it has been lost or has not been annotated in those of P1 and FT-101 (*SD, DS22*).

Discussion

The growing commercial importance of the fungal genus *Trichoderma* necessitates a better understanding of its biology and evolution. The genome resources we have presented in this study, which are the highest quality yet achieved, will help in mining *Trichoderma* genomes for candidate genes that can be manipulated for industrial and agricultural applications, as well as aid in research into fundamental biology.

The results of our chromosome synteny analyses have revealed that extensive gross genomic rearrangements are the main contributors to genomic divergence and speciation in the genus of *Trichoderma*. Disruption of synteny often occurs at or within long AT-rich blocks, which are relics of ancient retrotransposition and RIP events. One possibility is that mobilization of ancient retrotransposons might induce chromosome rearrangements, eventually leading to genome diversification within a species (microevolution) and between species (macroevolution). Alternatively, the AT-rich blocks might represent common fragile sites that are susceptible to breakages and rearrangements. In this regard, retrotransposition and RIP might act as a critical driver of genome reorganization and species diversification.

Locally, the order of biosynthetic genes, AT-rich blocks and retrotransposons in SM-BGCs provide detailed insights into the mechanisms of genome evolution. Our data indicate that evolutionarily conserved SM-BGCs often (but not always) lack or only contain few longer AT-rich blocks (≥ 500 bp). In contrast, the GTX-BGCs in FT-333 and Gv29-8 have ten such blocks. This unique property of GTX-BGCs had not been observed before because NGS

sequence reads are too short to assemble chromosomal regions of low sequence complexity (Bulgari, et al. 2020). We suggest that GTX-BGCs represent versatile systems for future experimental evolution studies to reveal the functional impacts of AT-rich blocks for the expression and regulation of GTX biosynthesis. Also noteworthy is that orthologs of GTX biosynthetic genes are randomly allocated across the genomes of *T. atroviride* and *T. asperellum*. The gene order of the putative GTX-BGC in *T. reesei* also differs from that of the authentic GTX-BGCs in *T. virens* (SI, Figure S12). We conclude that diverse and/or multistep mechanisms were involved in the formation of different SM-BGCs during evolution (Nutzmann, et al. 2018), such as lateral gene transfer, unequal crossing over or transposon-mediated gene transfer. Lateral gene transfer (also called horizontal gene transfer) represents the acquisition of genetic material from another organism (e.g., bacteria), whereas unequal crossing over involves the exchange of sequences between chromosomes. Since the GTX-BGCs of both FT-333 and Gv29-8 are located in the subtelomeres of their first chromosomes, it would be interesting to determine if BIR (Malkova, et al. 1996; Lydeard, et al. 2007) or other molecular mechanisms might play critical roles in generating or maintaining this *T. virens*-specific gene cluster.

High-quality and near complete genome sequences ensure accurate and comprehensive genome annotation. Based on NGS-based genome sequences, it was reported previously that *T. reesei* and its ancestor might have undergone rapid gene loss relative to other *Trichoderma* spp. (Kubicek, et al. 2019). In contrast, our data indicate that QM6a, CBS1-1, CBS1-2 and FT-101 encode more protein-encoding genes than the genomes of P1, Gv29-8 and FT-333, with the BUSCO genome and protein metrics of QM6a, CBS1-1, CBS1-2 and FT-101 being higher respectively than those of P1, Gv29-8 and FT-333 (Table 2). It is noteworthy that BUSCOs (i.e., Benchmarking Universal Single-Copy Orthologs) are ideal for quantifying genome completeness, as the assumptions for these single-copy genes are evolutionarily sound

(Waterhouse, et al. 2013). Notably, the genome sizes of QM6a, CBS1-1 and CBS1-2 are much smaller than those of FT-101, P1, Gv29-8 and FT-333. Thus, the genomes of *T. reesei* and its ancestor might not have experienced rapid gene loss (Kubicek, et al. 2019). Instead, compared to the genomes of QM6a and the two CBS999.97 strains, the genomes of P1, Gv29-8 and FT-333 appear to have experienced both gene loss and extensive gene family expansions during evolution. In contrast, the genome of FT-101 only underwent extensive gene family expansions and not gene loss (Table 2). These findings represent another example of why TGS-based genome sequences are superior to NGS-based genomes for comparative and functional studies. Since the overall numbers of interspersed AT-rich blocks (≥ 500 bp) in QM6a, CBS1-1 and CBS1-2 are much lower (50-70%) than those in FT-101, P1, Gv29-8 and FT-333, we infer that the larger genome sizes of these latter and their more extensive gene family expansions might have arisen from more profound mobilization and RIP of ancient retrotransposons in their ancestral genomes.

Filamentous fungi are known to harbor the genetic capacity for an arsenal of useful natural compounds. The genes required for biosynthesis of SMs are often clustered. Our comparative genomic analyses have revealed a comprehensive array of SM-BGCs in each of the four *Trichoderma* species we considered herein. The majority of SM-BGCs are poorly understood and, consequently, their corresponding products are largely unknown. Since a variety of genome mining methods and tools have been developed to guide characterization and activation of these gene clusters (Lim, et al. 2012; Ziemert, et al. 2016; Zhang, et al. 2019), our new data will help discover new secondary metabolites and their roles in mycoparasitism, fungal–plant–plant pathogen interactions, sexual reproduction, and the evolutionary mechanisms underlying SM-BGCs.

Conclusion

Our study highlights that high-quality and near-complete genome sequences are necessary tools for accurate comparative and functional genomic analyses. The data we have presented herein provide numerous insights into fundamental biology and industrial biotechnology.

Materials and Methods

All *Trichoderma* wild-isolate strains described in this study are listed in *SI, Table S1*. *T. asperellum* FT-101 and *T. virens* FT-333 (hereafter, simply referred to as FT-101 and FT-333, respectively) were isolated by Ruey-Shyang Chen (National Chia-Yi University). *T. reesei* CBS999.97(*MAT1-1*) (CBS1-1 in short), *T. reesei* CBS999.97(*MAT1-2*) (CBS1-2 in short), *T. virens* Gv29-8 (hereafter Gv29-8) and *T. atroviride* P1 (hereafter P1) were provided by Monika Schmoll (Austrian Institute of Technology, Austria). All plant fungal pathogens (i.e., *Phellinus noxius*, *Rhizoctonia solani*, *Sclerotium rolfsii* and *Phytophthora* spp.) used in this study were isolated by Hui-Fang Ni (Chiayi Agricultural Experiment Station, Council of Agriculture). Additional “Materials and Methods” are provided in the supplemental information (*SI*), including detailed information on fungal growth inhibition assays, whole-genome DNA sequencing, RNA sequencing and whole-genome gene prediction, analysis of genome-wide synteny and comparative genomic analysis.

Competing interests

The authors declare that they have no competing interests.

Supplemental Information

The supplemental information (*SI*) file includes supplemental "Materials and Methods", 9 supplemental tables and 17 supplemental figures. [Twenty-two supplemental datasets \(DS1-DS22\)](https://github.com/tfwangasimb/Supplemental-datasets/releases/tag/20210611) in Excel format are also provided separately online (<https://github.com/tfwangasimb/Supplemental-datasets/releases/tag/20210611>).

Table S1. List of all *Trichoderma* strains analyzed in this study

Table S2. Biocontrol activities of tested *Trichoderma* strains against four different plant fungal pathogens using a modified cellophane method

Table S3. Summary of the properties of seven near-complete *Trichoderma* genome sequences

Table S4. Numbers of annotated transcription factors encoded by the near-complete genomes of the seven *Trichoderma* spp.

Table S5. Numbers of SM-BGCs and CAZ-BGCs revealed by antiSMSH

Table S6. The location of centromeres in CBS1-2, Gv29-8, FT-101 and P1.

Table S7. The number of evolutionarily conserved fungal-plant interaction genes in different *Trichoderma* species

Table S8. The number of evolutionarily conserved vegetative growth and sexual development genes in different *Trichoderma* species

Table S9. Proteomic identification of proteins in culture filtrates

Fig. S1. Circular maps of the complete mitogenomes of QM6a and CBS1-2.

Fig. S2. Circular maps of the complete mitogenomes of Gv29-8 and FT-333.

Fig. S3. Circular maps of the complete mitogenomes of CBS1-2, Gv29-8, P1 and FT-333.

Fig. S4. Pairwise sequence alignments of the nucleotide sequences within and around the two NUMTs (in red) in the second chromosomes of Gv29-8 and FT-333.

Fig. S5. The two NUMTs located in a long AT-rich block.

Fig. S6. The numbers of species-specific genes that were functionally annotated by the Gene Ontology (GO) annotation software.

Fig. S7. QM6a encodes a defective HAM5 protein.

Fig. S8. Venn diagram of the number of CAZ-GCs and SM-BGCs in CBS1-2, Gv29-8, P1 and FT-101.

Fig. S9. Comparison of the nucleotide sequences within the SID-BGCs in CBS1-2, Gv29-8, P1 and FT-101.

Fig. S10. Comparison of the nucleotide sequences within the FRC-BGCs in CBS1-2, Gv29-8, P1 and FT-101.

Fig. S11. Comparison of the nucleotide sequences within the CGP-BGCs in CBS1-2, Gv29-8, P1 and FT-101.

Fig. S12. The entire *usk1-SOR-BGC-axe1-cip1-cel61a* chromosomal region encompassing the *T. reesei*-specific SOR-BGC contains 14 protein-encoding genes but no AT-rich blocks.

Fig. S13. Comparison of the nucleotide sequences within the TRI-BGCs in P1 and FT-101.

Fig. S14. Amino acid sequence alignment of *A. fumigatus* GliT protein (top row) and *Trichoderma* orthologs.

Fig. S15. Amino acid sequence alignment of *A. fumigatus* and *Trichoderma* GliC proteins.

Fig. S16. Amino acid sequence alignment of *P. chrysogenum* and *Trichoderma* SorD proteins.

Fig. S17. Relatively synchronous cross between the female CBS1-1 mycelium and the male QM6a or CBS1-2 conidia.

DS1. Genome-wide annotation of protein-encoding genes in QM6a.

DS2. Genome-wide annotation of protein-encoding genes in CBS1-1.

DS3. Genome-wide annotation of protein-encoding genes in CBS1-2.

DS4. Genome-wide annotation of protein-encoding genes in P1.

DS5. Genome-wide annotation of protein-encoding genes in FT-101.

DS6. Genome-wide annotation of protein-encoding genes in Gv29-8.

DS7. Genome-wide annotation of protein-encoding genes in FT-333.

DS8. Evolutionarily conserved protein-coding genes in QM6a, CBS1-1, CBS1-2, P1, FT-101, Gv29-8 and FT-333.

DS9. Annotation of the mitogenomes of QM6a, CBS1-1, CBS1-2, P1, FT-101, Gv29-8 and FT-333.

- DS10.** Comparative analysis of sexual genes in genomes of seven *Trichoderma* strains and *Neurospora crassa*.
- DS11.** List of CAZ-GCs in the genome of QM6a (A), CBS1-1 (B), CBS1-2 (C), P1 (D), FT-101 (E), Gv29-8 (F) and FT-333 (G).
- DS12.** Proteomic identification of the secreted proteins in the cultural filtrates of Gv29-8 (A, B), FT333 (C, D), CBS1-2 (E, F), P1 (G, H) and FT101 (I, J).
- DS13.** Transcriptionally up-regulated FPGRs in Gv29-8 at 6 hr.
- DS14.** Transcriptionally up-regulated AFCGs in Gv29-8 at 30 hr.
- DS15.** Evolutionarily conserved FPGRs and AFCGs in Gv29-8, CBS1-2, P1 and FT-101.
- DS16.** Transcriptionally up-regulated VGGs in CBS1-2.
- DS17.** Transcriptionally up-regulated SDIGs in CBS1-2.
- DS18.** Transcriptionally up-regulated ESDGs in CBS1-2.
- DS19.** Transcriptionally up-regulated MSDGs in CBS1-2.
- DS20.** Transcriptionally up-regulated LSDGs in CBS1-2.
- DS21.** Transcriptionally up-regulated CSDGs in CBS1-2.
- DS22.** Evolutionarily conserved VGGs, SDIGs, ESDG, MSDGs, LSDG and CSDGs in CBS1-2, Gv29-8, P1 and FT101.

Acknowledgments

This work was supported by the Institute of Molecular Biology, Academia Sinica, Taiwan, Republic of China. We thank Shu-Yun Tung (IMB Genomics Core) for the NGS sequencing service, Kun-Hai Ye (IMB Bioinformatics Core) for statistical assistance and bioinformatics consultancy, John O'Brien for English editing, and Yu-Tang Huang (IMB Computer Room) for maintaining the computer workstation. This work was supported by Academia Sinica, Taipei, Taiwan, Republic of China.

Authors' contributions

HFN and RSC provided the two biocontrol strains, *T. asperellum* FT-101 and *T. virens* FT-333. MS provided *T. virens* Gv29-8 and *T. atroviride* P1. WCL and CLC performed HMW gDNA isolation and Funannotate gene prediction. HCL perform lncRNA prediction. TZL performed comparative analyses of SMs. JLC performed nanopore sequencing. HNL assembled genome sequences. HLC performed in-solution trypsin digestion. WCL and TFW performed comparative genomic analyses. WTF conceived and designed the experiments. WTF, WCL and RSC wrote the manuscript. All authors read and approved the manuscript.

Data Availability

The complete genome sequences, annotation and raw datasets have been deposited in the National Center for Biotechnology Information (<https://www.ncbi.nlm.nih.gov/bioproject/>) under accession number PRJNA700774 (*SI, Table S1*). All study data are included in the article and SI Appendix.

Conflict of interest statement

None declared.

Ethics approval and consent to participate

Not applicable.

References

- Abe N, Sugimoto O, Arakawa T, Tanji K, Hirota A. 2001. Sorbicillinol, a key intermediate of bisorbicillinoid biosynthesis in *Trichoderma* sp. USF-2690. *Biosci Biotechnol Biochem* 65:2271-2279.
- Aramayo R, Selker EU. 2013. *Neurospora crassa*, a model system for epigenetics research. *Cold Spring Harb Perspect Biol* 5:a017921.
- Atanasova L, Knox BP, Kubicek CP, Druzhinina IS, Baker SE. 2013. The polyketide synthase gene *pks4* of *Trichoderma reesei* provides pigmentation and stress resistance. *Eukaryot Cell* 12:1499-1508.
- Bansal R, Sherkhane PD, Oulkar D, Khan Z, Banerjee K, Mukherjee PK. 2018. The Viridin Biosynthesis Gene Cluster of *Trichoderma virens* and Its Conservancy in the Bat White-Nose Fungus *Pseudogymnoascus destructans*. *Chemistryselect* 3:1289-1293.
- Barreau C, Iskandar M, Loubradou G, Levallois V, Begueret J. 1998. The *mod-A* suppressor of nonallelic heterokaryon incompatibility in *Podospora anserina* encodes a proline-rich polypeptide involved in female organ formation. *Genetics* 149:915-926.
- Beier S, Hinterdobler W, Monroy AA, Bazafkan H, Schmoll M. 2020. The Kinase USK1 Regulates Cellulase Gene Expression and Secondary Metabolite Biosynthesis in *Trichoderma reesei*. *Frontiers in Microbiology* 11:974.
- Blanchard JL, Schmidt GW. 1996. Mitochondrial DNA migration events in yeast and humans: integration by a common end-joining mechanism and alternative perspectives on nucleotide substitution patterns. *Mol Biol Evol* 13:537-548.
- Borkovich KA, Alex LA, Yarden O, Freitag M, Turner GE, Read ND, Seiler S, Bell-Pedersen D, Paietta J, Plesofsky N, et al. 2004. Lessons from the genome sequence of *Neurospora crassa*: tracing the path from genomic blueprint to multicellular organism. *Microbiol Mol Biol Rev* 68:1-108.

- Bulgari D, Fiorini L, Gianoncelli A, Bertuzzi M, Gobbi E. 2020. Enlightening Gliotoxin Biological System in Agriculturally Relevant *Trichoderma* spp. *Frontiers in Microbiology* 11:200.
- Cardoza RE, Malmierca MG, Hermosa MR, Alexander NJ, McCormick SP, Proctor RH, Tijerino AM, Rumbero A, Monte E, Gutierrez S. 2011. Identification of loci and functional characterization of trichothecene biosynthesis genes in filamentous fungi of the genus *Trichoderma*. *Appl Environ Microbiol* 77:4867-4877.
- Chang SS, Zhang Z, Liu Y. 2012. RNA interference pathways in fungi: mechanisms and functions. *Annu Rev Microbiol* 66:305-323.
- Chen CL, Kuo HC, Tung SY, Hsu PW, Wang CL, Seibel C, Schmoll M, Chen RS, Wang TF. 2012. Blue light acts as a double-edged sword in regulating sexual development of *Hypocrea jecorina* (*Trichoderma reesei*). *PLoS One* 7:e44969.
- Chuang YC, Li WC, Chen CL, Hsu PW, Tung SY, Kuo HC, Schmoll M, Wang TF. 2015. *Trichoderma reesei* meiosis generates segmentally aneuploid progeny with higher xylanase-producing capability. *Biotechnol Biofuels* 8:30.
- Coghlan A, Eichler EE, Oliver SG, Paterson AH, Stein L. 2005. Chromosome evolution in eukaryotes: a multi-kingdom perspective. *Trends Genet* 21:673-682.
- Contreras-Cornejo HA, Macías-Rodríguez L, del-Val E, Larsen J. 2020. Interactions of *Trichoderma* with plants, insects, and plant pathogen microorganisms: chemical and molecular bases. In: Mérillon JM, Ramawat KG, editors. *Co-Evolution of Secondary Metabolites*: Springer International Publishing. p. 263-290.
- Corner EJH. 1932. The identification of the brown-root fungus. *Gdms Bull Straits Settlem* 5:317-350.

Dattenbock C, Tisch D, Schuster A, Monroy AA, Hinterdobler W, Schmoll M. 2018. Gene regulation associated with sexual development and female fertility in different isolates of *Trichoderma reesei*. *Fungal Biol Biotechnol* 5:9.

Derntl C, Guzman-Chavez F, Mello-de-Sousa TM, Busse HJ, Driessen AJM, Mach RL, Mach-Aigner AR. 2017. In Vivo Study of the Sorbicillinoid Gene Cluster in *Trichoderma reesei*. *Frontiers in Microbiology* 8:2037.

Djonovic S, Pozo MJ, Dangott LJ, Howell CR, Kenerley CM. 2006. Sm1, a proteinaceous elicitor secreted by the biocontrol fungus *Trichoderma virens* induces plant defense responses and systemic resistance. *Mol Plant Microbe Interact* 19:838-853.

Druzhinina I, Kubicek CP. 2005. Species concepts and biodiversity in *Trichoderma* and *Hypocrea*: from aggregate species to species clusters? *J Zhejiang Univ Sci B* 6:100-112.

Druzhinina IS, Kubicek CP. 2016. Familiar Stranger: Ecological Genomics of the Model Saprotroph and Industrial Enzyme Producer *Trichoderma reesei* Breaks the Stereotypes. *Adv Appl Microbiol* 95:69-147.

du Buy HG, Riley FL. 1967. HYBRIDIZATION BETWEEN THE NUCLEAR AND KINETOPLAST DNA'S OF *Leishmania enriettii* AND BETWEEN NUCLEAR AND MITOCHONDRIAL DNA'S OF MOUSE LIVER. *Proc Natl Acad Sci U S A* 57:790-797.

Fischer MS, Glass NL. 2019. Communicate and Fuse: How Filamentous Fungi Establish and Maintain an Interconnected Mycelial Network. *Frontiers in Microbiology* 10:619.

Gladyshev E. 2017. Repeat-Induced Point Mutation and Other Genome Defense Mechanisms in Fungi. *Microbiol Spectr* 5.

Guo Y, Yao S, Yuan T, Wang Y, Zhang D, Tang W. 2019. The spatiotemporal control of KatG2 catalase-peroxidase contributes to the invasiveness of *Fusarium graminearum* in host plants. *Mol Plant Pathol* 20:685-700.

Guzman-Guzman P, Porras-Troncoso MD, Olmedo-Monfil V, Herrera-Estrella A. 2019. *Trichoderma* Species: Versatile Plant Symbionts. *Phytopathology* 109:6-16.

Hammond TM, Xiao H, Boone EC, Perdue TD, Pukkila PJ, Shiu PK. 2011. SAD-3, a Putative Helicase Required for Meiotic Silencing by Unpaired DNA, Interacts with Other Components of the Silencing Machinery. *G3 (Bethesda)* 1:369-376.

Harman GE, Howell CR, Viterbo A, Chet I, Lorito M. 2004. *Trichoderma* species--opportunistic, avirulent plant symbionts. *Nat Rev Microbiol* 2:43-56.

Hunter N. 2015. Meiotic Recombination: The Essence of Heredity. *Cold Spring Harb Perspect Biol* 7.

Karlsson J, Saloheimo M, Siika-Aho M, Tenkanen M, Penttila M, Tjerneld F. 2001. Homologous expression and characterization of Cel61A (EG IV) of *Trichoderma reesei*. *Eur J Biochem* 268:6498-6507.

Khan RAA, Najeeb S, Mao Z, Ling J, Yang Y, Li Y, Xie B. 2020. Bioactive Secondary Metabolites from *Trichoderma* spp. against Phytopathogenic Bacteria and Root-Knot Nematode. *Microorganisms* 8.

Kim KE, Peluso P, Babayan P, Yeadon PJ, Yu C, Fisher WW, Chin CS, Rapicavoli NA, Rank DR, Li J, et al. 2014. Long-read, whole-genome shotgun sequence data for five model organisms. *Sci Data* 1:140045.

Krzywinski M, Schein J, Birol I, Connors J, Gascoyne R, Horsman D, Jones SJ, Marra MA. 2009. Circos: an information aesthetic for comparative genomics. *Genome Res* 19:1639-1645.

Kubicek CP, Herrera-Estrella A, Seidl-Seiboth V, Martinez DA, Druzhinina IS, Thon M, Zeilinger S, Casas-Flores S, Horwitz BA, Mukherjee PK, et al. 2011. Comparative genome sequence analysis underscores mycoparasitism as the ancestral life style of *Trichoderma*. *Genome Biol* 12:R40.

Kubicek CP, Steindorff AS, Chenthamara K, Manganiello G, Henrissat B, Zhang J, Cai F, Kopchinskiy AG, Kubicek EM, Kuo A, et al. 2019. Evolution and comparative genomics of the most common *Trichoderma* species. *BMC Genomics* 20:485.

Lai YJ, Lin FM, Chuang MJ, Shen HJ, Wang TF. 2011. Genetic requirements and meiotic function of phosphorylation of the yeast axial element protein red1. *Mol Cell Biol* 31:912-923.

Li W-C, Chuang Y-C, Chen C-L, Wang T-F. 2016. Hybrid Infertility: The Dilemma or Opportunity of Applying Sexual Development to Improve *Trichoderma reesei* Industrial Strains. In: Schmoll M, Dattenböck C, editors. *Gene Expression Systems in Fungi: Advancements and Applications*. Cham: Springer International Publishing. p. 351-359.

Li WC, Chen CL, Wang TF. 2018. Repeat-induced point (RIP) mutation in the industrial workhorse fungus *Trichoderma reesei*. *Appl Microbiol Biotechnol* 102:1567-1574.

Li WC, Chuang YC, Chen CL, Timofejeva L, Pong WL, Chen YJ, Wang CL, Wang TF. 2019. Two different pathways for initiation of *Trichoderma reesei* Rad51-only meiotic recombination. In. *BioRxiv*.

Li WC, Huang CH, Chen CL, Chuang YC, Tung SY, Wang TF. 2017. *Trichoderma reesei* complete genome sequence, repeat-induced point mutation, and partitioning of CAZyme gene clusters. *Biotechnol Biofuels* 10:170.

Li WC, Lee CY, Lan WH, Woo TT, Liu HC, Yeh HY, Chang HY, Chuang YC, Chen CY, Chuang CN, et al. 2021. *Trichoderma reesei* Rad51 tolerates mismatches in hybrid meiosis with diverse genome sequences. *Proc Natl Acad Sci* 118:e2007192118.

Li WC, Liu HC, Lin YJ, Tung SY, Wang TF. 2020. Third-generation sequencing-based mapping and visualization of single nucleotide polymorphism, meiotic recombination, illegitimate mutation and repeat-induced point mutation *NAR Genomics and Bioinformatics* 2:lqaa056.

- Li WC, Wang TF. 2021. PacBio Long-Read Sequencing, Assembly, and Funannotate Reannotation of the Complete Genome of *Trichoderma reesei* QM6a. *Methods Mol Biol* 2234:311-329.
- Lim FY, Sanchez JF, Wang CC, Keller NP. 2012. Toward awakening cryptic secondary metabolite gene clusters in filamentous fungi. *Methods Enzymol* 517:303-324.
- Linke R, Thallinger GG, Haarmann T, Eidner J, Schreiter M, Lorenz P, Seiboth B, Kubicek CP. 2015. Restoration of female fertility in *Trichoderma reesei* QM6a provides the basis for inbreeding in this industrial cellulase producing fungus. *Biotechnol Biofuels* 8:155.
- Liu HC, Li WC, Wang TF. 2021. TSETA: A Third-Generation Sequencing-Based Computational Tool for Mapping and Visualization of SNPs, Meiotic Recombination Products, and RIP Mutations. *Methods Mol Biol* 2234:331-361.
- Lundblad V, Blackburn EH. 1993. An alternative pathway for yeast telomere maintenance rescues est1- senescence. *Cell* 73:347-360.
- Lydeard JR, Jain S, Yamaguchi M, Haber JE. 2007. Break-induced replication and telomerase-independent telomere maintenance require Pol32. *Nature* 448:820-823.
- Malinich EA, Wang K, Mukherjee PK, Kolomiets M, Kenerley CM. 2019. Differential expression analysis of *Trichoderma virens* RNA reveals a dynamic transcriptome during colonization of *Zea mays* roots. *BMC Genomics* 20:280.
- Malkova A, Ivanov EL, Haber JE. 1996. Double-strand break repair in the absence of RAD51 in yeast: a possible role for break-induced DNA replication. *Proc Natl Acad Sci U S A* 93:7131-7136.
- Martinez D, Berka RM, Henrissat B, Saloheimo M, Arvas M, Baker SE, Chapman J, Chertkov O, Coutinho PM, Cullen D, et al. 2008. Genome sequencing and analysis of the biomass-degrading fungus *Trichoderma reesei* (syn. *Hypocrea jecorina*). *Nat Biotechnol* 26:553-560.

Masunaka A, Hyakumachi M, Takenaka S. 2011. Plant growth-promoting fungus, *Trichoderma koningi* suppresses isoflavonoid phytoalexin vestitol production for colonization on/in the roots of *Lotus japonicus*. *Microbes Environ* 26:128-134.

McCluskey K, Wiest AE, Grigoriev IV, Lipzen A, Martin J, Schackwitz W, Baker SE. 2011. Rediscovery by Whole Genome Sequencing: Classical Mutations and Genome Polymorphisms in *Neurospora crassa*. *G3 (Bethesda)* 1:303-316.

Medema MH, Blin K, Cimermancic P, de Jager V, Zakrzewski P, Fischbach MA, Weber T, Takano E, Breitling R. 2011. antiSMASH: rapid identification, annotation and analysis of secondary metabolite biosynthesis gene clusters in bacterial and fungal genome sequences. *Nucleic Acids Res* 39:W339-346.

Meng J, Wang X, Xu D, Fu X, Zhang X, Lai D, Zhou L, Zhang G. 2016. Sorbicillinoids from Fungi and Their Bioactivities. *Molecules* 21.

Montenecourt BS, Eveleigh DE. 1977. Preparation of mutants of *Trichoderma reesei* with enhanced cellulase production. *Appl Environ Microbiol* 34:777-782.

Mousa WK, Raizada MN. 2015. Biodiversity of genes encoding anti-microbial traits within plant associated microbes. *Front Plant Sci* 6:231.

Mukherjee PK, Horwitz BA, Kenerley CM. 2012. Secondary metabolism in *Trichoderma*--a genomic perspective. *Microbiology* 158:35-45.

Nielsen H. 2017. Predicting Secretory Proteins with SignalP. *Methods Mol Biol* 1611:59-73.

Nutzmann HW, Scazzocchio C, Osbourn A. 2018. Metabolic Gene Clusters in Eukaryotes. *Annu Rev Genet* 52:159-183.

Funannotate: Fungal genome annotation scripts. [Internet]. 2017. Available from: <https://github.com/nextgenusfs/funannotate>

Peterson R, Nevalainen H. 2012. *Trichoderma reesei* RUT-C30--thirty years of strain improvement. *Microbiology* 158:58-68.

Purdy LH. 1979. *Sclerotinia sclerotiorum*: History, diseases and symptomatology, host range, geographic distribution, and impact. The American Phytopathological Society 69:875-880.

Ramirez-Valdespino CA, Casas-Flores S, Olmedo-Monfil V. 2019. *Trichoderma* as a Model to Study Effector-Like Molecules. Frontiers in Microbiology 10:1030.

Ricchetti M, Fairhead C, Dujon B. 1999. Mitochondrial DNA repairs double-strand breaks in yeast chromosomes. Nature 402:96-100.

Ricchetti M, Tekaia F, Dujon B. 2004. Continued colonization of the human genome by mitochondrial DNA. PLoS Biol 2:E273.

Rovenich H, Boshoven JC, Thomma BP. 2014. Filamentous pathogen effector functions: of pathogens, hosts and microbiomes. Curr Opin Plant Biol 20:96-103.

Scharf DH, Heinekamp T, Remme N, Hortschansky P, Brakhage AA, Hertweck C. 2012. Biosynthesis and function of gliotoxin in *Aspergillus fumigatus*. Appl Microbiol Biotechnol 93:467-472.

Scharf DH, Remme N, Heinekamp T, Hortschansky P, Brakhage AA, Hertweck C. 2010. Transannular disulfide formation in gliotoxin biosynthesis and its role in self-resistance of the human pathogen *Aspergillus fumigatus*. J Am Chem Soc 132:10136-10141.

Schmoll M, Dattenbock C, Carreras-Villasenor N, Mendoza-Mendoza A, Tisch D, Aleman MI, Baker SE, Brown C, Cervantes-Badillo MG, Cetz-Chel J, et al. 2016. The Genomes of Three Uneven Siblings: Footprints of the Lifestyles of Three *Trichoderma* Species. Microbiol Mol Biol Rev 80:205-327.

Schrettl M, Carberry S, Kavanagh K, Haas H, Jones GW, O'Brien J, Nolan A, Stephens J, Fenelon O, Doyle S. 2010. Self-protection against gliotoxin--a component of the gliotoxin biosynthetic cluster, GliT, completely protects *Aspergillus fumigatus* against exogenous gliotoxin. PLoS Pathog 6:e1000952.

- Schuster A, Schmoll M. 2010. Biology and biotechnology of *Trichoderma*. *Appl Microbiol Biotechnol* 87:787-799.
- Seidl V, Seibel C, Kubicek CP, Schmoll M. 2009. Sexual development in the industrial workhorse *Trichoderma reesei*. *Proc Natl Acad Sci U S A* 106:13909-13914.
- Selker EU. 1990. Premeiotic instability of repeated sequences in *Neurospora crassa*. *Annu Rev Genet* 24:579-613.
- Smith KM, Galazka JM, Phatale PA, Connolly LR, Freitag M. 2012. Centromeres of filamentous fungi. *Chromosome Res* 20:635-656.
- Sneh B. 1996. *Rhizoctonia* Species: taxonomy, molecular biology, ecology, pathology and disease control. : Springer.
- Steyaert JM, Weld RJ, Mendoza-Mendoza A, Stewart A. 2010. Reproduction without sex: conidiation in the filamentous fungus *Trichoderma*. *Microbiology* 156:2887-2900.
- Tamura K, Peterson D, Peterson N, Stecher G, Nei M, Kumar S. 2011. MEGA5: molecular evolutionary genetics analysis using maximum likelihood, evolutionary distance, and maximum parsimony methods. *Mol Biol Evol* 28:2731-2739.
- Teichert I, Poggeler S, Nowrousian M. 2020. *Sordaria macrospora*: 25 years as a model organism for studying the molecular mechanisms of fruiting body development. *Appl Microbiol Biotechnol* 104:3691-3704.
- Teng SC, Zakian VA. 1999. Telomere-telomere recombination is an efficient bypass pathway for telomere maintenance in *Saccharomyces cerevisiae*. *Mol Cell Biol* 19:8083-8093.
- Terrapon N, Lombard V, Gilbert HJ, Henrissat B. 2015. Automatic prediction of polysaccharide utilization loci in Bacteroidetes species. *Bioinformatics* 31:647-655.
- Tsuji J, Frith MC, Tomii K, Horton P. 2012. Mammalian NUMT insertion is non-random. *Nucleic Acids Res* 40:9073-9088.

Tyler BM. 2002. Molecular basis of recognition between phytophthora pathogens and their hosts. *Annu Rev Phytopathol* 40:137-167.

Vinale F, Sivasithamparam K, Ghisalberti EL, Ruocco M, Wood S, Lorito M. 2012. Trichoderma secondary metabolites that affect plant metabolism. *Nat Prod Commun* 7:1545-1550.

Vleeshouwers VG, Oliver RP. 2015. Effectors as Tools in Disease Resistance Breeding Against Biotrophic, Hemibiotrophic, and Necrotrophic Plant Pathogens. *Mol Plant Microbe Interact* 2015:40-50.

Wang KD, Gorman Z, Huang PC, Kenerley CM, Kolomiets MV. 2020. Trichoderma virens colonization of maize roots triggers rapid accumulation of 12-oxophytodienoate and two -ketols in leaves as priming agents of induced systemic resistance. *Plant Signal Behav* 15:1792187.

Wang S, Zickler D, Kleckner N, Zhang L. 2015. Meiotic crossover patterns: obligatory crossover, interference and homeostasis in a single process. *Cell Cycle* 14:305-314.

Waterhouse RM, Tegenfeldt F, Li J, Zdobnov EM, Kriventseva EV. 2013. OrthoDB: a hierarchical catalog of animal, fungal and bacterial orthologs. *Nucleic Acids Res* 41:D358-365.

White M. 1978. Modes of speciation. In: San Francisco : W. H. Freeman, c1978.

Yao L, Tan C, Song J, Yang Q, Yu L, Li X. 2016. Isolation and expression of two polyketide synthase genes from *Trichoderma harzianum* 88 during mycoparasitism. *Braz J Microbiol* 47:468-479.

Yu X, Gabriel A. 1999. Patching broken chromosomes with extranuclear cellular DNA. *Mol Cell* 4:873-881.

Zamocky M, Gasselhuber B, Furtmuller PG, Obinger C. 2012. Molecular evolution of hydrogen peroxide degrading enzymes. *Arch Biochem Biophys* 525:131-144.

Zhang GJ, Dong R, Lan LN, Li SF, Gao WJ, Niu HX. 2020. Nuclear Integrants of Organellar DNA Contribute to Genome Structure and Evolution in Plants. *Int J Mol Sci* 21.

Zhang X, Hindra, Elliot MA. 2019. Unlocking the trove of metabolic treasures: activating silent biosynthetic gene clusters in bacteria and fungi. *Curr Opin Microbiol* 51:9-15.

Ziemert N, Alanjary M, Weber T. 2016. The evolution of genome mining in microbes - a review. *Nat Prod Rep* 33:988-1005.

Zubenko GS, Jones EW. 1981. Protein degradation, meiosis and sporulation in proteinase-deficient mutants of *Saccharomyces cerevisiae*. *Genetics* 97:45-64.

Fig. 1. *Trichoderma* spp. produce and secrete different SMs. (A) Maximum likelihood phylogeny of seven common *Trichoderma* strains based on 500 conserved single-copy genes. The number on the scale bar represents percentage genetic variation (4%). (B) Variation in color and morphology of fungal colonies of *Trichoderma* spp. upon growth on Potato Dextrose Agar (PDA). (C) Growth of *Trichoderma* spp. in PDB liquid medium.

Fig. 2. CIRCOS plots of the genomes of indicated *Trichoderma* spp. and their syntenic relationships. The outer circle indicates the seven chromosomes of the near complete genome sequences. The GC contents (window size of 5000 bp) are shown in the middle traces. Chromosome synteny (with ≥ 10 consecutive protein-encoding genes) is depicted in the inner ribbon tracks of the diagrams. (A) Chromosome synteny between *T. reesei* CBS1-2, *T. asperellum* FT-101, *T. atroviride* P1 and *T. virens* FT-333. (B) Chromosome synteny between *T. virens* FT-333 and *T. virens* Gv29-8. There is a reciprocal translocation event between the first chromosome of FT-333 and the fourth chromosome of Gv29-8. The right telomeres of the second chromosomes in FT-333 and Gv29-8 contain two NUMTs, which are depicted by a black ribbon track in the inner circle of the plot. (C) Chromosome synteny between *T. asperellum* FT-101 and *T. atroviride* P1. *T. asperellum* and *T. atroviride* are members of Section *Trichoderma*. There are at least six reciprocal translocation events between the genomes of these two biocontrol agents.

Fig. 3. Chromosome maps of CBS1-2, Gv29-8, P1 and FT-101. Centromere locations are represented by narrowed segments. All CAZyme genes (in grey) were chosen along the sequence to be used as location markers. All CAZ-GCs and SM-BGCs are indicated in cyan and red, respectively.

Fig. 4. Comparison of the predicted proteomes encoded by the near complete genome sequences of CBS1-2, FT-101, P1, and Gv29-8. (A) Venn diagram of the overall number of annotated protein-encoding genes. (B) Venn diagram of evolutionarily conserved FPRGs and AFCGs (*SD*, *DS15*). (C) Venn diagram of evolutionarily conserved VGGs, SDIGs, ESDGs, MSDGs, LSDGs and CSDGs (*SD*, *DS22*).

Fig. 5. Comparison of the nucleotide sequences within and around the mating-type loci in P1, FT-101, CBS1-1, CBS1-2, Gv29-8 and FT-333. The tracks between two strains are color-coded to indicate nucleotide sequence identity. The mating-type genes (*mat1-1-1*, *mat1-1-2*, *mat1-1-3* and *mat1-2-1*) are dissimilar in sequence, but they are found at the same loci on the third chromosomes and are all flanked by two evolutionarily conserved genes, the DNA lyase *apn2* and the complex I intermediate-associated protein 30 gene *cia30*.

Fig. 6. Comparative analysis of GTX- or GTX-like BGCs and secondarily metabolites in *T. virens* and *T. reesei*. (A-B) Comparison of the nucleotide sequences within and around GTX- or GTX-like BGCs of Gv29-8 and FT-333 (A), and of QM6a and CBS1-2 (B). Gene name, gene identities, and chromosome locations of all biosynthetic genes are indicated. The GC contents (window size of 5000 bp) are also shown. (C) TLC images of the crude SMs isolated from vegetative mycelia and the developing fruiting bodies at different stages (D2-D6) during sexual developmental. GTX (compound 1) was used as a control for TLC.

Table 1. Summary of the annotated genes in the seven near-complete *Trichoderma* genome sequences.

Species	T. reesei	T. reesei	T. reesei	T. virens	T. virens	T. asperellum	T. atroviride
Strain	CBS999.97 (MATI-1)	CBS999.97 (MATI-2)	QM6a	Gv29-8	FT-333	FT-101	P1
Sequencing technology	PacBio	PacBio	PacBio	Nanopore	Nanopore	PacBio	Nanopore
Locus_tag	TRC1	TRC2	TrQ	TrV	TrVFT-333	TrA	TrAt
Genome size (base pairs)	34,319,199	34,324,311	34,922,528	40,979,523	41,418,917	37,545,380	37,300,646
BUSCO protein metrics (%) ¹	99.2	98.9	98.4	95.5	94.2	98.6	90.8
Single complete (S) %	S:93.5	S:92.2	S:91.8	S: 95.1	S:93.7	S:96.2	S: 88.4
Duplicated complete (D) %	D:5.7	D:6.7	D:6.6	D:0.4	D:0.5	D:2.4	D:2.4
Fragment (F) %	F:0.3	F:0.3	F:0.3	F:1.6	F:2.5	F:0.8	F:5.9
Missing (M) %	M:0.5	M:0.8	M:1.3	M:2.9	M:3.3	M:0.6	M:3.3
Protein-coding genes	10,292	10,225	10,238	12,263	11,895	12,041	13,327
Proteins	11,090	11,087	11,038	12,064	11,698	12,454	13,583
tRNA genes	150	144	159	202	200	184	185
Predicted gene clusters	32	32	31	57	57	54	45
Transcription Factors (TF)	691	710	680	739	641	882	843
HET domains (PF06985)	41	42	44	68	60	55	75
Ankyrins (PF00023)	79	81	75	117	137	105	114
CAZymes ²							
Auxiliary Activity (AA)	48	51	49	66	58	54	58
Carbohydrate-binding Modules (CBM)	11	11	12	16	14	13	13
Carbohydrate Esterases (CE)	33	32	33	47	42	43	39
Glycoside Hydrolases (GH)	199	198	199	230	222	240	229
Glycosyl Transferases (GT)	89	91	88	71	72	77	77
Polysaccharide Lyases (PL)	7	7	7	7	7	8	9
CAZ-GCs	31	31	29	35	26	31	33
Predicted proteins with signal peptides (SP)							
Total secretory signal peptides ³	866	874	840	1072	995	1050	1041
Oxidoreductases (GO:0016491)	43	42	45	61	60	50	54
Hydrolases (GO:0016787)	203	197	202	247	225	253	250
Transferases (GO:0016740)	10	11	11	9	8	11	9
Catalytic activity (GO:0003824)	284	280	282	357	328	347	350
Lysases (GO:0016829)	2	2	2	3	4	3	4
Ligases (GO:0016874)	0	1	0	0	0	0	1
Isomerases (GO:0016853)	5	5	5	5	5	5	7
Peptidoglycan murelytic activity (GO:0061783)	2	2	2	2	2	0	1
Secondary metabolite biosynthesis (SMB)							
NRPS	7	7	11	17	22	15	7
PKS/NRPS-like proteins	17	19	18	32	40	38	30
Hybrid PKS-NRPS	4	4	1	11	12	5	4

NRPS-like proteins	8	8	7	16	16	19	22
Type I Iterative PKS	10	10	10	16	15	12	16
PKS-like proteins	0	0	0	0	1	0	0
Cytochrome P450 (CYP) PF00067	79	80	71	104	87	72	67
Predicted gene clusters							
SM-BGC ⁴	32	32	32	58	57	52	46

1. Gene annotation completeness was evaluated in BUSCO (v4.1.4) using the database for fungi_odb10.
2. The CAZyme genes were determined by using the dbCAN2 meta server (<http://ccb.unl.edu/dbCAN2>) {Zhang, 2018}.
3. The *SignalP* server was used to predict the presence and location of signal peptide cleavage sites in amino acid sequences {Nielsen, 2017}.
4. The gene clusters were determined by using the antiMASH software tool.

Table 2. Comparative analyses of seven previously characterized SM-BGCs. OrthoVenn2 (<https://orthovenn2.bioinfotoolkits.net/home>) was applied to search ($E = \leq 10^{-5}$) for orthologs of SMB-GCs. Gene IDs and their chromosome number (in brackets) are indicated.

Siderophore (SID)-BGCs					
JGI ID	CBS1-2	Gv29-8	P1	FT-101	Annotation
	SM-BGC 4.2	SM-BGC 6.4	SM-BGC 7.3	SM-BGC 6.5	
Tr-67189	TRC2_005918 (IV)	TrV_010298 (VI)	TrAt_012594 (VII)	TrA_010145 (VI)	pks4 NRPS
Tr-66999	TRC2_005919 (IV)	TrV_010299 (VI)	TrAt_012595 (VII)	TrA_010144 (VI)	NRPS-like protein
Tr-67109	TRC2_005920 (IV)	TrV_010300 (VI)	TrAt_012596 (VII)	TrA_010143 (VI)	O-acetyltransferase sat14
Tr-67026	TRC2_005921 (IV)	TrV_010301 (VI)	TrAt_012597 (VII)	TrA_010142 (VI)	Siderochrome iron transporter 1
Tr-5206	TRC2_005922 (IV)	TrV_010302 (VI)	TrAt_012598 (VII)	TrA_010141 (VI)	Oxidoreductase
Tr-110499	TRC2_005923 (IV)	TrV_004842 (III)	TrAt_012599 (VII)	TrA_010139 (VI)	ABC transporter
Ferrichrome (FRC)-BGCs					
JGI ID	CBS1-2	Gv29-8	P1	FT-101	Annotation
	SM-BGC 2.2	SM-BGC 7.5	SM-BGC 2.3	SM-BGC 4.3	
Tr-69972	TRC2_002088 (II)	TrV_011144 (VII)	TrAt_002779 (II)	TrA_006545 (IV)	TF
Tr-23368	TRC2_002089 (II)	TrV_011145 (VII)	TrAt_002782 (II)	TrA_006547 (IV)	PAK-GC kinase sid1
-	TRC2_002090 (II)	-	-	-	Hypothetical protein
Tr-69946	TRC2_002091 (II)	TrV_011146 (VII)	TrAt_002784 (II)	TrA_006548 (IV)	NRPS
Tr-23367	TRC2_002092 (II)	TrV_011147 (VII)	TrAt_002785 (II)	TrA_006549 (IV)	Oxidoreductase
Tr-52375	TRC2_002093 (II)	TrV_011148 (VII)	TrAt_002786 (II)	TrA_006551 (IV)	Aldehyde dehydrogenase
Conidial green pigment (CGP)-BGCs					
JGI ID	CBS1-2	Gv29-8	P1	FT-101	Annotation
	SM-BGC 4.3	SM-BGC 6.10	SM-BGC 5.9	SM-BGC 6.9	
Tr-37950	TRC2_006733 (IV)	TrV_004745 (II)	-	-	Cytochrome P450
Tr-70127	TRC2_006734 (IV)	TrV_004747 (II)	-	-	RTA-like protein
Tr-112115	TRC2_006735 (IV)	TrV_004748 (II)	-	-	MFS transporter
Tr-112114	TRC2_006736 (IV)	-	-	-	Hypothetical protein
Tr-124079	TRC2_006737 (IV)	TrV_010890 (VI)	TrAt_010562 (V)	TrA_010757 (VI)	Multicopper oxidase
Tr-52476	TRC2_006738 (IV)	TrV_010891 (VI)	TrAt_010561 (V)	TrA_010756 (VI)	Hypothetical protein
Tr-82208	TRC2_006739 (IV)	TrV_010892 (VI)	TrAt_010560 (V)	TrA_010755 (VI)	Polyketide synthase
Tr-112105	TRC2_006740 (IV)	TrV_009953 (VI)	TrAt_011901 (VI)	-	Hypothetical protein
Sorbicillinoid (SOR)-BGCs					
JGI ID	CBS1-2	Gv29-8	P1	FT-101	Annotation
	SM-BGC 5.2	-	-	-	
Tr-53776	TRC2_007361 (V)	TrV_007389 (IV)	TrAt_007655 (IV)	TrA_011012 (VII)	usk1
Tr-102492	TRC2_007362 (V)	TrV_008886 (V)	TrAt_000805 (I)	TrA_001604 (I)	sor8
Tr-73618	TRC2_007363 (V)	TrV_000231 (I)	TrAt_002239 (I)	TrA_000207 (I)	sor1
Tr-73621	TRC2_007364 (V)	-	TrAt_008990 (IV)	-	sor2

Tr-73623	TRC2_007365 (V)	-	TrAt_008980 (IV)	TrA_006848 (IV)	sor5
Tr-43701	TRC2_007366 (V)	-	-	-	sor4 (MSF)
Tr-43701	TRC2_007367 (V)	-	-	-	sor4_2
Tr-102497	TRC2_007368 (V)	TrV_006641 (III)	TrAt_010535 (V)	-	sor3/ypr2
Tr-73631	TRC2_007369 (V)	TrV_006377 (III)	-	-	sor7
Tr-102499	TRC2_007370 (V)	-	-	-	sor6/ypr1
Tr-102500	TRC2_007371 (V)	TrV_008882 (V)	TrAt_000804 (I)	TrA_001605 (I)	
Tr-73632	TRC2_007372 (V)	TrV_006140 (III)	TrAt_004919 (II)	TrA_004330 (II)	axe1_1
Tr-73638	TRC2_007373 (V)	TrV_012230 (VII)	TrAt_005240 (III)	TrA_011951 (VII)	
Tr-73643	TRC2_007374 (V)	TrV_012229 (VII)	TrAt_005241 (III)	TrA_011952 (VII)	cel61a
Trichothecene (TRI)-BGCs					
JGI ID	FT-101	CBS1-2	Gv29-8	P1	Annotation
	SM-BGC 7.3	-	-	-	
Ta-61224	TrA_010945 (VII)	-	TrV_005642 (III)	TrAt_010304 (V)	Oxidoreductase
Ta-30398	TrA_010946 (VII)	-	TrV_006414 (III)	-	
Ta-201111	TrA_010947 (VII)	-	-	TrAt_010299 (V)	Alcohol dehydrogenase
Ta-72517	TrA_010948 (VII)	TRC2_008174 (VI)	TrV_006398 (III)	TrAt_010298 (V)	
Ta-448313	TrA_010949 (VII)	-	-	-	Trichodiene synthase tri5
Ta-142955	TrA_010950 (VII)	TRC2_008175 (VI)	TrV_006397 (III)	TrAt_010297 (V)	
Ta-72520	TrA_010951 (VII)	-	-	TrAt_010295 (V)	
Ta-92494	TrA_010952 (VII)	TRC2_008193 (VI)	TrV_006337 (III)	TrAt_010294 (V)	
Viridin (VIR)-BGCs					
JGI ID	Gv29-8	CBS1-2	P1	FT-101	Annotation
	SM-BGC 5.5	-	-	-	
Tv-53366	TrV_009673 (V)	-	-	TrA_010785 (VII)	vdn1 CYP
Tv-53375	TrV_009674 (V)	-	-	TrA_010784 (VII)	vdn2 CYP
Tv-230790	TrV_009675 (V)	-	-	-	vdn3 CYP
Tv-53368	TrV_009676 (V)	-	-	-	vdn3
Tv-60010	TrV_009677 (V)	-	-	-	-
-	TrV_009678 (V)	-	-	-	-
Tv-151179	TrV_009679 (V)	-	-	-	vdn21
Tv-78733	TrV_009680 (V)	-	-	-	vdn22
Tv-216144	TrV_009681 (V)	-	-	-	vdn19 CYP
Tv-128161	TrV_009682 (V)	-	-	-	vdn18 , oxidoreductase
Tv-91392	TrV_009683 (V)	-	-	-	vdn17
Tv-151341	TrV_009684 (V)	-	-	-	vdn10
Tv-135139	TrV_009685 (V)	-	-	-	vdn16 , MFS superfamily
Tv-70971	TrV_009686 (V)	-	-	-	vdn15

Tv-53581	TrV_009687 (V)	-	-	-	vdn14 , glyoxalase/dioxygenase
Tv-53582	TrV_009688 (V)	-	-	-	vdn13
Tv-151337	TrV_009689 (V)	-	-	-	vdn12 CYP
Tv-230740	TrV_009690 (V)	-	-	-	vdn1 O-methyltransferase
Tv-151142	TrV_009691 (V)	-	-	-	vdn9
Tv-53690	TrV_009692 (V)	-	-	-	vdn8 CYP
Tv-151220	TrV_009693 (V)	-	-	-	vdn7 dehydrogenase/reductase
Tv-78735	TrV_009694 (V)	-	-	-	vdn6
Tv-170440	TrV_009695 (V)	-	-	-	vdn5
Glitoxin (GTX)-BGCs					
JGI ID	Gv29-8	CBS1-2	P1	FT-101	Annotation
	SM-BGC 1.1	SM-BGC 6.4			
Tv-216146	TrV_000002 (I)	-	-	-	gliA
Tv-83751	TrV_000003 (I)	-	-	-	
Tv-83751	TrV_000004 (I)	-	-	-	
Tv-216149	TrV_000005 (I)	-	-	-	
Tv-138628	TrV_000006 (I)	-	-	-	gliT
Tv-216138	TrV_000007 (I)	-	-	-	gliH
Tv-78708	TrV_000008 (I)	TRC2_008324 (VI)	-	-	gliP
Tv-216161	TrV_000009 (I)	TRC2_008325 (VI)	-	-	gliC
Tv-91355	TrV_000010 (I)	-	-	-	gliN
Tv-151379	TrV_000011 (I)	TRC2_008323 (VI)	-	-	gliK
Tv-53497	TrV_000012 (I)	TRC2_008326 (VI)	-	-	gliI
Tv-216157	TrV_000013 (I)	TRC2_008321 (VI)	-	-	gliG
Tv-91346	TrV_000014 (I)	-	-	-	gliF
Tv-216154	TrV_000015 (I)	TRC2_008316 (VI)	-	-	gliM
Tv-159420	TrV_000016 (I)	-	-	-	
Tv-201436	TrV_000017 (I)	-	-	-	
Tv-216163	TrV_000018 (I)	TRC2_008322 (VI)	-	-	gliJ

Table3. The number of protein-encoding genes transcriptionally upregulated during vegetative growth and different sexually developmental stages

	The overall gene number in the Gv29-8 genome (12006 protein-encoding genes)	FPRG (365)	AFCG (2082)
Signal peptide protein	1073	42	349
CAZyme	425	14	158
Protease	415	13	100
Membrane protein	2528	75	433
TFs	560	1	87
SM-BGC 1.1 ^{GTX}	11		10
SM-BGC 1.3	16	1	6
SM-BGC 1.4	13		3
SM-BGC 1.5	12		4
SM-BGC 1.6	16		5
SM-BGC 1.7	13	2	4
SM-BGC 1.8	10		3
SM-BGC 1.9	6		3
SM-BGC 1.10	21	1	8
SM-BGC 1.12	12		2
SM-BGC 1.14	14	2	1
SM-BGC 2.1	9		4
SM-BGC 2.2	14		2
SM-BGC 2.3	7		3
SM-BGC 2.4	6		
SM-BGC 2.5	20		4
SM-BGC 2.6	13	1	2
SM-BGC 2.8	13		1
SM-BGC 2.9	17	5	7
SM-BGC 3.1	16		3
SM-BGC 3.2	11	1	
SM-BGC 3.3	7		2
SM-BGC 3.4	14		3
SM-BGC 3.5	8		1
SM-BGC 3.6	7		
SM-BGC 3.7	11	1	
SM-BGC 3.8	17		8
SM-BGC 3.9	6		5
SM-BGC 3.10	12		1
SM-BGC 3.11	9		5
SM-BGC 3.12	19		1
SM-BGC 3.13	12	4	
SM-BGC 4.1	12		2

SM-BGC 4.3	15		5
SM-BGC 4.4	11		1
SM-BGC 4.5	7		1
SM-BGC 5.1	15		7
SM-BGC 5.2	13		
SM-BGC 5.3	15		
SM-BGC 5.4	9		3
SM-BGC 5.5 ^{VIR}	23		
SM-BGC 6.1	7		3
SM-BGC 6.2	15		4
SM-BGC 6.3	15		
SM-BGC 6.4 ^{SID}	11		
SM-BGC 6.5	8		2
SM-BGC 6.6	11		3
SM-BGC 6.7	5		
SM-BGC 6.8	14		2
SM-BGC 6.9	7		
SM-BGC 6.10 ^{CGP}	15		7
SM-BGC 6.11	11		6
SM-BGC 7.1	14	2	1
SM-BGC 7.2	1		
SM-BGC 7.3	8		3
SM-BGC 7.4	15		1
SM-BGC 7.5 ^{FER}	16	1	2
SM-BGC 7.6	18		
SM-BGC 7.7	7		4
CAZ-GC 1.1	11		5
CAZ-GC 1.2	6		
CAZ-GC 1.3	6	1	3
CAZ-GC 1.4	6	1	
CAZ-GC 1.5	7		2
CAZ-GC 2.1	5		1
CAZ-GC 2.2	6		
CAZ-GC 2.3	4		
CAZ-GC 2.4	5		2
CAZ-GC 2.5	5		2
CAZ-GC 3.1	5		2
CAZ-GC 3.2	6		3
CAZ-GC 3.3	7	1	2
CAZ-GC 3.4	6		2
CAZ-GC 4.1	9	1	1

CAZ-GC 4.2	6		4
CAZ-GC 4.3	4		2
CAZ-GC 4.4	5		
CAZ-GC 4.5	12	1	2
CAZ-GC 4.6	8		5
CAZ-GC 4.7	8		5
CAZ-GC 5.1	12		4
CAZ-GC 5.2	8		3
CAZ-GC 5.3	6		1
CAZ-GC 5.4	5		1
CAZ-GC 5.5	6		2
CAZ-GC 5.6	5		3
CAZ-GC 5.7	6		
CAZ-GC 6.1	8		5
CAZ-GC 6.2	7	1	3
CAZ-GC 6.3	4		
CAZ-GC 6.4	6		3
CAZ-GC 7.1	5		
CAZ-GC 7.2	5		1
CAZ-GC 7.3	7	1	2

Table 4. The number of protein-encoding genes transcriptionally upregulated during vegetative growth and different sexually developmental stages

	Overall gene number in the CBS999.97(MATI-2) genome (11087 protein-encoding genes)	VGG (1217)	SDIG (767)	ESDG (1219)	MSDG (403)	LSDG (81)	CSDG (851)
Signal peptide protein	874	165	85	132	34	14	78
CAZyme	375	63	27	58	10	2	31
Protease	344	43	22	37	10	3	20
Membrane protein	2476	291	192	236	77	18	192
TFs	525	57	29	42	4	3	24
SM-BGC 1.1	15	2	2	4	1		
SM-BGC 1.2	14	5		1			
SM-BGC 1.3	6		2		1		
SM-BGC 1.4	10	3	2	1			
SM-BGC 1.6	15	9		2			3
SM-BGC 2.1	8	5		2			
SM-BGC 2.2 ^{FER}	18	2	3	2			3
SM-BGC 2.3	15	8					1
SM-BGC 2.4	7	1					2
SM-BGC 2.6	12	5		2	1		2
SM-BGC 2.7	7				2		
SM-BGC 2.8	8	2	1				
SM-BGC 3.1	18						
SM-BGC 3.2	12		3	2	1		1
SM-BGC 3.3	7	2					1
SM-BGC 3.4	20	4	1	2			5
SM-BGC 4.1	11	1	3		1		
SM-BGC 4.2 ^{SID}	14	4		2			1
SM-BGC 4.3 ^{CGP}	8	1	1	3			
SM-BGC 5.1	17	4		3			2
SM-BGC 5.2 ^{SOR}	16	10		3			1
SM-BGC 5.3	14	2		2	1		1
SM-BGC 6.1	14		2	6			
SM-BGC 6.2	15	2		8	3		1
SM-BGC 6.3	6	3					
SM-BGC 6.4 ^{GTX}	19	1	2	12			
SM-BGC 6.5	16	3		6			
SM-BGC 6.6	15	3	1				
SM-BGC 6.7	3		2				
SM-BGC 7.1	14	4		3			
SM-BGC 7.2	5	2	1				
SM-BGC 7.4	7	1	2		1		

CAZ-GC 1.1	5			2			
CAZ-GC 1.2	11	2		7			2
CAZ-GC 1.3	12	6		1			1
CAZ-GC 1.4	5	1			1		1
CAZ-GC 1.5	5	1					
CAZ-GC 2.1	5			1			2
CAZ-GC 2.2	5	2	1	2			
CAZ-GC 2.3	7	1	2	2			
CAZ-GC 2.4	10	1	1	1			1
CAZ-GC 2.5	5	1			1		
CAZ-GC 2.6	6	2					1
CAZ-GC 2.7	7	7					
CAZ-GC 3.1	7	1	1	1			
CAZ-GC 3.2	6	1	11	1			
CAZ-GC 3.3	8	3		2			
CAZ-GC 4.1	4			1	2		
CAZ-GC 4.2	5	2		2			1
CAZ-GC 4.3	6			1			2
CAZ-GC 4.4	5	1		1			
CAZ-GC 4.5	6	2					
CAZ-GC 4.6	5	1		2			1
CAZ-GC 5.1	10	6		2			
CAZ-GC 5.2	5			2			1
CAZ-GC 5.3	6						1
CAZ-GC 6.1	4	1		2			
CAZ-GC 6.2	6	3		2			
CAZ-GC 6.3	6			1	2		
CAZ-GC 7.1	5	1	2				
CAZ-GC 7.2	5						1
CAZ-GC 7.3	5		1				
CAZ-GC 7.4	6	2	1	1			2

Supporting Information

Complete Genome Sequences and Genome-Wide of *Trichoderma* Biocontrol Agents Provide New Insights into their Evolution and Variation in Fungal-Plant Interactions, Sexual Development and Genome Defense

Wan-Chen Li¹, Ting-Chan Lin^{1,2}, Chia-Ling Chen¹, Hou-Cheng Liu¹, Hisn-Nan Lin¹, Ju-Lan Chao¹, Cheng-Hsilin Hsieh¹, Hui-Fang Ni³, Ruey-Shyang Chen^{2,#}, Ting-Fang Wang^{1,2,#}

¹Institute of Molecular Biology, Academia Sinica, Taipei 115, Taiwan

²Department of Biochemical Science and Technology, National Chiayi University, Chiayi 600, Taiwan

³Department of Plant Protection, Chiayi Agricultural Experiment Station, Council of Agriculture, Chiayi 600, Taiwan

#Corresponding authors:

RSC, email: rschen@mail.ncyu.edu.tw

TFW, email: tfwang@gate.sinica.edu.tw

The supplemental information (*SI*) file includes supplemental "Materials and Methods", 9 supplemental tables and 17 supplemental figures.

Twenty-two supplemental datasets (DS1-DS22) in Excel format are also provided separately online (<https://github.com/tfwangasimb/Supplemental-datasets/releases/tag/20210611>).

Other authors' email addresses:

Wan-Chen Li	wanwan9121@hotmail.com
Ting-Chan Lin:	jacky1998711@gmail.com
Chia-Ling Chen	chialing.chen1118@gmail.com
Hou-Cheng Liu	hc666@gate.sinica.edu.tw
Ju-Lan Chao	julan@gate.sinica.edu.tw
Hsin-Nan Lin	arith@gate.sinica.edu.tw
Cheng-Hsilin Hsieh	imbhch@gate.sinica.edu.tw
Hui-Fang Ni	hfni@tari.gov.tw

Additional “Materials and Methods”

Fungal growth inhibition assays

A slightly modified cellophane method [1] was applied to determine the growth-inhibition capabilities of *Trichoderma* spp. against four different plant fungal pathogens. In brief, a 5-mm diameter Potato Dextrose Agar (PDA) plug of *Trichoderma* spp. was placed at the center of a sterilized cellophane sheet placed over a PDA plate. After incubation at 25 °C for 2 days, the cellophane was removed and a single 5-mm diameter mycelial plug of plant pathogen was placed at the center of each plate. Each pathogen growing on PDA alone served as a control. All PDA plates were incubated at 25 °C.

Extraction and analysis of secondary metabolites (SMs)

Vegetative mycelia and developing fruiting bodies were harvested from the MEA plates covering cellophane. The materials were then frozen in liquid nitrogen and stored at -80 °C until use. About 0.3 g of material was crushed in liquid nitrogen and transferred to a 2 mL eppendorf tube with 1 mL chloroform. After vortexing for 20 min, samples were centrifuged (10,000x g, 5 min). The organic phase was collected and solvents were evaporated in a chemical extraction cabinet. The crude SM materials were resuspended in 20 µl of chloroform. Then, 2 µl was resolved in chloroform:acetone (9:1) on thin layer chromatography (TLC) Silica gel 60 F254 plates (Sigma Aldrich). Gliotoxin (GTX; Sigma Aldrich) was spotted as a standard. The TLC plates were then visualized using a 365-nm handheld UV lamp (YouLee Technology Co., New Taipei City, Taiwan).

Whole-genome DNA sequencing, RNA sequencing and whole-genome gene prediction

Isolation of genomic DNA and RNA, as well as PacBio single-molecular real-time (SMRT) genome sequencing and assembly, were carried out as described previously [2-4]. Oxford Nanopore direct DNA library preparation and sequencing were performed at the Genomic Core

Lab of the Institute of Molecular Biology, Academia Sinica. In brief, small fragments of purified genomic DNA were removed by using the Ampure cleanup kit (Beckman Coulter). High-quality DNA quantification was conducted using a Qubit fluorometer (Thermo Fisher Scientific). The average length of genomic DNA was scaled using Femto Pulse (Agilent). We used 1 µg of genomic DNA for library construction. The library contained 3 or 4 DNA samples barcoded using the EXP-NBD104 Rapid barcoding kit (Oxford Nanopore Technologies), and adaptors were added with the SQK-LSK109 ligation sequencing kit (Oxford Nanopore Technologies). The library was loaded onto a R9.4.1 Flow Cell system (FLO-MIN106) and processed for 48 h on the MinION platform. The sequencing process was controlled using the MinKNOW software (version 3.6.5; Oxford Nanopore Technologies). All experimental procedures were carried out according to the manufacturer's instructions. FAST5 data files were generated upon completion of sequencing. These files were converted into FASTQ files in Guppy (v3.6.0). All reads were split into separate FASTQ files based on their barcode by means of the qcat tool (v1.1.0), and then Canu (v2.1.1) [5] was used to perform whole-genome assembly on the corresponding FASTQ files for each sample. The parameter of corOutCoverage was set to 60 based on sequencing read depth. Finally, we used medaka (v1.2.0) to polish the assemblies with the raw reads. Since Canu generated expected numbers of contigs for each sample, and a comparison of the genome to wild-type QM6a indicated no broken contigs, we did not perform any manual finishing or validation. The completeness of genome assemblies was evaluated in BUSCO (v4.0.6) [6] against the database of fungi_odb10 (OrthoDB; <https://www.orthodb.org>). All other experimental procedures have been described previously in detail [3], including PacBio SMRT genome sequencing, NGS-based RNA sequencing, as well as genome-wide gene prediction in Funannotate [7]. BUSCO was also used to quantitatively measure the completeness of genome assemblies and gene predictions. We selected evolutionarily-informed expectations of gene content based on the Ascomycota odb9

database from OrthoDB (<https://www.orthodb.org>). Genome or protein matrix scores >95% for model organisms are generally deemed complete reference genomes [8].

Analysis of genome-wide synteny

For comparative genome analyses and to identify duplicated regions, we identified orthologous gene pairs using the annotation results generated in Funannotate [7]. Alignments were performed using BLASTP with an expect value ($E \leq 1e-20$). CIRCOS [9] was used to display sequence similarity and conservation. The inner ribbon track of CIRCOS outputs is used to show synteny, whereas the exterior tracks quantify the degree of sequence conservation between the *T. reesei* CBS1-2 genome and those of the three other *Trichoderma* species.

Comparative genomic analysis

Funannotate [7] not only parses the protein-coding models from the annotation, but also identifies numbers and classes of CAZymes, proteases, transcriptional factors, secreted proteins, polyketide synthases (PKSs), and nonribosomal peptide synthetases (NRPSs). To predict secreted proteins, we downloaded the SignalP 4.1 [10], TMHMM 2.0 [11], and big-PI Fungal Predictor [12] programs into “Funannotate” and then applied them with default settings. In this study, effector candidate proteins were defined as predicted secreted proteins (with a signal peptide present, but no transmembrane domains or glycosylphosphatidylinositol anchors) having lengths of <300 amino acids. To assess if effector candidates presented similarity to known proteins, we performed BlastP analysis (cutoff E-value 10^{-5}) using the Swiss-Prot database (downloaded October 22, 2016). To determine phylogenetic relationships between different *Trichoderma* wild isolates, we employed the orthology detection tool “Proteinortho” to select 500 single-copy orthologous proteins from each individual strain [13]. The multiple sequence alignment program MAFFT was then used to align concatenated protein sequences. Next, we used IQ-tree (<http://www.iqtree.org/>) to generate phylogenetic trees using the

concatenation matrix [14]. These phylogenetic tools are incorporated into “Funannotate” [7]. TreeViewer (TreeDyn version 198.3) was used to plot phylogenetic trees.

The carbohydrate-active enzymes (CAZymes) were reannotated using a meta server dbCAN2 (<http://bcb.unl.edu/dbCAN2>) [15]. We have developed a software tool ‘IBM-CAZ’ to predict potential CAZyme gene clusters (CAZ-GCs) with the following requirements: a potential CAZ-GC must contain ≥ 3 CAZyme genes or ≥ 2 CAZyme genes with ≥ 1 other specific signature genes (namely transporters or transcription factors), and be present within ≤ 2 intergenic distances. Although the prediction requirements in IBM-CAZ are more stringent than those of dbCAN2 [15] and dbCAN-PUL [16], IBM-CAZ was able to identify all four previously identified CAZ-GCs in *Trichoderma reesei* QM6a, the ancestor of all currently used cellulase-producing mutants [17].

Proteomics analysis of secreted proteins

To identify proteins secreted by different *Trichoderma* strains, we germinated 1 mL of conidia ($OD_{600nm} = 0.3$) and cultured it at 25 °C in 250 mL potato dextrose broth (PDB) medium in shake flasks at 120 rpm for 24 h. The vegetative mycelia and conidia were removed from culture medium by filtering through a 0.22- μ m filter (Merck Millipore, Darmstadt, Germany). The proteins in the culture filtrate (CF) were precipitated by adding ammonium sulfate (AS) to 80% saturation at 4 °C. After centrifugation at $10,000 \times g$ for 30 min at 4 °C, the AS precipitates were recovered and then resuspended in 500 μ L of 50 mM Tris-HCl (pH 7.5). Protein concentration was determined using Bradford reagent (Sigma Aldrich).

For in-solution trypsin digestion, the target proteins (20 μ g) were reconstituted in 100 mM Tris-HCl (pH 8.0) and 5% acetonitrile, mixed with 1/10 volume of 100 mM 1,4-dithiothreitol in 100 mM Tris-HCl (pH 8.0) and incubated at 37 °C for 1 h. Then, 1/10 volume of 550 mM iodoacetamide in 100 mM Tris-HCl (pH 8.0) was added to the mixture, gently vortexed, and incubated at room temperature for 1 h. Promega Sequencing Grade Modified

Trypsin was added to give a final substrate:trypsin ratio of 50:1. The digestion reaction was carried out overnight at 37 °C and then the reaction was stopped by adding 5% formic acid to adjust the pH of the solution to below pH 6.0. We determined pH by placing 1- μ L aliquots onto pH paper. The peptide digestion products were desalted using C₁₈ Zip-Tips (Millipore, ZTC 18M 096). The peptide solutions were dried down in a SpeedVac vacuum concentrator (Thermo Fisher Scientific) and stored at -20 °C before undergoing mass spectrometric analysis.

Liquid chromatography-mass spectrometry (LC-MS-MS) was applied to analyze trypsin-digested peptides. LC-MS analysis was performed by Shu-Yu Lin (Institute of Biological Chemistry, Academia Sinica) using an EASY-nLC 1200 system linked to a Thermo Orbitrap Fusion Lumos mass spectrometer equipped with a Nanospray Flex ion source (Thermo Fisher Scientific) located at the Academia Sinica Common Mass Spectrometry Facilities (<https://www.ibc.sinica.edu.tw/facilities/mass-spectrometry-facilities/>). All related experimental procedures were described previously in detail [18]. Proteomic data were searched against our whole-genome annotation datasets (see below) using the Mascot search engine (v.2.6.2; Matrix Science, Boston, MA, USA) in Proteome Discoverer (v 2.2.0.388; Thermo Fisher Scientific, Waltham, MA, USA). We used the search criterion trypsin digestion, the fixed modification was set as carbamidomethyl (C), and variable modifications were set as oxidation (M), acetylation (protein N-terminal) allowing up to two missed cleavages, and mass accuracy of 10 ppm for the parent ion and 0.6 Da for the fragment ions. The false discovery rate (FDR) was calculated with the Proteome Discoverer Percolator function, and identifications with an FDR > 1% were rejected.

Table S1. List of all *Trichoderma* strains analyzed in this study

Strain name	References
Trichoderma reesei QM6a	[2]
Trichoderma reesei CBS1-2	[19]
Trichoderma reesei CBS1-2	[19]
Trichoderma virens Gv29-8	NCBI Bioproject accession PRJNA700774
Trichoderma virens FT-333	NCBI Bioproject accession PRJNA700774
Trichoderma asperellum FT-101	NCBI Bioproject accession PRJNA700774
Trichoderma atroviride P1	NCBI Bioproject accession PRJNA700774

Table S2. Biocontrol activities of tested *Trichoderma* strains against four different plant fungal pathogens using a modified cellophane method

Plant fungal pathogens v.s. Trichoderma spp.		Phellinus noxius	Rhizoctonia solani	Sclerotium rolfsii	Phytophthora sp.
T. atroviride	P1	100%	100%	100%	100%
T. asperellum	FT-101	100%	100%	99±1%	100%
T. virens	FT-333	100%	100%	100%	100%
	Gv29-8	100%	100%	100%	100%
T. reesei	CBS1-2	92±6%	79±2%	92±7%	100%

Table S3. Summary of the properties of seven near-complete *Trichoderma* genome sequences

Species	T. reesei	T. reesei	T. reesei	T. virens	T. virens	T. asperellum	T. atroviride	
Strain	CBS1-1	CBS1-2	QM6a	Gv29-8	FT-333	FT-101	P1	
Sequencing technology	PacBio	PacBio	PacBio	Nanopore	Nanopore	PacBio	Nanopore	
Locus_tag	TRC1	TRC2	TrQ	TrV	TrVFT-333	TrA	TrAt	
Genome size (bp)	34,319,199	34,324,311	34,922,52	40,979,523	41,418,917	37,545,380	37,300,646	
No. chromosomes	7	7	7	7	7	7	7	
N50 (bp)	5,258,134	5,262,578	5,311,445	6,490,838	6,644,895	5,512,738	5,658,044	
GC (%)	51.63	51.63	51.08	47.35	47.07	47.06	48.72	
AT-blocks	2259	2250	2249	3577	3367	4570	5510	
Mitogenome size (bp)	38,995	39,005	42,130	27,943	31,081	30,285	29,981	
BUSCO genome metrics (%) ³	99.3	99.3	99.3	97.8	96.6	98.5	97.1	
Single complete (S) %	S:99.3	S:99.3	S:99.3	S: 97.4	S:95.9	S:98.5	S:96.8	
Duplicated complete (D) %	D:0.0	D:0.0	D:0.0	D:0.4	D:0.7	D:0.0	D:0.3	
Fragment (F) %	F:0.0	F:0.0	F:0.0	F:0.3	F:1.2	F:0.1	F:0.8	
Missing (M) %	M:0.7	M:0.7	M:0.7	M:1.9	M:2.2	M:1.4	M:2.1	
Transposable elements								
Overall	62	62	70	93	94	92	78	
Class I retrotransposons	Tad1-LINE	0	0	0	2	8	1	1
	RI-LINE	0	0	0	7	1	4	8
	Jockey-LINE	11	11	4	6	9	10	10
	other LINES	11	11	14	4	4	1	5
	Copia-LTR	5	5	8	4	4	4	3
	Gypsy-LTR	2	2	10	11	19	43	18
	other LTRs	6	6	4	8	7	7	9
Class II transposons	CMC-EnSpm	4	4	6	2	2	2	6
	MULE-MuDR	1	1	0	28	18	6	6
	hAT-Charlie	17	17	21	2	1	1	1
	TcMar -AntI	0	0	0	5	8	2	1
	PIF-Harbinger-like	0	0	0	1	0	1	0
	Others	5	5	3	13	13	10	10

Table S4. Numbers of annotated transcription factors encoded by the near-complete genomes of the seven *Trichoderma* spp.

InterPro	Description	CBS1-1	CBS1-2	QM6a	Gv29-8	FT-333	FT-101	P1
000967	NF-X1-type zinc finger	2	2	2	0	0	2	2
006856	Mating-type protein MAT alpha-1 HMG-Box	1	0	0	0	1	1	0
018501	DDT domain	1	1	1	1	1	1	1
007196	CCR4-Not complex component	1	1	1	1	1	1	1
007396	Putative FMN-binding domain	1	1	1	3	3	1	1
004595	TFIIH C1-like domain	1	1	1	1	0	1	0
004181	MIZ zinc finger	4	4	4	3	3	3	3
000818	TEA/ATTS domain family	1	1	1	1	1	1	1
001387	Helix-turn-helix	1	1	1	1	1	1	1
001289	CCAAT-binding TF (CBF-B/NF-YA) subunit B	1	1	1	1	1	1	1
003120	STE-like TF	1	1	1	1	1	1	1
003150	RFX DNA-binding domain	2	2	2	2	2	2	2
004198	Zinc finger	1	1	1	1	1	1	1
007604	CP2 TF	1	1	1	1	1	2	1
008895	YL1 nuclear protein	1	1	1	1	1	1	2
010770	SGT1 protein	1	1	1	1	1	1	1
018004	KilA-N domain	5	6	4	4	2	4	4
024061	NDT80/PhoG-like DNA-binding family	4	6	4	3	3	3	4
003656	BED zinc finger	1	0	0	2	1	1	1
018060	Bacterial regulatory HTH proteins	1	1	1	2	1	4	4
005011	SART-1 family	1	1	1	1	1	1	1
002100	SRF-type TF (DNA-binding and dimerization domain)	2	1	2	2	2	2	3
010666	GRF zinc finger	2	2	2	1	1	3	2
000232	HSF-type DNA-binding	4	3	5	3	3	4	4
001766	Fork head domain	4	4	4	5	5	4	4
001356	Homeobox domain	9	10	12	9	9	12	11
000679	GATA zinc finger	8	9	7	7	6	7	8
013767	PAS fold	1	1	1	0	0	0	0
001878	Zinc knuckle (CCHC)	13	13	11	10	10	11	14
04827	Basic region leucine zipper 2	27	28	28	27	28	28	26
00788	Helix-turn-helix	0	0	1	1	1	1	1
011598	Helix-loop-helix DNA-binding domain	10	10	10	10	9	10	10
001005	Myb-like DNA-binding domain	18	17	17	17	15	16	17
004827	bZIP TF 1	27	28	28	27	28	28	26
007087	Zinc finger (C2H2)	86	83	85	81	73	90	85
001138	Fungal Zn(2)-Cys(6) binuclear cluster domain	276	291	267	293	227	384	374
007219	Fungal-specific TF domain	171	176	170	215	197	248	225

Table S5. Numbers of SM-BGCs and CAZ-BGCs revealed by antiSMH

	Chromosome	CBS1-1	CBS1-2	QM6a	Gv29-8	FT-333	FT-101	P1
SM-BGC	ChI	5	5	5	11	10	9	8
	ChII	8	7	7	8	8	7	8
	ChIII	4	4	4	13	13	7	5
	ChIV	2	3	3	5	3	4	7
	ChV	3	3	3	4	6	6	9
	ChVI	7	7	7	11	10	9	2
	CVII	3	3	2	6	7	10	6
	Total	32	32	31	58	57	54	45
CAZ-BGC	ChI	5	5	4	5	3	6	6
	ChII	7	7	5	5	3	2	4
	ChIII	3	3	2	4	3	4	5
	ChIV	6	6	7	7	8	3	3
	ChV	3	3	3	7	2	5	3
	ChVI	3	3	3	4	4	7	6
	CVII	4	4	5	3	3	4	6
	Total	31	31	29	35	26	31	33

Table S6. The location of centromeres in CBS1-2, Gv29-8, FT-101 and P1

Chromosome	CBS1-2	Gv29-8	FT-101	P1
ChI	3184501-3359500	3568001-3671500	4246001-4351000	3038001-3130500
ChII	1892001-2062000	4049001-4156500	3025501-3130500	3084001-3208000
ChIII	1677501-1830500	4163501-4275500	1795501-1897500	1919001-2039000
ChIV	1453501-1617500	2098501-2211500	3176001-3297500	2118501-2220000
ChV	1102501-1259500	3087001-3195500	2775001-2874000	1969501-2113000
ChVI	1637001-1810500	1910501-1992500	2199001-2280000	3363501-3410500
CVII	1858001-2018500	2680501-2767000	1580001-1692000	1611001-1704000

Table S7. The number of evolutionarily conserved fungal-plant interaction genes in different *Trichoderma* species

	Gv29-8		CBS1-2	FT-333	FT-101	P1
Complete genome	All protein-encoding genes	12006	8541	10740	9076	9120
	Signal peptide proteins	1073	730	981	799	796
	CAZymes	425	326	406	351	352
	Proteases	415	292	393	322	332
	Membrane proteins	2528	1919	2313	2022	2014
	Transcription factors	560	393	509	455	445
FPRGs	All protein-encoding genes	365	215	326	244	248
	Signal peptide proteins	42	17	39	25	25
	CAZymes	14	7	13	9	10
	Proteases	13	6	13	10	10
	Membrane proteins	75	38	63	48	47
	Transcription factors	1	1	1	1	1
AFCGs	All protein-encoding genes	2082	1192	1803	1349	1375
	Signal peptide proteins	349	243	323	277	268
	CAZymes	158	123	151	135	134
	Proteases	100	62	94	72	76
	Membrane proteins	433	285	381	300	294
	Transcription factors	87	39	73	55	52

Table S8. The number of evolutionarily conserved vegetative growth and sexual development genes in different *Trichoderma* species

	CBS1-2	Gv29-8	FT-333	FT-101	P1	
Complete genome	All protein-encoding genes	11087	8437	8321	8328	8371
	Signal peptide proteins	874	722	710	694	697
	CAZymes	375	316	326	315	319
	Proteases	344	290	290	286	289
	Membrane proteins	2476	1953	1905	1908	1913
	Transcription factors	525	418	424	420	424
VGGs	All protein-encoding genes	1217	1023	980	960	988
	Signal peptide proteins	165	150	146	142	143
	CAZymes	63	56	61	56	57
	Proteases	43	43	42	42	42
	Membrane proteins	291	261	246	235	240
	Transcription factors	57	51	52	49	50
SDIGs	All protein-encoding genes	767	610	591	568	593
	Signal peptide proteins	85	65	65	58	64
	CAZymes	27	26	26	24	26
	Proteases	22	18	18	15	16
	Membrane proteins	192	154	146	147	150
	Transcription factors	29	27	28	27	27
ESDGs	All protein-encoding genes	1219	1016	999	968	1002
	Signal peptide proteins	132	113	114	110	115
	CAZymes	58	53	53	53	54
	Proteases	37	34	36	32	33
	Membrane proteins	236	204	202	196	203
	Transcription factors	42	38	38	37	40
MSDGs	All protein-encoding genes	403	268	263	239	250
	Signal peptide proteins	34	31	30	26	27
	CAZymes	10	8	10	7	8
	Proteases	10	10	10	10	10
	Membrane proteins	77	54	52	49	49
	Transcription factors	4	4	3	4	4
LSDGs	All protein-encoding genes	81	50	48	47	51
	Signal peptide proteins	14	12	11	12	12
	CAZymes	2	2	2	2	2
	Proteases	3	3	2	2	3
	Membrane proteins	18	14	14	16	16
	Transcription factors	3	2	1	2	2
CSDGs	All protein-encoding genes	851	576	565	537	561
	Signal peptide proteins	78	57	58	54	55
	CAZymes	31	30	30	27	29
	Proteases	20	19	18	18	20
	Membrane proteins	192	150	145	141	146
	Transcription factors	24	22	23	21	21

Table S9. Proteomic identification of proteins in culture filtrates

Strains	CBS1-2	FT-333	Gv29-8	FT-101	P1
Total proteins	8	74	98	21	37
SP proteins	6	30	34	17	29
CAZymes	3	14	17	9	17
Oxidoreductases	1	21	21	0	6
Proteases	0	16	16	3	11
Catalytic activity	2	16	15	2	3
Transferases	0	1	0	1	0
Lysases	0	2	1	0	0

Fig. S1. Circular maps of the complete mitogenomes of *T. reesei* QM6a and CBS1-2. The GC contents (window size of 5000 bp) are shown in the middle traces.

Fig. S2. Circular maps of the complete mitogenomes of *T. virens* Gv29.8 and FT-333. The nucleotide sequences homologous to two NUMTs are represented by red blocks. The GC contents (window size of 5000 bp) are shown in the middle traces.

Fig. S3. Circular maps of the complete mitogenomes of CBS1-2, Gv29.8, P1 and FT-333. The GC contents (window size of 5000 bp) are shown in the middle traces.

Fig. S4. Pairwise sequence alignments of the nucleotide sequences within and around the two NUMTs (in red) in the second chromosomes of Gv29-8 and FT-333. Grey shading highlights differences.

Fig. S5. The two NUMTs located in a long AT-rich block. NUMTs are represented by red blocks. The tracks between the two strains are color-coded to indicate nucleotide sequence identity. The GC contents (window size of 120 bp) of the seven chromosomes are shown.

Fig. S6. The numbers of species-specific genes that were functionally annotated by the Gene Ontology (GO) annotation software. GO terms are represented as general function categories.

Fig. S7. QM6a encodes a defective HAM5 protein. (A) Schematic illustration of the *ham5* gene locus in QM6a. Exons, introns and protein are indicated in dark grey, white, and light grey, respectively. Two putative protein-encoding genes (TrQ_000864 and TrQ_000865) were annotated in QM6a due to a G-to-T point mutation (indicated by a yellow arrow in B). (B) Alignment of the nucleotide sequences of the 5' portions (exon 1 to exon 4) of the *ham5* genes from four different *Trichoderma* genomes. Nucleotide sequence alignment was performed using MAFFT. The translational initiation site ("ATG" in red) of the FT-333 *ham5* gene differs from those of QM6a, CBS1-2 and FT-101.

Fig. S8. Venn diagram of the number of CAZ-GCs (A) and SM-BGCs (B) in CBS1-2, Gv29-8, P1 and FT-101.

Fig. S9. Comparison of the nucleotide sequences within the SID-BGCs in CBS1-2, Gv29-8, P1 and FT-101. The tracks between two strains are color-coded to indicate nucleotide sequence identity. Annotated SM biosynthetic genes and AT-rich blocks are indicated in different colors, respectively.

Fig. S10. Comparison of the nucleotide sequences within the FRC-BGCs in CBS1-2, Gv29-8, P1 and FT-101. The tracks between two strains are color-coded to indicate nucleotide sequence identity. Annotated SM biosynthetic genes and AT-rich blocks are indicated in different colors, respectively.

Fig. S11. Comparison of the nucleotide sequences within the CGP-BGCs in CBS1-2, Gv29-8, P1 and FT-101. The tracks between two strains are color-coded to indicate nucleotide sequence identity. Annotated SM biosynthetic genes and AT-rich blocks are indicated in different colors, respectively.

Fig. S12. The entire *usk1*-SOR-BGC-*axe1-cip1-cel61a* chromosomal region encompassing the *T. reesei*-specific SOR-BGC contains 14 protein-encoding genes but no AT-rich blocks. Gene name, gene identities, and chromosome locations (in brackets) of all biosynthetic genes are indicated. The GC content of the chromosomal region in *T. reesei* (window size of 5000 bp) is also shown at top.

Fig. S13. Comparison of the nucleotide sequences within the TRI-BGCs in P1 and FT-101. The tracks between two strains are color-coded to indicate nucleotide sequence identity. Annotated SM biosynthetic genes and AT-rich blocks are indicated in different colors, respectively. The ortholog of the trichodiene synthetase *tri5* (red arrow) is absent from P1.

Fig. S14. Amino acid sequence alignment of *A. fumigatus* GliT protein (top row) and *Trichoderma* orthologs. The amino acid sequences of signal peptides are indicated in red.

Fig. S15. Amino acid sequence alignment of *A. fumigatus* and *Trichoderma* GliC proteins. The amino acid sequences of signal peptides are indicated in red.

Fig. S16. Amino acid sequence alignment of *P. chrysogenum* and *Trichoderma* SorD proteins. The amino acid sequences of signal peptides are indicated in red.

Fig. S17. Relatively synchronous cross between the female CBS1-1 mycelium and the male QM6a conidia. (A) Photographs of representative developing stromata at the indicated day (D1-D14) upon induction of sexual development (scale bars: 0.5 cm). (C) Frozen sections of stromata (scale bars: 20 µm) were visualized by hematoxylin and eosin staining, as described previously [20].

References:

1. Dennis C, Webster J: **Antagonistic properties of species-groups of *Trichoderma*** *Trans Br Mycol Soc* 1971, **57**:25-39.
2. Li WC, Huang CH, Chen CL, Chuang YC, Tung SY, Wang TF: ***Trichoderma reesei* complete genome sequence, repeat-induced point mutation, and partitioning of CAZyme gene clusters.** *Biotechnol Biofuels* 2017, **10**:170.
3. Li WC, Wang TF: **PacBio Long-read sequencing, assembly, and Funannotate reannotation of the complete genome of *Trichoderma reesei* QM6a.** *Methods Mol Biol* 2021, **2234**:311-329.
4. Woo TT, Chuang CN, Wang TF: **Budding yeast Rad51: a paradigm for how phosphorylation and intrinsic structural disorder regulate homologous recombination and protein homeostasis.** *Curr Genet* 2021.
5. Koren S, Walenz BP, Berlin K, Miller JR, Bergman NH, Phillippy AM: **Canu: scalable and accurate long-read assembly via adaptive k-mer weighting and repeat separation.** *Genome Res* 2017, **27**:722-736.
6. Simao FA, Waterhouse RM, Ioannidis P, Kriventseva EV, Zdobnov EM: **BUSCO: assessing genome assembly and annotation completeness with single-copy orthologs.** *Bioinformatics* 2015, **31**:3210-3212.
7. **Funannotate: Fungal genome annotation scripts.**
[<https://github.com/nextgenusfs/funannotate>]
8. Seppely M, Manni M, Zdobnov EM: **BUSCO: Assessing Genome Assembly and Annotation Completeness.** *Methods Mol Biol* 2019, **1962**:227-245.
9. Krzywinski M, Schein J, Birol I, Connors J, Gascoyne R, Horsman D, Jones SJ, Marra MA: **Circos: an information aesthetic for comparative genomics.** *Genome Res* 2009, **19**:1639-1645.

10. Petersen TN, Brunak S, von Heijne G, Nielsen H: **SignalP 4.0: discriminating signal peptides from transmembrane regions.** *Nat Methods* 2011, **8**:785-786.
11. Krogh A, Larsson B, von Heijne G, Sonnhammer EL: **Predicting transmembrane protein topology with a hidden Markov model: application to complete genomes.** *J Mol Biol* 2001, **305**:567-580.
12. Eisenhaber B, Schneider G, Wildpaner M, Eisenhaber F: **A sensitive predictor for potential GPI lipid modification sites in fungal protein sequences and its application to genome-wide studies for *Aspergillus nidulans*, *Candida albicans*, *Neurospora crassa*, *Saccharomyces cerevisiae* and *Schizosaccharomyces pombe*.** *J Mol Biol* 2004, **337**:243-253.
13. Lechner M, Findeiss S, Steiner L, Marz M, Stadler PF, Prohaska SJ: **Proteinortho: detection of (co-)orthologs in large-scale analysis.** *BMC Bioinformatics* 2011, **12**:124.
14. Trifinopoulos J, Nguyen LT, von Haeseler A, Minh BQ: **W-IQ-TREE: a fast online phylogenetic tool for maximum likelihood analysis.** *Nucleic Acids Res* 2016, **44**:W232-235.
15. Zhang H, Yohe T, Huang L, Entwistle S, Wu P, Yang Z, Busk PK, Xu Y, Yin Y: **dbCAN2: a meta server for automated carbohydrate-active enzyme annotation.** *Nucleic Acids Res* 2018, **46**:W95-W101.
16. Ausland C, Zheng J, Yi H, Yang B, Li T, Feng X, Zheng B, Yin Y: **dbCAN-PUL: a database of experimentally characterized CAZyme gene clusters and their substrates.** *Nucleic Acids Res* 2021, **49**:D523-D528.
17. Martinez D, Berka RM, Henrissat B, Saloheimo M, Arvas M, Baker SE, Chapman J, Chertkov O, Coutinho PM, Cullen D, et al: **Genome sequencing and analysis of the biomass-degrading fungus *Trichoderma reesei* (syn. *Hypocrea jecorina*).** *Nat Biotechnol* 2008, **26**:553-560.

18. Shih PY, Hsieh BY, Lin MH, Huang TN, Tsai CY, Pong WL, Lee SP, Hsueh YP: **CTTNBP2 controls synaptic expression of zinc-related autism-associated proteins and regulates synapse formation and autism-like behaviors.** *Cell Rep* 2020, **31**:107700.
19. Li WC, Lee CY, Lan WH, Woo TT, Liu HC, Yeh HY, Chang HY, Chuang YC, Chen YC, Chuang CN, et al: ***Trichoderma reesei* Rad51 tolerates mismatches in hybrid meiosis with diverse genome sequences.** *Proc Natl Acad Sci USA* 2021, **118**:e2007192118.
20. Chen CL, Kuo HC, Tung SY, Hsu PW, Wang CL, Seibel C, Schmoll M, Chen RS, Wang TF: **Blue light acts as a double-edged sword in regulating sexual development of *Hypocrea jecorina* (*Trichoderma reesei*).** *PLoS One* 2012, **7**:e44969.

Response to Reviewers

Our point-by-point responses to reviewers' comments:
(the reviewers' comments are copied here in black, followed by our responses)

Reviewer #1 (Comments for the Author):

In this work, Li et al. continue the previous research line of the Ting-Fang Wang's group on the gapless genomics of *Trichoderma*. This group has previously reported a PacBio-based genome of *Trichoderma reesei*. In this work, they take the other three previously sequenced and best-annotated species (*T. virens*, *T. atroviride*, and *T. asperellum*) and present their third-generation genomics. As the authors also deposit the annotations, the work has its value and therefore is valuable for the community.

[tfwang] Thank you.

First, however, the authors must correct embarrassing errors in the ms. The plant-pathogenic fungi-like stramenopiles are not fungi, and they must be corrected all over the manuscript. *Trichoderma* is not an arbuscular mycorrhizal fungus. The number of plant cell wall degrading enzymes in *Trichoderma* is not vast (but very limited, as we see from the broad fungal genomic survey). And many others.

[tfwang] We have corrected the errors pointed out by this reviewer. In the revised manuscript, we have also deleted the first paragraph of our "Results" section, including original Figure 1 (*Trichoderma* spp. produce and secrete different SMs) and SI, Table 2 (Biocontrol activities of tested *Trichoderma* strains against four different plant fungal pathogens using a modified cellophane method).

As the ms has no line numbering, I cannot list it here but marked a pdf file.

[tfwang] Apologies, now corrected.

Even though I support this ms, I regretfully inform the editor that most of the analyses performed here are redundant to previously published data, trivial, or incorrect. Furthermore, the authors ignore the currently available knowledge on *Trichoderma* and fungal genomics and analyze their results as there are only these four species sequenced (mainly consider their previous work for the results even though they cite many studies in the introduction), however only for *T. atroviride* there are at least six genomes available in the public domain, while the total number of the genomes for this genus is > 50. Of course, third-generation sequencing presents an advantage, but it does not justify the ignorance of previous results.

[tfwang] We do not agree with this comment. First, as pointed out by this reviewer, we cited many studies in the Introduction, including the first NGS genome sequence paper on *T. atroviride* (Kubicek et al. 2011 *Genome Biology* 2, R40). Second, although >50 different *Trichoderma* genomes have been determined by NGS technology, these NGS genomic drafts are far from near-complete or chromosome-level assemblies due to the shortcomings of short NGS reads and the exceedingly low-complexity nature of *Trichoderma* genomes. The seven near-complete *Trichoderma* genomes we have determined by TGS technology represent the highest quality yet achieved. Third, the greatest benefit of precise genome assemblies is that they enable accurate determination of structural components in each genome (e.g., telomeres, centromeres, interspersed AT-rich blocks and authentic transposable elements), as well as the chromosome synteny between different *Trichoderma* species.

The work completely lacks any evolutionary context (the tree in Figure 1 is no advantage compared to the first comparative genomic study on *Trichoderma* published ten years ago).
[tfwang] Acknowledged. We have now deleted Figure 1 and Table S2 from our manuscript.

Trichoderma belongs to the order Hypocreales, where also there are many (maybe > 100) genomes available.

[tfwang] Again, to us, quality is more important than quantity. High-quality and near-complete TGS genomes are essential resources for flawless genomic and evolutionary studies.

However, the comparisons are made to *Neurospora crassa* and eurotialean fungi. It was a standard a decade ago.

[tfwang] We disagree with this comment. The classical fungal model organisms (i.e., *Neurospora crassa*, *Sordaria macrospora*, *Saccharomyces cerevisiae*) have been much better functionally annotated than any other fungal species, particularly for genes involved in sexual development, meiosis and genome defense.

Surprisingly, the work also contains some wet lab results, but they are equally inconclusive. First, the authors test whether their *Trichoderma* strains secrete antimicrobial water-soluble metabolites while cultivated on rich media. Yes, they do, it has been published numerous (very many) times for fungi and water molds (Oomycota), but it does not mean that *Trichoderma* will overgrow these (or other fungi) in direct confrontations. It has also been reported many times. [This section should either be deleted or described correctly, it is not an inhibition by *Trichoderma* but by *Trichoderma*'s WSM]/

[tfwang] Acknowledged, we have now deleted Figure 1 and Table S2 from the revised manuscript.

The secretome study could be potentially interesting, but again, it was done in catabolite repressing conditions and repeated numerous previous reports with highly overlapping results.

[tfwang] The aim of our secretome study was to validate the data from our genome annotation. To our knowledge, we are the first group to experimentally demonstrate that *T. reesei* Sor7 and *T. virens* GliT are secreted proteins. This finding is important for advancing our understanding of SM biosynthetic pathways in *Trichoderma*.

The transcriptional profiling of the fruiting bodies formation is also strange to find in this article as it is only relevant to one species and not to the others. So, yes, this part could be presented as a focused stand-alone publication.

[tfwang] Agreed. We have deleted the transcriptional profiling data on fruiting body formation from the revised manuscript.

The same applies to the mitochondrial genome: all results are confirmatory to the previous knowledge.

[tfwang] Although our results on *Trichoderma* mitogenomes are consistent with those of previous reports, we reveal for the first time that the mitogenomes of Gv29-8 and FT-333 harbor three nuclear-encoded mitochondrial sequences (NUMTs). NUMTs have not been explicitly reported for filamentous fungi before.

The Discussion section is short and superficial as the analysis presents minor novelty.

[tfwang] We disagree with this comment. Please refer to our responses to the reviewer's previous comments.

However, the dataset itself is valuable. Thus, this ms fits the scope of the Microbiology Spectrum.

[tfwang] Thank you.

Please note many comments in the results and supplements. Moreover, the terms "pathogen" and "biocontrol" are used incorrectly in this study. The authors should consider that Trichoderma is a mycoparasite and the fact that it attacks plant-pathogenic fungi does not allow to call them "pathogens" in the absence of plants. In this study, the term "pathogen" is more suitable for Trichoderma. Biocontrol is an agricultural practice. Trichoderma is just a fungus that interacts with other organisms what is used for biological control.

[tfwang] We have now clarified our terminology and have now deleted Figure 1 and Table S2 from our manuscript.

Reviewer #2 (Comments for the Author):

In this manuscript, the authors describe the generation of genome assemblies for four Trichoderma biocontrol agent strains using long-read sequencing. They perform detailed characterization of these genome assemblies, annotate them and attempt to gain insights into the evolution of biosynthetic gene clusters in this species. The manuscript is well written except in a few places, the methods are well described and the results are clear. There are a few places where the manuscript can be better organized and results need to be discussed/explained more. These results provide a number of interesting insights into these genomes and will make a very significant contribution to the field. I have a few comments/suggestions that might help them to improve the manuscript.

[tfwang] We thank this reviewer for his/her positive comments on our original manuscript, as well as the helpful comments and suggestions for improving it.

The result section 1 where they describe the activity against plant pathogens does not seem to fit into this manuscript. It does not connect with the rest of the manuscript. While the growth assays described are relevant, the fungal pathogen part is irrelevant and can be removed.

[tfwang] Agreed, we have now deleted Figure 1 and Table S2 from our manuscript.

Throughout the manuscript, the authors compare already published 3 wild-type isolate genomes with 4 biocontrol agents published in this study. They always refer to it by the strain names. I would suggest they use "wild-type isolates" versus "biocontrol agents" when doing this comparison. It will be easy to read and grasp than just the names of the strains.

[tfwang] Thank you. We have now modified our nomenclature as suggested.

As a follow-up to the previous comment, it would be nice if the authors provide genome comparative maps for all the seven strains they discuss in this paper. Maybe show chromosome maps for all seven strains in one figure - similar to figure 3 but with a genome comparative view like a synteny analysis.

[tfwang] We very much appreciate this excellent suggestion. We now present two new respective figures in the revised manuscript (Figure 1 and SI, Figure S1).

The authors state "Extensive chromosome rearrangements are likely the main determinants responsible for reproductive isolation of different Trichoderma species." While it is a

possibility, it can very well be the other way around. Chromosome rearrangements could have occurred after the speciation. Unless authors have a compelling argument or analysis to support their conclusion, I would suggest they refrain from making such a conclusion.

[tfwang] Acknowledged. We have now modified our subtitle to: “Extensive gross chromosome rearrangements between the genomes of different *Trichoderma* species”.

In the next sentence, they say that disruption of synteny mainly occurs next to the AT-rich regions. What do they mean by mainly - what fraction of synteny breaks are associated with AT-rich regions and how many are not?

[tfwang] We appreciate this insightful comment. We have now revised the statement in our manuscript and now mark all the chromosomal breakpoints that occur nearby or within the AT-rich regions in Figure 1.

Do these AT-rich regions include centromeres? If so, they may want to discuss this in more detail in light of a number of recent papers describing centromere mediated breaks in other fungi.

[tfwang] Yes, they do. In the revised manuscript, we now cite the paper by Yadav et al. (2020; Centromere scission drives chromosome shuffling and reproductive isolation. PNAS 117, 7917-7928).

The authors define centromeres as "most prominent or longest AT-rich blocks". What do they mean by "most prominent"? Also, how long are the other AT-rich blocks in the genome? Are there AT-rich blocks that are longer than current centromeres but were not considered as centromeres?

[tfwang] Apologies for the confusion, we have now deleted the term “most prominent” from this statement.

As in the filamentous fungal model organism *Neurospora crassa* (Sordariales, Ascomycota) (45-47) and QM6a (21), the putative centromeric loci in all seven *Trichoderma* genomes are not only the longest AT-rich blocks but also the longest regions of each chromosome lacking an open-reading frame (ORF) or putative protein-encoding genes (SI, Figure S4-S8 and Table S3). Using BLASTN with an E value of $1e-8$ (identity >80%) (48), all putative centromeric loci contain an array of repeats that are either short repetitive sequences or the relics derived from historical transposition and RIP events. Notably, there is no or very few authentic transposons in all putative centromeres (SI, Figure S4-S8 and Table S2). This is consistent with our recent finding that all seven putative centromeres of QM6a and CBS1-1 generated no or only a few RIP mutations upon sexual crosses of QM6a with CBS1-1 (26).

Or are there other prominent regions that could be centromeres but were considered so? If that is the case, I would suggest them to use the term "predicted centromeres" and not call them centromeres at this point.

[tfwang] Agreed. We have adopted the term "predicted centromeres" in the revised manuscript.

I would also suggest they include a figure showing specifically the organization of these regions, especially showing the low AT-rich content. Are there any transposons in them?

[tfwang] We very much appreciate this excellent suggestion. As described above, we now provide five new figures (Figure S4-S8) to fulfill this request.

The authors state "The higher numbers of AT-rich blocks in these four *Trichoderma* species might also account for (at least partly) their larger genome sizes". What is the basis behind this statement? Have they done any specific analysis to suggest that?

[tfwang] We now provide new data in Table S1 to support this statement. The overall numbers and genomic content of AT-rich blocks are 2249 and 8.95% (QM6a) (21), 2259 and 7.37% (CBS1-1), 2250 and 7.77% (CBS1-2) (26,38), 3577 and 11.33% (Gv29-8), 3367 and 11.72% (FT-333), 4570 and 13.76% (FT-101), and 5510 and 12.90% (P1), respectively.

What if some of the transposons were just RIPPed to make them AT-rich in these four species but not in other species. One would observe a higher number in that case as well.

[tfwang] CBS1-1 and CBS1-2 were just RIPPed because they were derived from two ascospores of a heterothallic fruiting body. In contrast, QM6a and the four *Trichoderma* biocontrol agents (Gv 29-8, FT333, FT101 and P1) have been propagated asexually since they were isolated. All seven genomes we studied herein contain very few authentic transposons (SI, Table S2).

The authors describe the presence of NUMTs very well but they do not mention anything about their functional part? For example, what part of mitochondrial DNA do these NUMTs belong to? Which ancestral genome are they referring to? Are these NUMT functional or have a gene sequence?

[tfwang]

1. FT-333 and Gv29-8 each contains three almost identical NUMTs in the subtelomeric regions of the left arm of their second chromosome (indicated by a black line in SI, Figure S2B). Their sequence lengths and coordinates in FT-333 are 139 bp (211,858-211,996), 166 bp (211,996-212,161) and 170 bp (115,955-116,124), whereas in Gv29-8 they are 146 bp (15,319-115,464), 168 bp (115,464-115,631) and 170 bp (115,955-116,124) (SI, Figure S23). All three NUMTs in FT-333 and Gv29-8 are located within an AT-rich block (~1500 bps in length) that lacks protein-coding sequences (SI, Figure S24).
2. The corresponding sequences of these three NUMTs are located within two mitochondrial NADH dehydrogenase subunit genes (*nad5* and *nad6*) and a mitochondrial non-coding sequence, respectively (SI, Figure S21).

In the second half of the manuscript, where the authors describe sexual development, gene clusters, and BMGs, they talk about very specific genes such as *ham5*?

[tfwang] To validate the accuracy of our genome annotation results, we first confirmed that only the *ham5* gene in QM6a encodes a truncated protein (19) (SI, Figure S10).

In some cases, they fail to describe what these genes code for or their functional relevance? Also, why did the authors specifically focus on these specific genes? There is no justification provided. People who are not familiar with the field will find it hard to understand.

[tfwang] We surveyed ~160 gene orthologs in CBS1-2 and/or three other filamentous fungal model organisms [*Neurospora crassa*, *Sordaria macrospora* (Sphaeriales, Ascomycota), *Saccharomyces cerevisiae* (Saccharomycetales, Ascomycota)] (SD, DS9). All these gene orthologs have been implicated as being involved in or even essential to fungal sexual development (see reviews of (2,3,50-53)), and their annotated functions were described in SD, DS9.

The last results section where authors describe transcriptome analysis in parts A and B can be

three separate sections. A and B can be two separate sections with the last part where they describe the evolutionarily conserved genes being the third section. Also, this section can be discussed and explain in more depth with emphasis on their biocontrol activity.

[tfwang] As suggested by the first reviewer, we have now deleted part B from the revised manuscript.

The transcriptome analysis in the sexual development could be much better described as a heat map in a figure. It is too much information in the text otherwise and table 4 and the datasets are not very helpful in clearly understanding it. Besides, if the pattern is as clean as the authors describe in the text, it will be a nice figure to have in the manuscript.

In the same section, authors should clearly define what they call as VGGs, SDIGs, ESDGs, MSDGs, LSDGs, and CSDGs. I think the definition should come first and then the classification - not the other way around as they have presented now.

[tfwang] Acknowledged. As suggested by the first reviewer, we have now deleted this entire section and the data related to transcriptional profiling of fruiting body formation from the revised manuscript.

The second paragraph of the discussion needs references.

[tfwang] Thank you. We have now added three new references to the revised manuscript:
Yadav V. et al. Centromere scission drives chromosome shuffling and reproductive isolation. (2020) PNAS 117, 7917-7928.

White, M. J. D. *Modes of Speciation*. (1978) San Francisco, CA: W. H. Freeman.

Potter, S. et al. Chromosomal speciation in the genomics era: Disentangling phylogenetic evolution of rock-wallabies. (2017) *Frontiers in Genetics*. 8:10.

Figure 1A - With the very good genome assemblies that authors generated in this study, it would be great to have a phylogenetic tree based on the whole genome data. This will provide a more confident phylogeny analysis as well.

[tfwang] Thank you. However, as recommended by the first reviewer, we have now deleted Figure 1 from the manuscript.

Figure 5 (and S9, S10, S11, S13) - Please do not use Watson-Crick nomenclature for the DNA strands. I would suggest using plus-minus strand nomenclature.

[tfwang] Agreed. We have now converted the Watson-Crick nomenclature to the plus-minus strand format.

Tables 2, 3, and 4 could be moved to the supplementary information.

[tfwang] Agreed. Tables 2, 3, and 4 are now Tables S5, S6 and S8 in the revised manuscript.

October 26, 2021

Dr. Ting-Fang Wang
Academia Sinica
Institute of Molecular Biology
Taipei, Taipei 115
Taiwan

Re: Spectrum00663-21R1 (**Complete Genome Sequences and Genome-Wide Characterization of *Trichoderma* Biocontrol Agents Provide New Insights into their Evolution and Variation in Genome Organization, Sexual Development and Fungal-Plant Interactions**)

Dear Dr. Ting-Fang Wang:

Thank you for addressing the reviewer comments in this revised manuscript. There are only a few minor issues that need to be addressed before I can recommend acceptance.

First, please release the data for PRJNA700774, including raw sequence, genome assemblies, and associated annotations. Currently there is no public record returned in NCBI for PRJNA700774. Similarly the github links for datasets do not retrieve any records (line 627-628); please make these pages public. Please also clarify if the github site is where the supplemental data sets described at line 601-602 can be accessed. Please do not resubmit the paper until all data has been made public, and the points below are addressed.

Please ensure that all methods are cited and have versions and parameters specified. Overall the methods appear quite thorough in this regard but some programs appear to be missing these specifications (ie MAFFT).

In the legend for Figure 1, please clarify how chromosomal assignment between CBS1-2 and the other genomes was carried out, ie mapping of orthologs described in the methods or by whole genome alignment.

In the title for Table S5, please change "antiSMASH" to antiSMASH, and add a footnote to spell out the abbreviations for SM-BGCs and CAZ-BGCs

Thank you for submitting your manuscript to Microbiology Spectrum. When submitting the revised version of your paper, please provide (1) point-by-point responses to the issues raised by the reviewers as file type "Response to Reviewers," not in your cover letter, and (2) a PDF file that indicates the changes from the original submission (by highlighting or underlining the changes) as file type "Marked Up Manuscript - For Review Only". Please use this link to submit your revised manuscript - we strongly recommend that you submit your paper within the next 60 days or reach out to me. Detailed information on submitting your revised paper are below.

Link Not Available

Sincerely,

Christina Cuomo

Journals Department
Reviewer comments:

Staff Comments:

Preparing Revision Guidelines

Please return the manuscript within 60 days; if you cannot complete the modification within this time period, please contact me. If you do not wish to modify the manuscript and prefer to submit it to another journal, please notify me of your decision immediately so that the manuscript may be formally withdrawn from consideration by Microbiology Spectrum.

Our point-by-point responses to editor's suggestions.

First, please release the data for PRJNA700774, including raw sequence, genome assemblies, and associated annotations. Currently there is no public records returned in NCBI for PRJNA700774. Similarly, the github links for datasets do not retrieve any records (line 627-628); please make these pages public. Please also clarify if the github site is where the supplemental data sets described at line 601-602 can be accessed. Please do not resubmit the paper until all data has been made public, and the points below are addressed.

1. NCBI has released the following genome submission(s) for PRJNA700774. all data will be available from our various Entrez servers and in Entrez genomes within a few days.
GenBank CP084943-CP084950 *Trichoderma asperellum* FT101
GenBank CP071115-CP071122 *Trichoderma virens* FT-333
GenBank CP084935-CP084942 *Trichoderma atroviride* P1
GenBank CP071107-CP071114 *Trichoderma virens* Gv29-8
2. All genomic databases data and 15 supplemental datasets are also available at the following websites: (<https://github.com/tfwangasimb/Trichoderma-biocontrol/releases>)
3. We have clarified the github site is where the supplemental data sets described at line 607-608 can be accessed.

Please ensure that all methods are cited and have versions and parameters specified. Overall the method appears quite thorough in this regard but some programs appear to be missing these specifications (i.e., MAFFT).

Acknowledged. Since we have deleted Figure 1 (i.e., phylogenetic tree) from the original manuscript, MAFFT was not used anymore in the revised manuscript.

In the legend for Figure 1, please clarify how chromosomal assignment between CBS1-2 and the other genomes was carried out, i.e., mapping of orthologs described in the methods or by whole genome alignment.

Fig. 1. Diagrammatic representations of the seven chromosomes of CBS1-2 (A), QM6a (B), Gv29-8 (C), FT-333 (D), P1 (E) and FT-101 (F). For comparative genome analyses, we identified orthologous gene pairs using the annotation results generated by Funannotate v1.8 (91) (Supplemental dataset DS8). The colors of chromosome fragments represent orthologous proteins consistent with their colors in CBS1-2 to clearly show chromosomal rearrangements. Locations of predicted centromeres are shown by restricted width. White fragments in QM6a, Gv29-8, FT-333, P1 and FT-101 represent strain-specific sequences that do not exist in CBS1-2, respectively. Locations of AT-rich blocks are indicated by black bars in the middle chromosomal maps. Black, white and gray arrows indicate disruption of synteny occurring at AT-rich blocks of the subject genomes, the CBS1-2 genome, or both, respectively.

In the title for Table S5, please change "antiSMSH" to antiSMASH, and add a footnote to spell out the abbreviations for SM-BGCs and CAZ-BGCs

[tfwang] Acknowledged.

From: gb-admin@ncbi.nlm.nih.gov
To: tfwang@gate.sinica.edu.tw , wanwan9121@gmail.com
Date: Thu, 28 Oct 2021 02:24:24
Subject: GenBank CP071107-CP071114

Dear GenBank Submitter:

We have released the following genome submission(s).

SUBID	BioProject	BioSample	Localid	Accession	Organism
SUB9047655	PRJNA700774	SAMN17838940	scaffold_1	CP071107	Trichoderma virens Gv29-8
SUB9047655	PRJNA700774	SAMN17838940	scaffold_2	CP071108	Trichoderma virens Gv29-8
SUB9047655	PRJNA700774	SAMN17838940	scaffold_3	CP071109	Trichoderma virens Gv29-8
SUB9047655	PRJNA700774	SAMN17838940	scaffold_4	CP071110	Trichoderma virens Gv29-8
SUB9047655	PRJNA700774	SAMN17838940	scaffold_5	CP071111	Trichoderma virens Gv29-8
SUB9047655	PRJNA700774	SAMN17838940	scaffold_6	CP071112	Trichoderma virens Gv29-8
SUB9047655	PRJNA700774	SAMN17838940	scaffold_7	CP071113	Trichoderma virens Gv29-8
SUB9047655	PRJNA700774	SAMN17838940	mitochondrion	CP071114	Trichoderma virens Gv29-8

Your data will be available from our various Entrez servers and in Entrez genomes within a few days.

Minor changes may have been made to your original submission in order to conform to database annotation conventions. For example:

- Spelling
- Citation data (page span, etc.)
- Product names adjusted to conform to NCBI protein naming conventions
- Taxonomic and source data

If your submission contained unpublished organism names, the scientific names have been changed to temporary names. Please notify us when the organism names are published and we will update them accordingly.

Please reply using the current Subject line.

Sincerely,

The GenBank Direct Submission Staff

genomes@ncbi.nlm.nih.gov (for updates/replies to GenBank entries)
info@ncbi.nlm.nih.gov (for general questions regarding GenBank)

From: gb-admin@ncbi.nlm.nih.gov
To: tfwang@gate.sinica.edu.tw , wanwan9121@gmail.com
Date: Thu, 28 Oct 2021 02:25:59
Subject: GenBank CP084943-CP084950

Dear GenBank Submitter:

We have released the following genome submission(s).

SUBID	BioProject	BioSample	Localid	Accession	Organism
SUB9047655	PRJNA700774	SAMN17838724	scaffold_1_1	CP084943	Trichoderma asperellum FT101
SUB9047655	PRJNA700774	SAMN17838724	scaffold_2_1	CP084944	Trichoderma asperellum FT101
SUB9047655	PRJNA700774	SAMN17838724	scaffold_3_1	CP084945	Trichoderma asperellum FT101
SUB9047655	PRJNA700774	SAMN17838724	scaffold_4_1	CP084946	Trichoderma asperellum FT101
SUB9047655	PRJNA700774	SAMN17838724	scaffold_5_1	CP084947	Trichoderma asperellum FT101
SUB9047655	PRJNA700774	SAMN17838724	scaffold_6_1	CP084948	Trichoderma asperellum FT101
SUB9047655	PRJNA700774	SAMN17838724	scaffold_7_1	CP084949	Trichoderma asperellum FT101
SUB9047655	PRJNA700774	SAMN17838724	FT101_mitochondria	CP084950	Trichoderma asperellum FT101

Your data will be available from our various Entrez servers and in Entrez genomes within a few days.

Minor changes may have been made to your original submission in order to conform to database annotation conventions. For example:

- Spelling
- Citation data (page span, etc.)
- Product names adjusted to conform to NCBI protein naming conventions
- Taxonomic and source data

If your submission contained unpublished organism names, the scientific names have been changed to temporary names. Please notify us when the organism names are published and we will update them accordingly.

Please reply using the current Subject line.

Sincerely,

The GenBank Direct Submission Staff

 genomes@ncbi.nlm.nih.gov (for updates/replies to GenBank entries)
 info@ncbi.nlm.nih.gov (for general questions regarding GenBank)

From: gb-admin@ncbi.nlm.nih.gov
To: tfwang@gate.sinica.edu.tw , wanwan9121@gmail.com
Date: Thu, 28 Oct 2021 02:25:23
Subject: GenBank CP071115-CP071122

Dear GenBank Submitter:

We have released the following genome submission(s).

SUBID	BioProject	BioSample	Localid	Accession	Organism
SUB9047655	PRJNA700774	SAMN03202112	Ch_1	CP071115	Trichoderma virens FT-333
SUB9047655	PRJNA700774	SAMN03202112	Ch_2	CP071116	Trichoderma virens FT-333
SUB9047655	PRJNA700774	SAMN03202112	Ch_3	CP071117	Trichoderma virens FT-333
SUB9047655	PRJNA700774	SAMN03202112	Ch_4	CP071118	Trichoderma virens FT-333
SUB9047655	PRJNA700774	SAMN03202112	Ch_5	CP071119	Trichoderma virens FT-333
SUB9047655	PRJNA700774	SAMN03202112	Ch_6	CP071120	Trichoderma virens FT-333
SUB9047655	PRJNA700774	SAMN03202112	Ch_7	CP071121	Trichoderma virens FT-333
SUB9047655	PRJNA700774	SAMN03202112	mitochondrion_1	CP071122	Trichoderma virens FT-333

Your data will be available from our various Entrez servers and in Entrez genomes within a few days.

Minor changes may have been made to your original submission in order to conform to database annotation conventions. For example:

- Spelling
- Citation data (page span, etc.)
- Product names adjusted to conform to NCBI protein naming conventions
- Taxonomic and source data

If your submission contained unpublished organism names, the scientific names have been changed to temporary names. Please notify us when the organism names are published and we will update them accordingly.

Please reply using the current Subject line.

Sincerely,

The GenBank Direct Submission Staff

genomes@ncbi.nlm.nih.gov (for updates/replies to GenBank entries)
info@ncbi.nlm.nih.gov (for general questions regarding GenBank)

From: gb-admin@ncbi.nlm.nih.gov
To: tfwang@gate.sinica.edu.tw , wanwan9121@gmail.com
Date: Thu, 28 Oct 2021 02:24:45
Subject: GenBank CP084935-CP084942

Dear GenBank Submitter:

We have released the following genome submission(s).

Table with 6 columns: SUBID, BioProject, BioSample, Localid, Accession, Organism. Rows list Trichoderma atroviride P1 submissions with scaffold IDs 1-7 and mitochondria.

Your data will be available from our various Entrez servers and in Entrez genomes within a few days.

Minor changes may have been made to your original submission in order to conform to database annotation conventions. For example:

- Spelling
- Citation data (page span, etc.)
- Product names adjusted to conform to NCBI protein naming conventions
- Taxonomic and source data

If your submission contained unpublished organism names, the scientific names have been changed to temporary names. Please notify us when the organism names are published and we will update them accordingly.

Please reply using the current Subject line.

Sincerely,

The GenBank Direct Submission Staff

genomes@ncbi.nlm.nih.gov (for updates/replies to GenBank entries)
info@ncbi.nlm.nih.gov (for general questions regarding GenBank)

November 9, 2021

Dr. Ting-Fang Wang
Academia Sinica
Institute of Molecular Biology
Section 2, 128 Academia Road,
Nankang
Taipei, Taipei 11529
Taiwan

Re: Spectrum00663-21R2 (**Complete Genome Sequences and Genome-Wide Characterization of *Trichoderma* Biocontrol Agents Provide New Insights into their Evolution and Variation in Genome Organization, Sexual Development and Fungal-Plant Interactions**)

Dear Dr. Ting-Fang Wang:

Your manuscript has been accepted, and I am forwarding it to the ASM Journals Department for publication. You will be notified when your proofs are ready to be viewed. A potential delay in publishing your manuscript may occur if the sequence data is not all released as soon as possible. Currently, it appears that the raw sequence is not linked to the bioproject (SRA submission of read data) and that the assemblies do not contain information of the genome annotation. Please update your submission to include the raw sequence and gene annotations, as noted in your data availability statement.

Sincerely,

Christina Cuomo
Editor, Microbiology Spectrum
